# Explicitly Modeling Censoring Produces Superior Survival Predictors

**Shi-ang Qi** [1 2 3]  **Yakun Yu** [4]  **Russell Greiner** [1 2]

## Abstract

Likelihood-based training is the dominant paradigm in survival prediction. Under independent censoring, we can factorize the likelihood and optimize only the terms related to event modeling, effectively treating the censoring mechanism as incidental. This is justified when censoring is *non-informative*, *i.e.*, when the censoring process shares no parameters with the event-time model. However, this may not hold in practice, and ignoring censoring contributions may discard useful signals for learning representations that can help to effectively estimate event distributions. Motivated by this, we argue that explicitly modeling censoring can improve representation learning and time-to-event estimation, particularly when event and censoring processes are coupled. We introduce a latent decomposition view in which observed covariates are mapped to latent components corresponding to event-specific, censoring-specific, confounding, and irrelevant information. We then learn decomposed representations for the first three categories to guide a better estimation of the event distribution. We instantiate our method on 4 popular deep-learning survival models and evaluate on 10 datasets (2 semi-synthetic and 8 real-world), showing consistent gains over strong baselines and multiple SOTA methods.

## 1. Introduction

Survival prediction is a useful tool with numerous applications in medicine (*e.g.*, time to relapse/death), engineering (*e.g.*, time to parts failure), and social sciences (*e.g.*, time to war/peace cessation). Given features $X$ for an instance, our goal is to estimate a distribution for the event time $E$.

[1]Computing Science, University of Alberta, Edmonton, Canada [2]Alberta Machine Intelligence Institute, Edmonton, Canada [3]Vector Institute, Toronto, Canada [4]Electrical Computer Engineering, University of Alberta, Edmonton, Canada. Correspondence to: Shi-ang Qi <shiang.qi@vectorinstitute.ai>.

*Proceedings of the 43rd International Conference on Machine Learning*, Seoul, South Korea. PMLR 306, 2026. Copyright 2026 by the author(s).

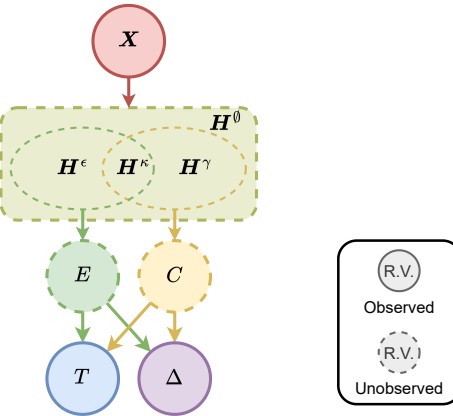

*Figure 1.* An illustration of the underlying directed acyclic graph (DAG) in a survival analysis scenario.

The key challenge in survival analysis arises from *censoring*, a common phenomenon where the exact event time is not observed for some instances. Instead, we only know a lower bound of the event time – referred to as the censoring time $C$. For example, suppose our goal is to estimate the time to cancer relapse ($E$). If patients pass away ($C$) before relapse, we only observe their death time, not the relapse time. Consequently, in a survival dataset, what we see is the observed time $T = \min\{E, C\}$, and an indicator of whether the time is an event or censored ($\Delta = \mathbb{1}[E \leq C]$) – see Figure 1.

Conventionally, survival analysis models are learned by maximizing an observed-data likelihood (Klein & Moeschberger, 2003). Under the standard *independent censoring* assumption, the likelihood admits a convenient factorization into event components and censoring components. This has led to a common practice: optimize the parameter using only the event-related terms in the likelihood, treating the censoring mechanism as incidental. This paradigm remains dominant in modern survival modeling.

In this paper, we argue that this simplification is justified only under an additional, often implicit condition: the event and censoring distributions must be *non-informative* – they do not share parameters. When this condition is violated (*i.e.*, the two processes are coupled through shared parameters), discarding censoring-related terms can lose information that is useful for identifying and efficiently estimating the event-time distribution. In such settings, explicitly mod-

eling the censoring distribution $C \mid \boldsymbol{X}$ should improve statistical efficiency and, potentially, robustness.

Moreover, motivated by representation-learning approaches that disentangle latent sources of dependence to address confounding in causal inference (Johansson et al., 2016; Shalit et al., 2017; Hassanpour & Greiner, 2019; Wu et al., 2022), we explore how to separate the latent factors that drive the event process and/or the censoring process. Specifically, we posit that a latent representation learned from the observed features $\boldsymbol{X}$ can be organized into four conceptual components; see Figure 1:

- $\boldsymbol{H}^\epsilon$ affects only the time-to-event distribution $E$;

- $\boldsymbol{H}^\kappa$ jointly influences the time to both event and censoring distributions $(E, C)$;

- $\boldsymbol{H}^\gamma$ affects only the time-to-censoring distribution $C$;

- $\boldsymbol{H}^\emptyset$ is irrelevant to either event or censoring.

For example, consider cancer relapse as the target event and death as the censoring mechanism. Different clinical factors may then contribute to different latent components: *genetic markers* may be relapse-specific (*i.e.*, $\boldsymbol{H}^\epsilon$); *cancer stage* may affect both relapse and mortality ($\boldsymbol{H}^\kappa$); and *non-cancer comorbidities* (*e.g.*, heart disease) may primarily affect mortality and hence censoring ($\boldsymbol{H}^\gamma$).

**Contributions.** The above perspective motivates two goals in observational survival data: (i) explicitly model the censoring distribution $C \mid \boldsymbol{X}$ as it might carry information about parameters of interest, and (ii) learn disentangled latent factors ($\boldsymbol{H}^\epsilon, \boldsymbol{H}^\kappa, \boldsymbol{H}^\gamma$) to better characterize and mitigate the bias introduced by partially observed event. Concretely, this paper makes the following contributions:

- Section 2 revisits the likelihood for right-censored outcomes and makes explicit the conditions under which censoring-related terms can be safely ignored.

- Section 3 proposes a simple extension (2B) of standard survival models that incorporates censoring-related information, and establishes its asymptotic efficiency advantage over the conventional approach.

- Section 4 introduces **S**urvival **A**nalysis via **La**tent **D**ecomposed representation (SALaD), which builds on 2B and learns a structured latent representation to find event-specific, censoring-specific, and confounding factors.

- Section 5 shows empirically that SALaD outperforms strong baselines, including 2B and multiple SOTA models, across 2 semi-synthetic and 8 real-world datasets. In addition, we provide evidence that SALaD recovers semantically meaningful decomposed factors.

## 2. Background

A survival dataset contains $N$ i.i.d. triplets $(\boldsymbol{x}_i, t_i, \delta_i) \sim \mathbb{P}(\boldsymbol{X}, T, \Delta)$. Here, for instance $i$, $\boldsymbol{x}_i \in \mathcal{X} \subseteq \mathbb{R}^d$ are features, $t_i \in \mathbb{R}_+$ is the observed time, and $\delta_i \in \{0, 1\}$ indicates whether the event was observed (1) or censored (0). Each instance has latent event and censoring times $e_i, c_i \in \mathbb{R}_+$. We observe $t_i = \min\{e_i, c_i\}$, $\delta_i = \mathbb{1}[e_i \leq c_i]$, where $\mathbb{1}[\cdot]$ is the indicator. Throughout, we adopt two standard assumptions (see Appendix A for details):

**Assumption 2.1** (Overlap). $0 < \mathbb{P}(\Delta = 1 \mid T, \boldsymbol{X}) < 1$.

**Assumption 2.2** (Independent censoring). $E \perp C \mid \boldsymbol{X}$.

Our goal is to estimate the *conditional event-time distribution* $E \mid \boldsymbol{X}$. Equivalently, we may characterize it via the conditional density $f_E(t \mid \boldsymbol{x}_i) = \mathbb{P}(E = t \mid \boldsymbol{X} = \boldsymbol{x}_i)$ or the conditional survival function $S_E(t \mid \boldsymbol{x}_i) = \mathbb{P}(E > t \mid \boldsymbol{X} = \boldsymbol{x}_i)$. It is also useful to define the *conditional censoring-time distribution* $C \mid \boldsymbol{X}$, through $f_C(t \mid \boldsymbol{x}_i) = \mathbb{P}(C = t \mid \boldsymbol{X} = \boldsymbol{x}_i)$ and $S_C(t \mid \boldsymbol{x}_i) = \mathbb{P}(C \geq t \mid \boldsymbol{X} = \boldsymbol{x}_i)$.

Most survival models learn parameters by maximizing the likelihood (Klein & Moeschberger, 2003). For an uncensored instance ($\delta_i = 1$), the likelihood corresponds to observing the event at time $t_i$ while censoring occurs after $t_i$; for a censored instance ($\delta_i = 0$), it corresponds to observing censoring at time $t_i$ while the event occurs after $t_i$. Aggregating across instances and using Assumption 2.2:

$$
\mathcal{L}(\boldsymbol{\omega}) = \prod_{i=1}^N \mathbb{P}(E = t_i, C \geq t_i \mid \boldsymbol{x}_i)^{\delta_i} \mathbb{P}(E > t_i, C = t_i \mid \boldsymbol{x}_i)^{1-\delta_i}
$$

$$
= \prod_{i=1}^N \left[ \mathbb{P}(E = t_i \mid \boldsymbol{x}_i) \cdot \mathbb{P}(C \geq t_i \mid \boldsymbol{x}_i) \right]^{\delta_i}
$$
$$
\times \left[ \mathbb{P}(E > t_i \mid \boldsymbol{x}_i) \cdot \mathbb{P}(C = t_i \mid \boldsymbol{x}_i) \right]^{1-\delta_i}
$$
$$
= \prod_{i=1}^N \left[ \widehat{f}_E(t_i \mid \boldsymbol{x}_i) \widehat{S}_C(t_i \mid \boldsymbol{x}_i) \right]^{\delta_i} \left[ \widehat{S}_E(t_i \mid \boldsymbol{x}_i) \widehat{f}_C(t_i \mid \boldsymbol{x}_i) \right]^{1-\delta_i},
$$
(1)

where $\widehat{\cdot}$ denotes model predictions, parameterized by $\boldsymbol{\omega}$.

In practice, the censoring mechanism is treated as incidental; consequently, it is common to discard the terms involving $C$ and optimize only

$$
\mathcal{L}(\boldsymbol{\omega}) \propto \prod_{i=1}^N \widehat{f}_E(t_i \mid \boldsymbol{x}_i)^{\delta_i} \cdot \widehat{S}_E(t_i \mid \boldsymbol{x}_i)^{1-\delta_i}. \quad (2)
$$

This *censoring-mechanism-free likelihood* underlies a large family of classical and deep survival models (Cox, 1972; Wei, 1992; Katzman et al., 2018; Zhong et al., 2021; Pearce et al., 2022; Norman et al., 2024). In deep learning formulations, a feature encoder $\Psi(\cdot)$ (*e.g.*, MLP, CNN, RNN, Transformer) is followed by a "survival layer" (Chen et al., 2024) to produce time-to-event predictions; see Figure 2a.

However, discarding the censoring terms is not *purely* justified by independent censoring. It additionally requires that

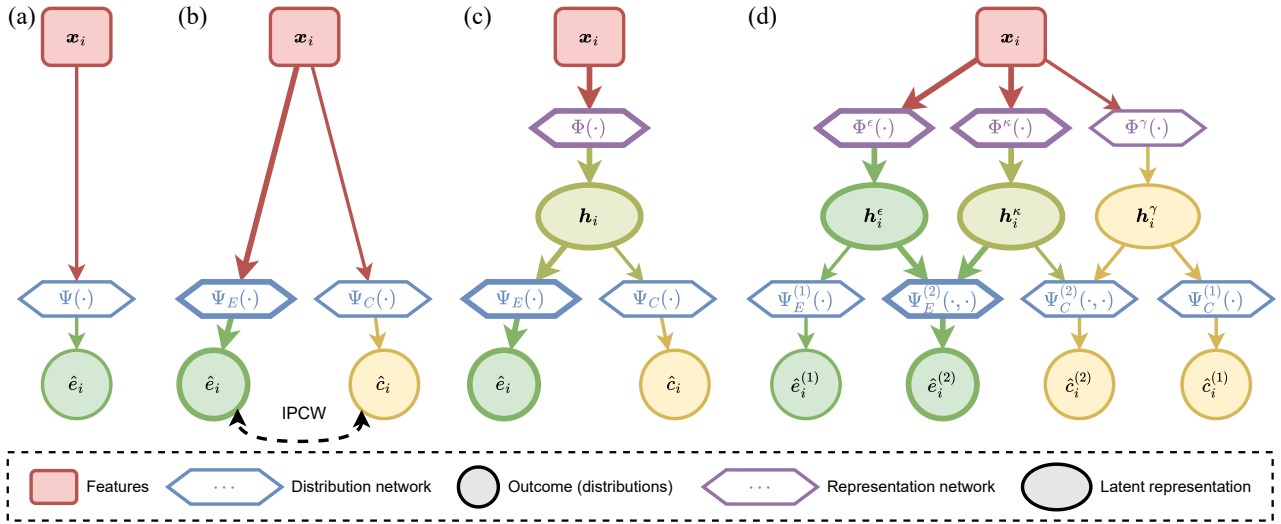

*Figure 2.* Overview of survival-model architectures. (a) Standard deep learning survival models. (b) Models incorporating IPCW to adjust for censoring. (c) The proposed `2B` framework. (d) The proposed `SALaD` framework. In (a), the full architecture is used for both training and inference. In (b) – (d), all components are used during training, whereas inference uses only the modules shown in **bold**.

the censoring contribution be constant with respect to the parameters governing $E \mid \boldsymbol{X}$. We state this as a separate assumption:

**Assumption 2.3** (Non-informative censoring)**.** The censoring distribution $(C \mid \boldsymbol{X})$ does not involve parameters that govern the event-time distribution $(E \mid \boldsymbol{X})$.

The literature sometimes conflates Assumption 2.2 with Assumption 2.3, or implicitly treats them as equivalent. We emphasize that these assumptions are *orthogonal*: neither implies the other (Kleinbaum & Klein, 2012). To illustrate, consider an exaggerated example where $E$ and $C$ are independent but informative (share parameters):

$$E \sim \text{Exp}(\lambda_0), \qquad C \sim \text{Exp}(\lambda_0), \qquad E \perp C.$$

Although $E$ and $C$ are independent, the event and censoring distributions are coupled through the shared rate $\lambda_0$; thus censoring is *informative* in the sense of Assumption 2.3 being violated. Appendix A provides additional examples and discussion.

Recent approaches (Cole & Hernán, 2004; Vock et al., 2016a; Han et al., 2021; Alberge et al., 2025) incorporate inverse probability of censoring weighting (IPCW) (Robins & Finkelstein, 2000) to adjust for censoring when optimizing prediction objectives. As shown in Figure 2b, these methods typically learn event and censoring models separately and then use an IPCW factor to reweight event-related losses. However, most IPCW-based formulations still implicitly rely on Assumption 2.3: the event model is optimized using weighted event terms, while the censoring model is trained in a separate stage (or as an auxiliary model) without parameter sharing. As a result, their statistical properties depend

critically on accurately estimating $S_C(\cdot \mid \boldsymbol{X})$. When censoring is informative through shared parameters, estimating $C \mid \boldsymbol{X}$ efficiently may itself require leveraging information from the event process – suggesting that a fully decoupled, two-stage procedure can be inefficient or brittle.

## 3. A Straightforward Extension

From a causal perspective, to use the backdoor criterion (Pearl, 2009) for identifying the bias of censoring on event time, one must condition on a set of variables that collectively block all "backdoor paths" between $C$ and $E$. In other words, after dealing with any confounders, any systematic correlation between $E$ and $C$ reflects the effect of $C$ on $E$.

In the above exaggerated example, the censoring observations are *informative* about the event-time parameters: if the parametric form is correctly specified, maximizing the *full* likelihood should exploit this additional information and reach a high-precision estimate of the shared parameter more rapidly than censoring-mechanism-free training.

This motivates a simple architectural change (depicted in Figure 2c): rather than treating the censoring model as an external nuisance component, we *jointly* model the event and censoring processes with a unified objective. Concretely, we use a shared representation network $\Phi(\cdot)$ to map features $\boldsymbol{X}$ to a latent representation that may contain information about $\boldsymbol{H}^\epsilon$, $\boldsymbol{H}^\kappa$, and $\boldsymbol{H}^\gamma$. This shared representation is then fed into two distribution heads, $\Psi_E(\cdot)$ and $\Psi_C(\cdot)$, which parameterize the event-time and censoring-time distributions, respectively. All parameters are trained end-to-end by maximizing the *full* likelihood in (1), which includes

both event and censoring contributions. We refer to this two-head, jointly trained framework as 2B (*two branches*).

From a causal perspective, censoring-induced bias can be viewed as arising from a spurious association between $E$ and $C$ through shared causes. Under the backdoor criterion (Pearl, 2009), eliminating such bias requires conditioning on variables that block all backdoor paths between $E$ and $C$. In representation-learning terms, learning a shared encoder $\Phi$ using *both* event and censoring distributions encourages $\Phi(\boldsymbol{X})$ to capture the common parameters and factors of $E$ and $C$, thereby reducing residual dependence and improving training efficiency. However, to our best knowledge, this straightforward extension has not been rigorously analyzed in the survival literature – see Section 6.

Let $\boldsymbol{\omega}_E$ denote the parameters relevant to the event distribution in 2B, *i.e.*, $\boldsymbol{\omega}_E = (\Phi, \Psi_E)$.[1] We next show that incorporating the censoring head during training cannot worsen asymptotic precision for $\boldsymbol{\omega}_E$, and it typically improves precision when the censoring distribution is informative.

**Theorem 3.1** (Efficiency gain from modeling censoring). *Assume independence (Assumption 2.2) and standard regularity conditions (Assumptions B.1 – B.2). Let $\widehat{\boldsymbol{\omega}}_E$ be the MLE obtained by maximizing the full likelihood in* (1)*, and let $\widehat{\boldsymbol{\omega}}_E^*$ be the empirical risk minimization (ERM) estimator obtained by maximizing* (2)*. Then, the asymptotic covariance matrix of $\widehat{\boldsymbol{\omega}}_E$ is smaller in the Loewner order:*

$$\Sigma\big(\widehat{\boldsymbol{\omega}}_E\big) \preceq \Sigma\big(\widehat{\boldsymbol{\omega}}_E^*\big),$$

*with strict inequality when Assumption 2.3 is violated.*

**Proof sketch.** The technical step compares the score (gradient) contributions under the two objectives. Viewing the censoring-mechanism-free score and the additional censoring-mechanism score as jointly distributed random vectors, their joint covariance matrix must be positive semidefinite. A Schur-complement argument then implies that the Fisher information for $\boldsymbol{\omega}_E$ under the full likelihood is at least as large as the information available under the censoring-mechanism-free objective. A complete proof is provided in Appendix B.

Intuitively, even though the censoring-mechanism-free likelihood is consistent, the full likelihood exploits additional score terms that can carry information about $\boldsymbol{\omega}_E$ through the shared encoder $\Phi$. When the censoring process depends on $\Phi$ (shares parameters with the event model), this extra information reduces asymptotic variance and leads to an efficiency gain. Empirical results in Section 5 also show that 2B leads to better models, serving as a proof-of-concept that explicitly modeling the censoring distribution can strengthen representation learning in survival modeling.

---

[1]We slightly abuse notation by using $\Phi$ and $\Psi_E$ to denote both network components and their associated parameters.

## 4. Latent Decomposed Representation

We now describe **S**urvival **A**nalysis via **La**tent **D**ecomposed Representation (SALaD), our framework for estimating time-to-event distributions. Section 4.1 summarizes the key propositions implied by the assumed causal structure. Section 4.2 then presents the overall SALaD architecture and explains each component in detail. Appendix C offers additional details of the proposed method. Source code is available at `https://github.com/shi-ang/SALaD`.

### 4.1. Propositions

In Section 1, we posit that the features $\boldsymbol{X}$ are inputs from which the model learns a latent representation $\boldsymbol{H} = \Phi(\boldsymbol{X})$, which we conceptually decompose into four components: $\boldsymbol{H}^\epsilon$ (event-specific factors), $\boldsymbol{H}^\kappa$ (confounding factors), $\boldsymbol{H}^\gamma$ (censoring-specific factors), and $\boldsymbol{H}^\emptyset$ (irrelevant/noise factors); see Figure 1. Under this view, the dependency between event and censoring mechanisms is driven by the confounding factors $\boldsymbol{H}^\kappa$. In practice, some representation sets may be empty, and the exact mapping (a latent re-expression) is unknown in practice.

The assumed DAG in Figure 1 implies the following **Propositions**, which serve as design principles for learning a decomposed representation:

**Prop. 4.1.** *The event and censoring times are conditionally independent given the confounding factors:* $E \perp C \mid \boldsymbol{H}^\kappa$

**Prop. 4.2.** *The censoring-specific factors carry no information about the event time:* $\boldsymbol{H}^\gamma \perp E$.

**Prop. 4.3.** *The event-specific factors carry no information about the censoring time:* $\boldsymbol{H}^\epsilon \perp C$.

**Prop. 4.4.** *The representations $\boldsymbol{H}^\epsilon$, $\boldsymbol{H}^\kappa$, and $\boldsymbol{H}^\gamma$ encode distinct, non-overlapping information (up to identifiability; not guaranteed without additional structural assumptions).*

A natural question is whether Assumption 2.2 ($E \perp C \mid \boldsymbol{X}$) already suggests a sufficient solution: condition on $\boldsymbol{X}$ directly, as in Figure 2c. However, two issues arise. First, when censoring is informative through shared parameters, modeling censoring can be statistically useful for learning the *shared* representation, and a decomposition makes explicit which latent information extracted from $\boldsymbol{X}$ should be shared across the event and censoring models. Second, from a statistical perspective, adjusting for irrelevant or outcome-independent covariates can degrade estimation quality by increasing finite-sample bias (Abadie & Imbens, 2006) and/or inflating variance (Hahn, 1998). Therefore, a principled decomposition of $\boldsymbol{X}$ into $\{\boldsymbol{H}^\epsilon, \boldsymbol{H}^\kappa, \boldsymbol{H}^\gamma, \boldsymbol{H}^\emptyset\}$ is essential.

Once such latent components are learned, then removing $\boldsymbol{H}^\gamma$ and $\boldsymbol{H}^\emptyset$ would not reduce information about the event-time distribution. Therefore, the event distribution can be modeled using only $(\boldsymbol{H}^\epsilon, \boldsymbol{H}^\kappa)$.

### 4.2. Solutions for Propositions

Motivated by our preliminary propositions, we introduce SALaD, a survival-layer-agnostic framework that estimates the event distribution by (i) learning decomposed latent representations for event-specific, confounding, and censoring-specific factors, and (ii) predicting the event distribution using only the event-relevant factors. As illustrated in Figure 2d, SALaD consists of three components.[2]

- **Three Representation Networks.** We use three separate encoders, $\Phi^\epsilon(\cdot)$, $\Phi^\kappa(\cdot)$, and $\Phi^\gamma(\cdot)$, to learn event-specific, confounding, and censoring-specific representations, respectively: $\boldsymbol{h}_i^\epsilon = \Phi^\epsilon(\boldsymbol{x}_i)$, $\boldsymbol{h}_i^\kappa = \Phi^\kappa(\boldsymbol{x}_i)$, $\boldsymbol{h}_i^\gamma = \Phi^\gamma(\boldsymbol{x}_i)$.

- **Four Distribution Networks.** These networks generate outputs used to calculate time-to-event or time-to-censoring distributions, such as risk scores, hazard functions, density functions, or survival functions. Each distribution network corresponds to a unique likelihood loss, enhancing the predictive power of the learned latent representations.

- **Three Decomposition Regularizers.** To encourage separation of the latent factors, we introduce three regularizers: two to discourage event-specific (resp. censoring-specific) representations from encoding information predictive of the *other* process, and a third promotes non-overlap across the learned representations.

The following sections elaborate on each component.

#### 4.2.1. SOLUTION FOR PROPOSITION 4.1

Proposition 4.1 suggests that conditioning on the confounding factors is sufficient to block the dependence between the event and censoring mechanisms. Accordingly, once we obtain $\boldsymbol{h}_i^\epsilon$, $\boldsymbol{h}_i^\kappa$, and $\boldsymbol{h}_i^\gamma$, we parameterize the event-time distribution using $(\boldsymbol{h}_i^\epsilon, \boldsymbol{h}_i^\kappa)$ and the censoring-time distribution using $(\boldsymbol{h}_i^\gamma, \boldsymbol{h}_i^\kappa)$:

$$\widehat{S}_E^{(2)}(t \mid \boldsymbol{h}_i^\epsilon, \boldsymbol{h}_i^\kappa), \qquad \widehat{S}_C^{(2)}(t \mid \boldsymbol{h}_i^\gamma, \boldsymbol{h}_i^\kappa).$$

We can find the parameters for each by maximizing the corresponding full likelihood:

$$\mathcal{L}^{(2)}\Big(\Phi^\epsilon, \Phi^\kappa, \Phi^\gamma, \Psi_E^{(2)}, \Psi_C^{(2)}\Big) = \prod_{i=1}^N \Big[\widehat{f}_E^{(2)}(t_i \mid \boldsymbol{h}_i^\epsilon, \boldsymbol{h}_i^\kappa) \cdot$$
$$\widehat{S}_C^{(2)}(t_i \mid \boldsymbol{h}_i^\gamma, \boldsymbol{h}_i^\kappa)\Big]^{\delta_i} \Big[\widehat{f}_C^{(2)}(t_i \mid \boldsymbol{h}_i^\gamma, \boldsymbol{h}_i^\kappa)\widehat{S}_E^{(2)}(t_i \mid \boldsymbol{h}_i^\epsilon, \boldsymbol{h}_i^\kappa)\Big]^{1-\delta_i}.$$
$$(3)$$

---

[2]The integer superscript on networks and predictions (*e.g.*, $e_i^{(1)}$ and $e_i^{(2)}$) indicates whether the prediction is conditioned on one or two types of latent representations.

Our framework is versatile, as it can naturally simplify into a more basic structure when certain latent components are empty. For instance, if $\boldsymbol{H}^\kappa = \emptyset$, then $E$ and $C$ share no dependency nor common parameters in our latent view; the model reduces to a decoupled event/censoring formulation (cf. Figure 2b), and there is no need to use shared representations or IPCW to adjust for censoring dependence. Conversely, if $\boldsymbol{H}^\epsilon = \boldsymbol{H}^\gamma = \emptyset$, then all predictive signal is shared and the architecture collapses to the 2B formulation in Figure 2c.

**Encouraging task-specific factors.** However, in the general scenario, the estimated $\boldsymbol{H}^\kappa$ can absorb excessive information for event-specific and censoring-specific – leading to weak $\boldsymbol{H}^\epsilon$ and $\boldsymbol{H}^\gamma$ (in the extreme, effectively empty). To address this, we introduce two additional distribution networks, $\Psi_E^{(1)}(\cdot)$ and $\Psi_C^{(1)}(\cdot)$, which predict the event distribution using only the event-specific representation and the censoring distribution using only the censoring-specific representation:

$$\widehat{S}_E^{(1)}(t \mid \boldsymbol{h}_i^\epsilon), \qquad \widehat{S}_C^{(1)}(t \mid \boldsymbol{h}_i^\gamma).$$

These auxiliary objectives encourage $\boldsymbol{H}^\epsilon$ and $\boldsymbol{H}^\gamma$ to capture information that is directly predictive of their respective processes. We optimize them via the additional likelihood:

$$\mathcal{L}^{(1)}\Big(\Phi^\epsilon, \Phi^\gamma, \Psi_E^{(1)}, \Psi_C^{(1)}\Big) = \prod_{i=1}^N \Big[\widehat{f}_E^{(1)}(t_i \mid \boldsymbol{h}_i^\epsilon) \cdot \widehat{S}_C^{(1)}(t_i \mid \boldsymbol{h}_i^\gamma)\Big]^{\delta_i}$$
$$\times \Big[\widehat{f}_C^{(1)}(t_i \mid \boldsymbol{h}_i^\gamma) \cdot \widehat{S}_E^{(1)}(t_i \mid \boldsymbol{h}_i^\epsilon)\Big]^{1-\delta_i}.$$
$$(4)$$

#### 4.2.2. SOLUTION FOR PROPOSITIONS 4.2 AND 4.3

The likelihood objectives in Section 4.2.1 encourage $\boldsymbol{H}^\epsilon$ to be predictive of event times, $\boldsymbol{H}^\gamma$ to be predictive of censoring times, and $\boldsymbol{H}^\kappa$ to capture shared drivers. However, these objectives alone do not prevent *information leakage*: in practice, any representation may still encode factors that belong to the other components (and in the extreme, all three representations can collapse to encoding the same information). To enforce Propositions 4.2 and 4.3, we introduce two additional regularizers that explicitly discourage $\boldsymbol{H}^\gamma$ from containing information about $E$ and discourage $\boldsymbol{H}^\epsilon$ from containing information about $C$.

**Key idea: invariance across early/late groups.** Consider Proposition 4.2, which states $\boldsymbol{H}^\gamma \perp E$. Operationally, this means that knowing $\boldsymbol{H}^\gamma$ should not help distinguish whether an instance's event time is *early* or *late*. Equivalently, the distribution of $\boldsymbol{H}^\gamma$ should be (approximately) the same across early- and late-event groups. We enforce this by splitting instances into early vs. late groups and penalizing the discrepancy between the corresponding representation distributions. A symmetric construction enforces

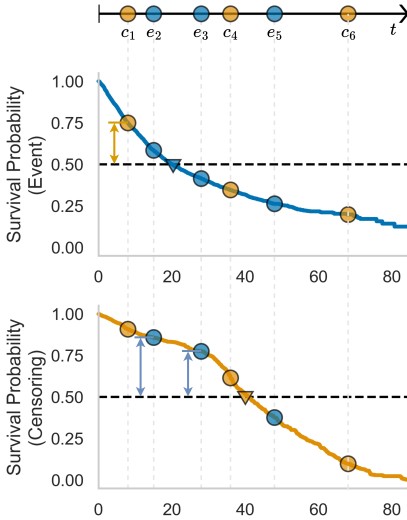

*Figure 3.* Partitioning a batch of survival data. *Top:* 6 instances ordered by observed time, showing the censored ($c$) and event ($e$) times. *Middle:* the marginal event-time survival function $S_E(t)$. *Bottom:* the marginal censoring-time survival function $S_C(t)$. The dashed horizontal lines indicate the 50% probability. The gray dashed vertical lines align points corresponding to same instances.

Proposition 4.3 by making $\boldsymbol{H}^\epsilon$ invariant to early versus late censoring times.

A challenge is that event times are unobserved for censored instances, so group membership is partially unknown. Assume for the moment access to the oracle marginal event-time survival function $S_E(t)$ over the training set, and let $\overline{e}$ denote its median time (triangle in Figure 3, middle). Each instance then falls into one of four cases:

1. **Uncensored and before** $\overline{e}$ (*e.g.*, $e_2$): belongs to the early group with probability 1;

2. **Uncensored and after** $\overline{e}$ (*e.g.*, $e_3, e_5$): belongs to the late group with probability 1.

3. **Censored after** $\overline{e}$ (*e.g.*, $c_4, c_6$): belongs to the late group with probability 1 (since $\overline{e} \leq t_i < e_i$).

4. **Censored before** $\overline{e}$ (*e.g.*, $c_1$): could be early or late. Using $t_i = c_i < e_i$ and the marginal survival curve,

$$\mathbb{P}(E < \overline{e} \mid E > t_1) = \frac{\mathbb{P}(c_1 < E < \overline{e})}{\mathbb{P}(E > c_1)} = \frac{S_E(c_1) - 0.5}{S_E(c_1)}.$$

Intuitively, as illustrated in Figure 3(middle), this probability is the fraction of remaining probability mass that lies before the median (the arrow relative to $c_1$'s total vertical span).

The above cases define the probability that instance $i$ be-

longs to the early-event group:

$$W_E(t_i, \delta_i) := \mathbb{P}(E < \overline{e} \mid T = t_i, \Delta = \delta_i) \quad (5)$$
$$= \mathbb{1}[S_E(t_i) > 0.5]\left(\delta_i + (1 - \delta_i)\frac{S_E(t_i) - 0.5}{S_E(t_i)}\right).$$

We then enforce Proposition 4.2 by minimizing the discrepancy between $\boldsymbol{H}^\gamma$ for the early- and late-event groups:

$$\mathcal{L}_\gamma(\Phi^\gamma) = \mathtt{disc}\bigg(\{W_E(t_i, \delta_i) \cdot \Phi^\gamma(\boldsymbol{x}_i)\}_{i=1}^N,$$
$$\{(1 - W_E(t_i, \delta_i)) \cdot \Phi^\gamma(\boldsymbol{x}_i)\}_{i=1}^N\bigg), \quad (6)$$

where $\mathtt{disc}(\cdot, \cdot)$ is a chosen discrepancy function. We use a weighted maximum mean discrepancy with a linear or an RBF kernel (Gretton et al., 2012) (see Appendix C.1).

Similarly, to enforce Proposition 4.3, we also split instances into early vs. late *censoring*-time groups and minimize the discrepancy between the corresponding event-specific representations:

$$\mathcal{L}_\epsilon(\Phi^\epsilon) = \mathtt{disc}\bigg(\{W_C(t_i, \delta_i) \cdot \Phi^\epsilon(\boldsymbol{x}_i)\}_{i=1}^N,$$
$$\{(1 - W_C(t_i, \delta_i)) \cdot \Phi^\epsilon(\boldsymbol{x}_i)\}_{i=1}^N\bigg), \quad (7)$$

where $W_C(t_i, \delta_i)$ is defined analogously using the marginal censoring survival function (Appendix C.2).

*Remark* 4.5. In practice, we approximate the oracle marginal survival functions $S_E(t)$ and $S_C(t)$ using Kaplan-Meier (KM, 1958); see Appendix C.3. While KM is consistent under *random* censoring ($E \perp C$), it can be biased when censoring depends on features (Assumption 2.2) (Campigotto & Weller, 2014).

Under Assumption 2.2, researchers may resort to 3 alternative methods for estimating unbiased marginal survival functions: (i) a conditional estimator to calculate IPCW (Dong et al., 2020), (ii) jointly modeling the event and censoring processes via a copula (Li et al., 2020), or (iii) analyzing the censoring cause of each individual in a pattern-mixture model (Li et al., 2020). However, (i) and (ii) are impractical because estimating IPCW or copula is as hard as directly modeling the time-to-event, and the third approach is often infeasible because real-world datasets frequently lack detailed censoring-cause information.

Importantly, our use of marginal survival curves is limited to *forming early/late groups*. If KM underestimates the true survival curve, the median computed from KM may no longer correspond to the true 50% quantile, leading mainly to an *imbalanced* split (*e.g.*, 30/70 rather than 50/50). This imbalance does not invalidate our discrepancy regularizers, which compare representation distributions across groups

and do not require equal group sizes. The main sensitivity arises only in Case 4, where grouping for a censored instance is probabilistic and depends on the estimated curve. Appendix C.4 discusses this further.

Finally, we also explored adversarial training as an alternative invariance mechanism; empirically it underperformed our discrepancy-based regularizers (Appendix E).

### 4.2.3. SOLUTION FOR PROPOSITION 4.4

The previous sections encourage $H^\epsilon$ (resp., $H^\gamma$) to retain only event-specific (resp., censoring-specific) information. What remains is to ensure that the shared representation $H^\kappa$ captures *only* the confounding factors – rather than duplicating information already present in $H^\epsilon$ or $H^\gamma$. To this end, we impose an explicit non-overlap constraint across the three representations, corresponding to Proposition 4.4.

A variety of independence/orthogonality penalties could be used for disentanglement. We adopt the *orthogonal regularizer for variable decomposition* (Appendix C.5) (Kuang et al., 2020; Xu et al., 2021; Wu et al., 2022) as a soft feature-usage penalty: it discourages overlap in the input directions used by the three representation networks, thereby supporting the separation of the learned latent components. This encourages non-overlapping use of input signal across the learned encoders, reduces leakage across latent representations, and helps suppress input directions not useful for any latent components (Wu et al., 2022).

### 4.2.4. OBJECTIVE FUNCTION

The overall objective function we want to minimize includes all the components above:

$$\mathcal{L}_{\text{total}} = -\log\big(\mathcal{L}^{(2)}\mathcal{L}^{(1)}\big) + \beta_1\mathcal{L}_{\text{orth}} + \beta_2(\mathcal{L}^\gamma + \mathcal{L}^\epsilon) + \lambda\mathcal{L}_{\text{reg}},$$

where $\mathcal{L}^{(2)}$, $\mathcal{L}^{(1)}$, $\mathcal{L}_\gamma$, and $\mathcal{L}_\epsilon$ correspond to (3), (4), (6), and (7), respectively. $\mathcal{L}_{\text{orth}}$ is the orthogonal penalty, and $\mathcal{L}_{\text{reg}}$ denotes standard $\ell_2$ weight decay applied to all parameters.

## 5. Experiments

**Datasets** We perform the experiments on 2 semi-synthetic datasets and 8 real-world datasets: **semi-SUPPORT**, **semi-METABRIC**, **HFCR**, **PBC**, **GBM**, **GBSG**, **METABRIC**, **NACD**, **SUPPORT**, and **MIMIC-IV**. Appendix D.1 provides dataset descriptions and preprocessing details. Due to space constraints, the main text reports representative results on **semi-SUPPORT**, **semi-METABRIC**, **HFCR** and **SUPPORT**; Appendix E presents the full results.

**Evaluation** We evaluate models using concordance index (CI) (Harrell Jr et al., 1996), integrated Brier score (IBS) (Graf et al., 1999), mean absolute error (MAE) (Qi et al., 2023a), and distribution calibration (D-cal) (Haider

et al., 2020); see Appendix D.2 for details.

**Benchmarks** We compare SALaD (Figure 2d) against (i) standard deep survival baselines (Figure 2a) and (ii) their two-branch extensions 2B (Figure 2c). We instantiate SALaD and 2B on four widely used neural survival models: DeepSurv (Katzman et al., 2018), N-MTLR (Fotso, 2018), AFTNN-Weibull (Norman et al., 2024), and AFTNN-LogLogistic (Norman et al., 2024).

In addition, we benchmark against 10 SOTA survival models: Nnet-survival (Gensheimer & Narasimhan, 2019), RSF (Ishwaran et al., 2008), GB (Hothorn et al., 2006), DeepHit (Lee et al., 2018), CoxTime (Kvamme et al., 2019), IWSG (Han et al., 2021), SODEN (Tang et al., 2022), CQRNN (Pearce et al., 2022), DCSurvival (Zhang et al., 2024) and SurvivalBoost (Alberge et al., 2025). See Appendix D.3 for details.

### 5.1. Main Results

**Compare with Baselines** Across datasets, SALaD consistently improves over its corresponding baselines, indicating that explicitly learning decomposed representations yields more accurate time-to-event estimation. As shown in Figure 4, SALaD attains higher CI and lower IBS/MAE in most settings. For D-cal, SALaD is comparable to or better than the baselines, with occasional gains. Aggregating all results (10 datasets × 4 base models × 4 metrics – 4 cases = 156 comparisons), SALaD achieves 109 wins (*i.e.*, better in the mean performance), 28 ties, and 19 losses, demonstrating robust improvements across metrics and datasets.

**Compare with SOTA** Enhancing baselines with SALaD yields performance that is competitive with, and often surpasses, existing SOTAs. For example, on **HFCR**, SALaD (DeepSurv) attains the highest CI, while SALaD (N-MTLR) achieves the lowest IBS and MAE, reflecting strong discrimination and accurate survival probability estimation. These results suggest that SALaD provides a general-purpose upgrade for deep survival models without requiring model-specific redesign.

**Effect of Decomposed Representations** To assess whether SALaD learns the intended semantics, we visualize the learned representations using t-SNE on the **semi-METABRIC** dataset and the real **SUPPORT** dataset (Figures 5 and 6). In both cases, the event-specific representation shows minimal separation wrt censoring times (lower left panel); the censoring-specific representation shows minimal separation wrt event times (upper right panel); and the confounder representation separates both (middle panels).

### 5.2. Ablation Studies

We study the contribution of key components: (1) how SALaD improves upon 2B, (2) the impact of the distance-

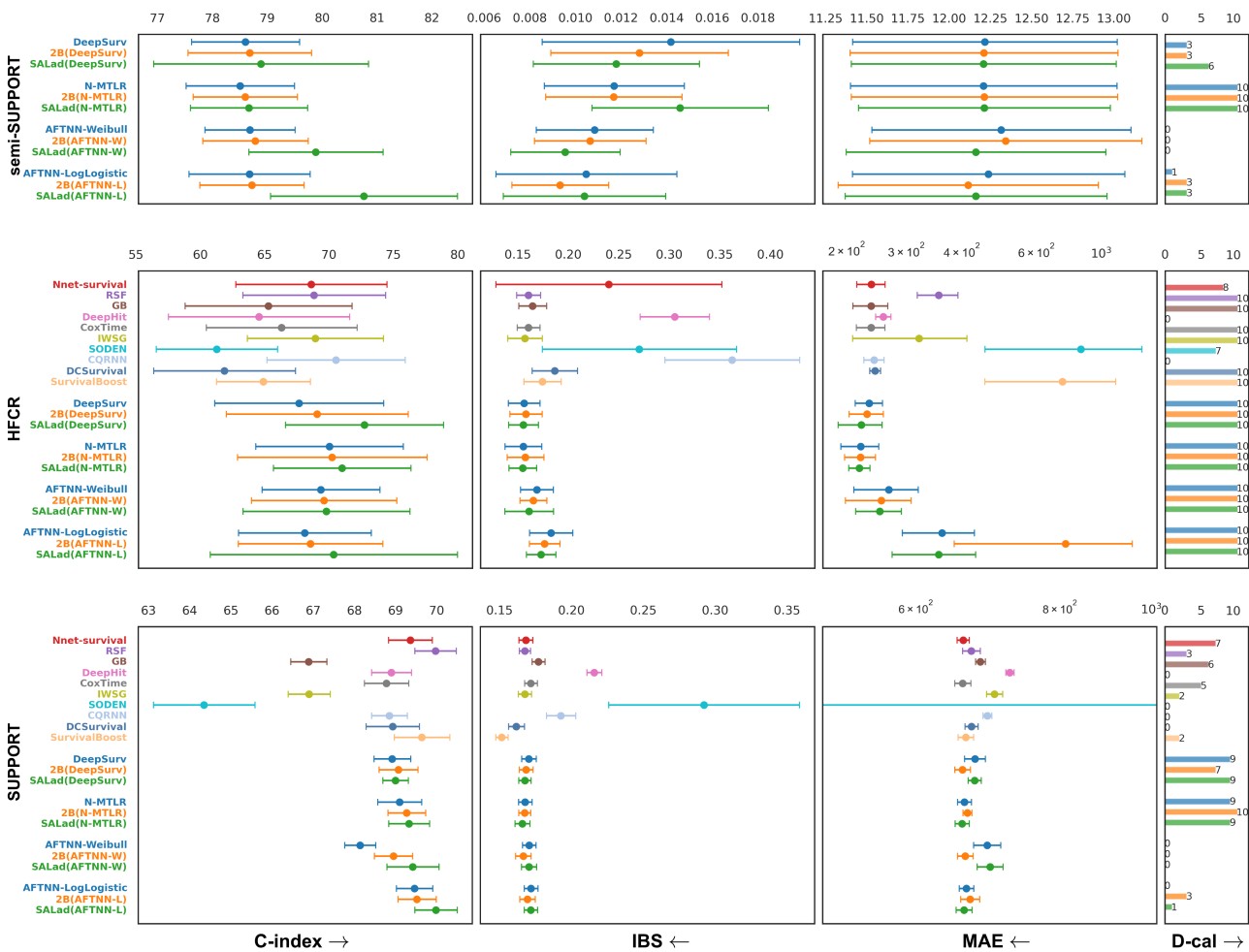

*Figure 4.* Results on **semi-SUPPORT**, **HFCR** and **SUPPORT**, reported as mean $\pm$ 95% confidence interval over 10 runs.

based invariance regularizers, and (3) the role of orthogonal decomposition. Full ablation results are in Appendix E.

**Compare with 2B** As shown in Figure 4, the simple 2B extension already improves over event-only baselines, highlighting the benefit of modeling censoring. Nevertheless, 2B is generally inferior to SALaD, indicating that explicitly disentangling representations provides additional gains. In some cases, 2B yields slightly better calibration (lower IBS and higher D-cal), plausibly due to its simpler hypothesis class, which can reduce overfitting in finite samples (Wang et al., 2021).

**Effect of the Distance Regularizers** Removing the distance-based invariance regularizers ($\beta_2 = 0$) consistently degrades performance.

**Effect of the Orthogonal Regularizer** Disabling the orthogonal decomposition penalty ($\beta_1 = 0$) also leads to consistent drops across metrics.

### 5.3. Complexity Analysis

Appendix E.5 provides the complexity analysis. Overall, SALaD incurs a moderate increase in parameter count and training time compared to 2B and baselines, while remaining comparable to other SOTA models in computation.

## 6. Related Work

We summarize prior work that leverages representation learning in survival analysis, with an emphasis on approaches that model event and censoring mechanisms or attempt disentanglement.

Engelhard & Henao (2022) proposed a disentangled mixture cure model that learns separate representations intended to capture event-related, censoring-related, and susceptibility-related factors, and uses them to estimate time-to-event distributions. While conceptually related, their formulation suffers from several key limitations. First, the model does not explicitly enforce a strict separation between event-

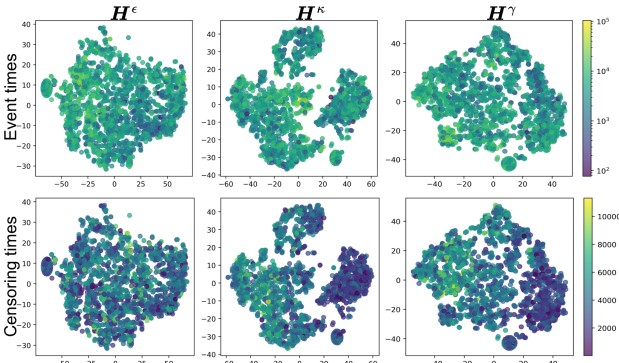

*Figure 5.* 2D t-SNE visualization of latent representations learned by `SALaD` (`N-MTLR`) on the **semi-METABRIC** dataset. The color represents the value of (event or censoring) time.

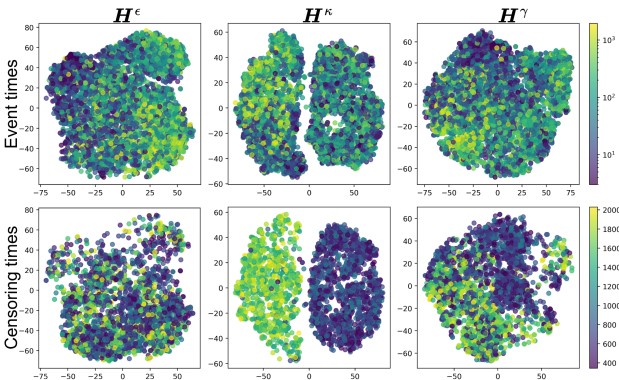

*Figure 6.* 2D t-SNE visualization of latent representations learned by `SALaD` (`N-MTLR`) on **SUPPORT**, same notation as Figure 5.

and censoring-related representations (beyond architectural inductive bias), whereas `SALaD` introduces dedicated invariance and orthogonality regularizers to promote factor-specific information. Second, their approach implicitly aligns with a non-informative censoring regime in the sense that censoring-related modeling is not used to identify event-time parameters through a shared mechanism. Finally, the cure-model assumption allows a non-susceptible subpopulation that never experiences the event, which is incompatible with our Assumption 2.1. As a result, their empirical evaluation is primarily conducted in settings where susceptibility is well-defined or can be simulated, which may not match many standard right-censoring applications.

Chapfuwa et al. (2021) studied counterfactual survival analysis and proposed a representation-learning architecture designed to address both selection bias and censoring-induced bias. Their model uses a shared representation followed by four outcome heads corresponding to treatment status (control vs. treated) and outcome type (event vs. censoring), *i.e.*, control-event, control-censoring, treated-event, and treated-censoring. This branching structure is related in spirit to our `2B` formulation (Figure 2c), though their primary objective

is causal estimation under treatment assignment.

Cui et al. (2025) introduced a disentangled model for longitudinal competing-risks survival analysis, using contrastive learning to induce cause-specific representations (one representation per competing risk). Similar to prior disentanglement-based formulations, their method does not explicitly provide guarantees of factor separation, implicitly assumes non-informative censoring, and typically relies on structural assumptions about the competing risks.

A growing body of applied work develops representation learning pipelines for multimodal healthcare survival prediction (Cheerla & Gevaert, 2019; Wu et al., 2023; Farooq et al., 2025). These approaches focus on learning informative modality-specific embeddings (*e.g.*, from clinical notes, imaging, and omics), fusing them (often by concatenation), and then applying standard survival modeling layers (*e.g.*, `CoxPH`). The core innovation in these works lies in the multimodal representation learning rather than in advancing survival analysis methodology itself. In contrast, our focus is on improving survival modeling under censoring by explicitly decomposing latent factors and coupling event/censoring learning objectives.

To our best knowledge, this work is the first to: (i) systematically investigate the benefit of *joint* event-censoring modeling via a unified objective (`2B` framework; Figure 2c), and (ii) propose and operationalize a decomposition into event-specific, censoring-specific, and shared/confounding latent factors (`SALaD` framework; Figure 2d), with extensive empirical evidence demonstrating consistent improvements across multiple datasets and base survival architectures.

# 7. Conclusion and Future Directions

This work studies time-to-event estimation under the independent censoring assumption. We argue that the commonly used censoring-mechanism-free likelihood is well motivated under non-informative censoring, but can discard useful signal when the event and censoring mechanisms are coupled through shared parameters. To address this, we propose `SALaD`, a survival-layer-agnostic framework that learns decomposed latent representations separating event-specific, censoring-specific, and confounding factors. Across 2 semi-synthetic and 8 real-world datasets, `SALaD` consistently improves over strong baselines, their `2B` extensions, and multiple SOTAs, while incurring a moderate computational overhead.

Several directions remain open. First, extending `SALaD` to *dependent censoring* settings is an important next step. Second, `SALaD` can be generalized to competing risks by learning representations that separate risk-specific and shared factors across causes, under conditional independence assumptions among causes and censoring.

## Acknowledgements

This research received support from the Natural Science and Engineering Research Council of Canada (NSERC) and the Alberta Machine Intelligence Institute (Amii). The authors extend their gratitude to the anonymous reviewers for their insightful feedback and valuable suggestions.

## Impact Statement

This paper presents work whose goal is to advance the field of Machine Learning. A primary positive impact is more accurate and better-calibrated time-to-event estimates, which can support downstream decision-making in high-stakes domains such as healthcare (*e.g.*, prognosis, follow-up scheduling, and resource planning), as well as engineering and social science applications where right-censoring and attrition are common. Potential negative impacts arise from misuse or over-trust in predictive outputs. Survival predictions can be incorrectly interpreted as causal effects or used to justify decisions without appropriate clinical or domain validation.

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

# A. Notation and Assumption

## A.1. Notation

We use capital letters (*e.g.*, $\boldsymbol{X}$, $E$, $C$) to denote random variables, and lowercase letters with subscripts (*e.g.*, $\boldsymbol{x}_i$, $e_i$, $c_i$) to denote realizations for instance $i$. Table 1 summarizes the key symbols used throughout the paper.

*Table 1.* Summary of symbols in the main text.

| Symbol | Definition |
|:---:|:---|
| *General Sets and Mathematical Notation* | |
| $\mathbb{E}[\cdot]$ | Expectation |
| $\widehat{(\cdot)}$ | Predicted value(s) |
| $\odot$ | Matrix multiplication |
| $\overline{(\cdot)}$ | Mean or median, depending on context |
| $\mathbb{R}$ | Set of real numbers |
| $(\cdot)^{\top}$ | Matrix Transpose |
| *Random Variables and Observations* | |
| $C$ | Censoring time (R.V.) |
| $c_i$ | Realization of $C$ for subject $i$ |
| $\mathcal{D}$ | Observational dataset |
| $d$ | Dimensionality of feature vector $\boldsymbol{X}$ |
| $\Delta$ | Event indicator (R.V.): $\Delta = \mathbb{1}[E \leq C]$ |
| $\delta_i$ | Realization of $\Delta$ for subject $i$ |
| $E$ | Event time (R.V.) |
| $e_i$ | Realization of $E$ for subject $i$ |
| $\boldsymbol{H}^{(\cdot)}$ | Latent R.V.s (*e.g.*, hidden factors) |
| $\boldsymbol{h}_i^{(\cdot)}$ | Realization of $\boldsymbol{H}^{(\cdot)}$ for subject $i$ |
| $N$ | Number of subjects in the dataset |
| $T$ | Observed time (R.V.): $T = \min(E, C)$ |
| $t_i$ | Realization of $T$ for subject $i$: $t_i = \min(e_i, c_i)$ |
| $\boldsymbol{X}$ | R.V.s of input features |
| $\boldsymbol{x}_i$ | Realization of $\boldsymbol{X}$ for subject $i$ |
| *Functions and Operators* | |
| $\mathbb{1}[\cdot]$ | Indicator function |
| $\mathtt{disc}(\cdot, \cdot)$ | Discrepancy function |
| $f(t \mid \boldsymbol{x}_i)$ | Conditional probability density function given $\boldsymbol{x}_i$ |
| $S(t)$ | Marginal survival function |
| $S(t \mid \boldsymbol{x}_i)$ | Conditional survival function given $\boldsymbol{x}_i$ |
| *Model Components and Parameters* | |
| $\beta_1, \beta_2$ | Balancing weights |
| $K$ | Number of hidden layers in the network |
| $\lambda$ | Weight decay coefficient |
| $M$ | Number of output units in the network |
| $\boldsymbol{\omega}$ | Model parameters |
| $\Phi^{(\cdot)}(\cdot)$ | Representation network |
| $\Psi(\cdot), \Psi(\cdot, \cdot)$ | Distribution network (univariate / bivariate) |
| *Superscript Notation* | |
| $(\cdot)^{\emptyset}$ | Independent of both $E$ and $C$ |
| $(\cdot)^{\epsilon}$ | Related to event time $E$ only |
| $(\cdot)^{\gamma}$ | Related to censoring time $C$ only |
| $(\cdot)^{\kappa}$ | Related to both $E$ and $C$ |

## A.2. Assumptions

### A.2.1. OVERLAP

We restate Assumption 2.1 (introduced in Section 2) and briefly discuss its implications.

**Assumption A.1** (Overlap). For any covariate value $x$ and any time $t$ in the support of $T \mid X = x$, both event and censoring occur with nonzero probability:

$$0 \;<\; \mathbb{P}(\Delta = 1 \mid T = t, X = x) \;<\; 1.$$

Equivalently, conditional on $(T, X)$, neither the event indicator nor the censoring indicator is deterministic.

This assumption is standard in survival analysis and mirrors the "positivity/overlap" condition used in causal inference: it rules out degenerate regimes where, for a certain group of instances or time regions, outcomes are *always* observed or *always* censored.

In many applications it is also substantively plausible. An obvious example is human mortality: "Valar morghulis" – all men must die; similarly, in mechanical systems, all components must break. In oncology, basic arguments suggest that over an infinite time horizon, the probability of developing cancer can approach one due to cumulative mutation opportunities (Weinberg, 2006). At the same time, real studies typically face numerous censoring mechanisms (*e.g.*, loss to follow-up, relocation, competing causes of death), making it unrealistic that a subject would never be censored in principle.

### A.2.2. INDEPENDENT CENSORING VS. NON-INFORMATIVE CENSORING

A key motivation for this paper is understanding the distinction between the *conditional independent censoring* assumption (Assumption 2.2) and the *non-informative censoring* assumption (Assumption 2.3). The literature sometimes conflates these two concepts (using different terms to refer to what is implicitly treated as the same condition), but they are logically separate. Helpful discussions include Chapter 1.XI, (pages 37-43) in Kleinbaum & Klein (2012), Chapter 2.7 (pages 20-22) in Emura & Chen (2018), and a related StackExchange thread (https://stats.stackexchange.com/questions/22497/problem-with-informative-censoring/22605).

**Key Differences.**

- **Conditional independent censoring** is a *stochastic independence* statement: $E \perp C \mid X$. Intuitively, after conditioning on observed features $X$, knowing an individual's censoring time does not provide additional information about that individual's event time (and vice versa).

- **Non-informative censoring** is a *parameter/separability* statement: the model for the censoring distribution $C \mid X$ does not share parameters with the model for the event distribution $E \mid X$. Intuitively, the censoring process does not carry information about parameters that govern the event-time mechanism.

Although these assumptions are often invoked together (and non-informative censoring is frequently justified by an independence argument), they are *orthogonal* in general: neither implies the other. In Section 2 we gave an exaggerated shared-parameter example illustrating that $E$ and $C$ can be independent yet censoring can still be informative in the sense of shared parameters in exponential distributions.

**A real-world illustration (independent yet informative).** Kleinbaum & Klein (2012, Ch. 1.XI, pp. 42) describes a study design in which whenever an event of interest occurs (*e.g.*, a death), another participant is randomly selected from the remaining risk set to leave the study (*e.g.*, a family member decides to leave the study after hearing the news). Conditional on relevant features (*e.g.*, stratifying by sex), such departures can be *independent* of the departing individual's own event time (because it actually depends on another person's event time), satisfying conditional independent censoring. However, censoring is still *informative* at the population level because the censoring mechanism is triggered by the occurrence of events among other participants, coupling the event and censoring processes through a shared mechanism. In the extreme case where this is the only censoring mechanism, event and censoring distribution can become exactly the same.

# B. Theoretical Analysis for 2B

This appendix analyzes the *asymptotic efficiency* of the proposed 2B framework and provides a detailed proof of Theorem 3.1 from Section 3.

We observe i.i.d. triples $y_i = (\boldsymbol{x}_i, t_i, \delta_i)$ for $i = 1, \ldots, N$. Let $Y = (\boldsymbol{X}, T, \Delta)$ denote the corresponding random vector, so that $y_i$ is a realization of $Y$.

*Model architecture and parameters:* Let $\boldsymbol{\omega} = (\Phi, \Psi_E, \Psi_C) \in \boldsymbol{\Omega} \subset \mathbb{R}^p$ denote the *variable* model parameter, where $\Phi$ parameterizes a shared representation network $\boldsymbol{H} = \Phi(\boldsymbol{X})$, and $\Psi_E, \Psi_C$ parameterize two distribution networks mapping $\boldsymbol{H}$ to the event and censoring distributions, respectively. Again, we abuse notation for both networks themselves and the parameters in the networks. Concretely, the model outputs two conditional survival functions for event and censoring times:

$$\widehat{S}_E(\cdot \mid \boldsymbol{X}; \boldsymbol{\omega}), \ \widehat{f}_E(\cdot \mid \boldsymbol{X}; \boldsymbol{\omega}) \quad \text{and} \quad \widehat{S}_C(\cdot \mid \boldsymbol{X}; \boldsymbol{\omega}), \ \widehat{f}_C(\cdot \mid \boldsymbol{X}; \boldsymbol{\omega}).$$

Under correct specification at $\boldsymbol{\omega}_0$, these outputs coincide with the true conditional survival/density functions.

The corresponding *event-relevant* subvector is

$$\boldsymbol{\omega}_E \ := \ (\Phi, \Psi_E) \ \in \ \boldsymbol{\Omega}_E \ \subset \ \mathbb{R}^{p_E}.$$

Let $\boldsymbol{\omega}_0 = (\Phi_0, \Psi_{E,0}, \Psi_{C,0})$ denote the (unknown) *true* parameter value under the posited model, and let $\boldsymbol{\omega}_{E,0} = (\Phi_0, \Psi_{E,0})$ denote its event-relevant subvector. We write $\mathbb{E}_{\boldsymbol{\omega}_0}[\cdot]$ for expectation under the true observed-data distribution $p(\cdot; \boldsymbol{\omega}_0)$.

*Full observed-data likelihood:* Under Assumption 2.2 ($E \perp C \mid \boldsymbol{X}$), the observed-data likelihood contribution of $y_i = (\boldsymbol{x}_i, t_i, \delta_i)$ equals

$$p(y_i; \boldsymbol{\omega}) \ = \ \left[\widehat{f}_E(t_i \mid \boldsymbol{x}_i; \boldsymbol{\omega}) \, \widehat{S}_C(t_i \mid \boldsymbol{x}_i; \boldsymbol{\omega})\right]^{\delta_i} \left[\widehat{S}_E(t_i \mid \boldsymbol{x}_i; \boldsymbol{\omega}) \, \widehat{f}_C(t_i \mid \boldsymbol{x}_i; \boldsymbol{\omega})\right]^{1-\delta_i}. \tag{8}$$

*Reduced (event-only) ERM objective:* The reduced criterion discards censoring terms and optimizes

$$q(y_i; \boldsymbol{\omega}_E) \ = \ \widehat{f}_E(t_i \mid \boldsymbol{x}_i; \boldsymbol{\omega}_E)^{\delta_i} \, \widehat{S}_E(t_i \mid \boldsymbol{x}_i; \boldsymbol{\omega}_E)^{1-\delta_i}, \tag{9}$$

Note that $q(\cdot; \boldsymbol{\omega}_E)$ is an ERM objective and need not equal the correct observed-data density when parameters are shared.

**Assumption B.1.** The full model $p(Y; \boldsymbol{\omega})$ in (8) is correctly specified at $\boldsymbol{\omega}_0$. Let

$$\boldsymbol{s}(Y; \boldsymbol{\omega}) := \nabla_{\boldsymbol{\omega}} \log p(Y; \boldsymbol{\omega})$$

be the gradient wrt the parameter vector $\boldsymbol{\omega}$ of the log-likelihood contribution from one observation, this is also called the *score vector*.

Assume conditions:

1. $\boldsymbol{\omega}_0$ is identifiable;

2. $\log p(Y; \boldsymbol{\omega})$ is twice continuously differentiable in a neighborhood of $\boldsymbol{\omega}_0$;

3. differentiation and integration may be interchanged;

4. the Fisher information matrix

$$\boldsymbol{I} \ := \ \mathbb{E}_{\boldsymbol{\omega}_0}\left[\boldsymbol{s}(Y; \boldsymbol{\omega}_0) \, \boldsymbol{s}(Y; \boldsymbol{\omega}_0)^\top\right],$$

   is finite and positive definite.

Intuitively, the Fisher information matrix $\boldsymbol{I}$ is essentially the covariance matrix of the score (because under correct specification, $\mathbb{E}_{\boldsymbol{\omega}_E}[\boldsymbol{s}(Y; \boldsymbol{\omega}_E)] = 0$, so $\mathbb{E}_{\boldsymbol{\omega}_E}[\boldsymbol{s}(Y; \boldsymbol{\omega}_E)\boldsymbol{s}(Y; \boldsymbol{\omega}_E)^\top] = \text{Var}(\boldsymbol{s}(Y; \boldsymbol{\omega}_E))$. Therefore, the Fisher matrix essentially means how much information one observation carries about $\boldsymbol{\omega}$.

**Assumption B.2.** Define the *reduced score* function

$$\boldsymbol{s}^*(Y; \boldsymbol{\omega}_E) := \nabla_{\boldsymbol{\omega}_E} \log q(Y; \boldsymbol{\omega}_E).$$

Assume conditions:

1. $\log q(Y; \boldsymbol{\omega}_E)$ is twice continuously differentiable in a neighborhood of $\boldsymbol{\omega}_{E,0}$;

2. interchange of differentiation and integration is justified;

3. The sensitivity matrix

$$\boldsymbol{A}_E := -\mathbb{E}_{\boldsymbol{\omega}_0}\Big[\nabla_{\boldsymbol{\omega}_E} \boldsymbol{s}^*(Y; \boldsymbol{\omega}_{E,0})\Big],$$

   and the variability matrix

$$\boldsymbol{J}_E := \mathbb{E}_{\boldsymbol{\omega}_0}\Big[\boldsymbol{s}^*(Y; \boldsymbol{\omega}_{E,0})\, \boldsymbol{s}^*(Y; \boldsymbol{\omega}_{E,0})^\top\Big],$$

   are both finite and positive definite.

Under these conditions, $\widehat{\boldsymbol{\omega}}_E^*$ is a regular $M$-estimator with sandwich covariance

$$\Sigma(\widehat{\boldsymbol{\omega}}_E) \;=\; \boldsymbol{A}_E^{-1}\, \boldsymbol{J}_E\, \boldsymbol{A}_E^{-\top}.$$

and the corresponding Godambe information is

$$\boldsymbol{G}_E \;=\; \boldsymbol{A}_E^\top\, \boldsymbol{J}_E^{-1}\, \boldsymbol{A}_E.$$

For a correctly specified full likelihood MLE, we additionally have the "information equality" $\boldsymbol{A}_E = \boldsymbol{J}_E = \boldsymbol{I}$, so the sandwich collapses to $\boldsymbol{I}^{-1}$.

**Lemma B.3.** *Under Assumptions 2.2 and B.2(i)-(ii), we can have*

$$\mathbb{E}_{\boldsymbol{\omega}_0}\Big[\boldsymbol{s}^*(Y; \boldsymbol{\omega}_{E,0})\Big] \;=\; 0.$$

*Proof.* Fix $\boldsymbol{X} = \boldsymbol{x}$ (when no confusion arises, we drop the subscript $i$ for simplicity). Based on (8), equivalently, separating the two cases $\delta \in \{0, 1\}$:

$$p(t, \delta = 1 \mid \boldsymbol{x}; \boldsymbol{\omega}_0) \;=\; \widehat{f}_E(t \mid \boldsymbol{x}; \boldsymbol{\omega}_0)\, \widehat{S}_C(t \mid \boldsymbol{x}; \boldsymbol{\omega}_0)$$
$$p(t, \delta = 0 \mid \boldsymbol{x}; \boldsymbol{\omega}_0) \;=\; \widehat{S}_E(t \mid \boldsymbol{x}; \boldsymbol{\omega}_0)\, \widehat{f}_C(t \mid \boldsymbol{x}; \boldsymbol{\omega}_0).$$

Next, spell out the reduced criterion according to (9):

$$\boldsymbol{s}^*(y; \boldsymbol{\omega}_E) \;=\; \boldsymbol{s}^*(t, \delta, \boldsymbol{x}; \boldsymbol{\omega}_E) \;=\; \delta\, \nabla_{\boldsymbol{\omega}_E} \log \widehat{f}_E(t \mid \boldsymbol{x}; \boldsymbol{\omega}_E) + (1 - \delta)\, \nabla_{\boldsymbol{\omega}_E} \log \widehat{S}_E(t \mid \boldsymbol{x}; \boldsymbol{\omega}_E).$$

Taking the conditional expectation under $\boldsymbol{\omega}_0$ and simplifying yields

$$\begin{aligned}
\mathbb{E}_{\boldsymbol{\omega}_0}\Big[\boldsymbol{s}^*(Y; \boldsymbol{\omega}_{E,0}) \,\Big|\, x\Big] \;&=\; \int_0^\infty \boldsymbol{s}^*(t, 1, \boldsymbol{x}; \boldsymbol{\omega}_{E,0})\, p(t, \delta = 1 \mid \boldsymbol{x}; \boldsymbol{\omega}_0)\, \mathrm{d}t + \int_0^\infty \boldsymbol{s}^*(t, 0, \boldsymbol{x}; \boldsymbol{\omega}_{E,0})\, p(t, \delta = 0 \mid \boldsymbol{x}; \boldsymbol{\omega}_0)\, \mathrm{d}t \\
&=\; \int_0^\infty \nabla_{\boldsymbol{\omega}_E} \log \widehat{f}_E(t \mid \boldsymbol{x}; \boldsymbol{\omega}_{E,0})\, \widehat{f}_E(t \mid \boldsymbol{x}; \boldsymbol{\omega}_0)\, \widehat{S}_C(t \mid \boldsymbol{x}; \boldsymbol{\omega}_0)\, \mathrm{d}t \\
&\quad + \int_0^\infty \nabla_{\boldsymbol{\omega}_E} \log \widehat{S}_E(t \mid \boldsymbol{x}; \boldsymbol{\omega}_{E,0})\, \widehat{S}_E(t \mid \boldsymbol{x}; \boldsymbol{\omega}_0)\, \widehat{f}_C(t \mid \boldsymbol{x}; \boldsymbol{\omega}_0)\, \mathrm{d}t \\
&=\; \int_0^\infty \nabla_{\boldsymbol{\omega}_E} \log \widehat{f}_E(t \mid \boldsymbol{x}; \boldsymbol{\omega}_{E,0})\, \widehat{f}_E(t \mid \boldsymbol{x}; \boldsymbol{\omega}_{E,0})\, \widehat{S}_C(t \mid \boldsymbol{x}; \boldsymbol{\omega}_0)\, \mathrm{d}t \\
&\quad + \int_0^\infty \nabla_{\boldsymbol{\omega}_E} \log \widehat{S}_E(t \mid \boldsymbol{x}; \boldsymbol{\omega}_{E,0})\, \widehat{S}_E(t \mid \boldsymbol{x}; \boldsymbol{\omega}_{E,0})\, \widehat{f}_C(t \mid \boldsymbol{x}; \boldsymbol{\omega}_0)\, \mathrm{d}t. \qquad (10)
\end{aligned}$$

In addition, in Assumption B.2(ii), we have $s^*(Y; \boldsymbol{\omega}_E)$ being differentiable. Recall that, for any positive differentiable function $g(\cdot; \boldsymbol{\omega}_E)$, we can have

$$\nabla_{\boldsymbol{\omega}_E} \log g(t; \boldsymbol{\omega}_E) = \frac{\nabla_{\boldsymbol{\omega}_E} g(t; \boldsymbol{\omega}_E)}{g(t; \boldsymbol{\omega}_E)} \quad \Longrightarrow \quad \left(\nabla_{\boldsymbol{\omega}_E} \log g(t; \boldsymbol{\omega}_E)\right) g(t; \boldsymbol{\omega}_E) = \nabla_{\boldsymbol{\omega}_E} g(t; \boldsymbol{\omega}_E).$$

Applying this identity to each term in (10) yields

$$
\begin{aligned}
\mathbb{E}_{\boldsymbol{\omega}_0}\Big[s^*(Y; \boldsymbol{\omega}_{E,0}) \mid \boldsymbol{x}\Big] &= \int_0^\infty \nabla_{\boldsymbol{\omega}_E} \widehat{f}_E(t \mid \boldsymbol{x}; \boldsymbol{\omega}_{E,0}) \, \widehat{S}_C(t \mid \boldsymbol{x}; \boldsymbol{\omega}_0) \, \mathrm{d}t + \int_0^\infty \nabla_{\boldsymbol{\omega}_E} \widehat{S}_E(t \mid \boldsymbol{x}; \boldsymbol{\omega}_{E,0}) \, \widehat{f}_C(t \mid \boldsymbol{x}; \boldsymbol{\omega}_0) \, \mathrm{d}t \\
&= \int_0^\infty \nabla_{\boldsymbol{\omega}_E} \widehat{f}_E(t \mid \boldsymbol{x}; \boldsymbol{\omega}_{E,0}) \left(\int_t^\infty \widehat{f}_C(u \mid \boldsymbol{x}; \boldsymbol{\omega}_0) \, du\right) \mathrm{d}t - \int_0^\infty \nabla_{\boldsymbol{\omega}_E} \widehat{F}_E(t \mid \boldsymbol{x}; \boldsymbol{\omega}_{E,0}) \, \widehat{f}_C(t \mid \boldsymbol{x}; \boldsymbol{\omega}_0) \, \mathrm{d}t \\
&= \int_0^\infty \widehat{f}_C(u \mid \boldsymbol{x}; \boldsymbol{\omega}_0) \left(\int_0^u \nabla_{\boldsymbol{\omega}_E} \widehat{f}_E(t \mid \boldsymbol{x}; \boldsymbol{\omega}_{E,0}) \, \mathrm{d}t\right) \mathrm{d}u - \int_0^\infty \nabla_{\boldsymbol{\omega}_E} \widehat{F}_E(t \mid \boldsymbol{x}; \boldsymbol{\omega}_{E,0}) \, \widehat{f}_C(t \mid \boldsymbol{x}; \boldsymbol{\omega}_0) \, \mathrm{d}t \\
&= \int_0^\infty \widehat{f}_C(u \mid \boldsymbol{x}; \boldsymbol{\omega}_0) \nabla_{\boldsymbol{\omega}_E} \widehat{F}_E(u \mid \boldsymbol{x}; \boldsymbol{\omega}_{E,0}) \, \mathrm{d}u - \int_0^\infty \nabla_{\boldsymbol{\omega}_E} \widehat{F}_E(t \mid \boldsymbol{x}; \boldsymbol{\omega}_{E,0}) \, \widehat{f}_C(t \mid \boldsymbol{x}; \boldsymbol{\omega}_0) \, \mathrm{d}t \\
&= 0
\end{aligned}
$$

Finally, taking the law of total expectation:

$$\mathbb{E}_{\boldsymbol{\omega}_0}\big[s^*(Y; \boldsymbol{\omega}_{E,0})\big] = \mathbb{E}_{\boldsymbol{\omega}_0}\Big[\mathbb{E}_{\boldsymbol{\omega}_0}\big[s^*(Y; \boldsymbol{\omega}_{E,0}) \mid \boldsymbol{x}\big]\Big] = 0,$$

which completes the proof. $\qquad\square$

**Lemma B.4.** *Let us partition $\boldsymbol{s}(Y; \boldsymbol{\omega})$ as*

$$\boldsymbol{s}(Y; \boldsymbol{\omega}) = \left(\boldsymbol{s}_E(Y; \boldsymbol{\omega})^\top, \, \boldsymbol{s}_C(Y; \boldsymbol{\omega})^\top\right)^\top,$$

*corresponding to $\boldsymbol{\omega} = (\boldsymbol{\omega}_E, \Psi_C)$. Under Assumptions B.1-B.2, we have*

$$
\begin{aligned}
\mathbb{E}_{\boldsymbol{\omega}_0}\Big[s^*(Y; \boldsymbol{\omega}_{E,0}) \, \boldsymbol{s}_E(Y; \boldsymbol{\omega}_0)^\top\Big] &= \boldsymbol{A}_E \\
\mathbb{E}_{\boldsymbol{\omega}_0}\Big[s^*(Y; \boldsymbol{\omega}_{E,0}) \, \boldsymbol{s}_C(Y; \boldsymbol{\omega}_0)^\top\Big] &= 0.
\end{aligned}
$$

*Proof.* We utilize the fact that the estimating equation is unbiased for any true parameter value (consistency). Define the vector-valued function:

$$u(\boldsymbol{\omega}) := \int s^*(y; \boldsymbol{\omega}_E) \, p(y; \boldsymbol{\omega}) \, \mathrm{d}y.$$

By Lemma B.3, $u(\boldsymbol{\omega}) = 0$ for all $\boldsymbol{\omega}$ in the parameter space (not just at $\boldsymbol{\omega}_0$), because the reduced score expectation is zero whenever the data generating distribution matches the parameter used in the score. Since $u(\boldsymbol{\omega})$ is identically zero, its gradient with respect to the full parameter vector $\boldsymbol{\omega}$ must be the zero matrix:

$$\nabla_{\boldsymbol{\omega}} u(\boldsymbol{\omega}) = 0.$$

Applying the product rule to differentiate under the integral sign:

$$
\begin{aligned}
0 &= \nabla_{\boldsymbol{\omega}} \int s^*(y; \boldsymbol{\omega}_E) \, p(y; \boldsymbol{\omega}) \, \mathrm{d}y \\
&= \int \left[\nabla_{\boldsymbol{\omega}} s^*(y; \boldsymbol{\omega}_E)\right] p(y; \boldsymbol{\omega}) \, \mathrm{d}y + \int s^*(y; \boldsymbol{\omega}_E) \left[\nabla_{\boldsymbol{\omega}} p(y; \boldsymbol{\omega})\right]^\top \mathrm{d}y.
\end{aligned}
$$

Now, we evaluate this expression at $\boldsymbol{\omega} = \boldsymbol{\omega}_0$. **Term 1:** Since $s^*$ depends only on $\boldsymbol{\omega}_E$, its gradient with respect to $\boldsymbol{\omega} = (\boldsymbol{\omega}_E, \Psi_C)$ is a block matrix:

$$\nabla_{\boldsymbol{\omega}} s^*(y; \boldsymbol{\omega}_E) = \left[\nabla_{\boldsymbol{\omega}_E} s^*(y; \boldsymbol{\omega}_E), \, 0\right].$$

Taking the expectation at $\boldsymbol{\omega}_0$, this term becomes:

$$\mathbb{E}_{\boldsymbol{\omega}_0}\Big[\nabla_{\boldsymbol{\omega}}\boldsymbol{s}^*(Y;\boldsymbol{\omega}_{E,0})\Big] \;=\; \big[-\boldsymbol{A}_E\,,\;\boldsymbol{0}\big].$$

**Term 2:** We use the log-derivative trick $\nabla p = p\nabla\log p = p\boldsymbol{s}$:

$$\int \boldsymbol{s}^*(y;\boldsymbol{\omega}_{E,0}),\boldsymbol{s}(y;\boldsymbol{\omega}_0)^\top,p(y;\boldsymbol{\omega}_0),\mathrm{d}y \;=\; \mathbb{E}_{\boldsymbol{\omega}_0}\Big[\boldsymbol{s}^*(Y;\boldsymbol{\omega}_{E,0}),\boldsymbol{s}(Y;\boldsymbol{\omega}_0)^\top\Big].$$

Partitioning the full score $\boldsymbol{s}$ into $(\boldsymbol{s}_E^\top,\boldsymbol{s}_C^\top)^\top$, this expectation term splits into:

$$\Big[\,\mathbb{E}_{\boldsymbol{\omega}_0}[\boldsymbol{s}^*\boldsymbol{s}_E^\top]\,,\;\mathbb{E}_{\boldsymbol{\omega}_0}[\boldsymbol{s}^*\boldsymbol{s}_C^\top]\,\Big].$$

Adding Term 1 and Term 2 and equating to zero:

$$\Big[\,-\boldsymbol{A}_E+\mathbb{E}_{\boldsymbol{\omega}_0}[\boldsymbol{s}^*\boldsymbol{s}_E^\top]\,,\;\boldsymbol{0}+\mathbb{E}_{\boldsymbol{\omega}_0}[\boldsymbol{s}^*\boldsymbol{s}_C^\top]\,\Big] \;=\; \big[\boldsymbol{0},\boldsymbol{0}\big].$$

This directly implies the two required identities:

$$\mathbb{E}_{\boldsymbol{\omega}_0}[\boldsymbol{s}^*\boldsymbol{s}_E^\top] \;=\; \boldsymbol{A}_E,$$
$$E_{\boldsymbol{\omega}_0}[\boldsymbol{s}^*\boldsymbol{s}_C^\top] \;=\; \boldsymbol{0}.$$

$\square$

We can now prove Theorem 3.1. For completeness, we restate it formally.

**Theorem B.5.** *Assuming 2.2, B.1, and B.2, let the full MLE be*

$$\widehat{\boldsymbol{\omega}} \;:=\; \arg\max_{\boldsymbol{\omega}\in\boldsymbol{\Omega}} \sum_{i=1}^{N}\log p(y_i;\boldsymbol{\omega}),$$

*and denote $\widehat{\boldsymbol{\omega}}_E$ the event-relevant subvector. Let the reduced ERM estimator be*

$$\widehat{\boldsymbol{\omega}}_E^* \;:=\; \arg\max_{\boldsymbol{\omega}_E\in\boldsymbol{\Omega}_E} \sum_{i=1}^{N}\log q(y_i;\boldsymbol{\omega}_E).$$

*Then*

$$\sqrt{N}(\widehat{\boldsymbol{\omega}}_E - \boldsymbol{\omega}_{E,0}) \;\xrightarrow{d}\; \mathcal{N}\big(0,\,[\boldsymbol{I}^{-1}]_{\boldsymbol{\omega}_E}\big)$$
$$\sqrt{N}(\widehat{\boldsymbol{\omega}}_E^* - \boldsymbol{\omega}_{E,0}) \;\xrightarrow{d}\; \mathcal{N}\big(0,\,\boldsymbol{G}_E^{-1}\big),$$

*and the full-likelihood estimator is asymptotically (weakly) more efficient:*

$$[\boldsymbol{I}^{-1}]_{\boldsymbol{\omega}_E} \;\preceq\; \boldsymbol{G}_E^{-1}. \tag{11}$$

*The inequality is strict unless the reduced score spans the efficient score for $\boldsymbol{\omega}_E$ almost surely.*

*Proof.* By Assumption B.1,

$$\sqrt{N}(\widehat{\boldsymbol{\omega}} - \boldsymbol{\omega}_0) \;\xrightarrow{d}\; \mathcal{N}(0,\boldsymbol{I}^{-1}),$$

hence the $\boldsymbol{\omega}_E$ subvector satisfies $\sqrt{N}(\widehat{\boldsymbol{\omega}}_E - \boldsymbol{\omega}_{E,0}) \xrightarrow{d} \mathcal{N}(0,[\boldsymbol{I}^{-1}]_{\boldsymbol{\omega}_E})$. By Lemma B.3 and Assumption B.2, $\widehat{\boldsymbol{\omega}}_E^*$ is a regular $M$-estimator with asymptotic covariance $\boldsymbol{G}_E^{-1} = \boldsymbol{A}_E^{-1}\,\boldsymbol{J}_E\,\boldsymbol{A}_E^{-\top}$.

Recall the definition in Assumption B.1:

$$\boldsymbol{I} \;:=\; \mathbb{E}_{\boldsymbol{\omega}_0}\Big[\boldsymbol{s}(Y;\boldsymbol{\omega}_0)\,\boldsymbol{s}(Y;\boldsymbol{\omega}_0)^\top\Big]$$

Now plug in the partition form:

$$s(Y;\boldsymbol{\omega}) = \left(s_E(Y;\boldsymbol{\omega})^\top,\ s_C(Y;\boldsymbol{\omega})^\top\right)^\top,$$

which yields:

$$I = \begin{pmatrix} I_{EE} & I_{EC} \\ I_{CE} & I_{CC} \end{pmatrix}.$$

Define the efficient score for $\boldsymbol{\omega}_E$ in the presence of nuisance $\Psi_C$ by

$$\tilde{s}_E := s_E - I_{EC}I_{CC}^{-1}s_C$$
$$\implies \qquad \mathbb{E}_{\boldsymbol{\omega}_0}[\tilde{s}_E\tilde{s}_E^\top] = I_{EE} - I_{EC}I_{CC}^{-1}I_{CE} =: I_{\text{eff}}.$$

A standard block-matrix inversion identity gives

$$[I^{-1}]_{\boldsymbol{\omega}_E} = I_{\text{eff}}^{-1}. \tag{12}$$

By Lemma B.4, $\mathbb{E}_{\boldsymbol{\omega}_0}\!\left[s^* \, s_C^\top\right] = 0$ and thus

$$\mathbb{E}_{\boldsymbol{\omega}_0}\!\left[s^* \, \tilde{s}_E^\top\right] = \mathbb{E}_{\boldsymbol{\omega}_0}\!\left[s^* \, s_E^\top\right] = A_E.$$

Therefore the covariance matrix of the stacked vector $\left(\tilde{s}_E^\top,\ s^{*\top}\right)^\top$ is positive semi-definite:

$$\mathbb{E}_{\boldsymbol{\omega}_0}\!\left[\begin{pmatrix} \tilde{s}_E \\ s^* \end{pmatrix}\begin{pmatrix} \tilde{s}_E \\ s^* \end{pmatrix}^\top\right] = \begin{pmatrix} I_{\text{eff}} & A_E^\top \\ A_E & J_E \end{pmatrix} \succeq 0.$$

Since $J_E \succ 0$, taking the Schur complement of $J_E$ yields

$$A_E^\top J_E^{-1} A_E \preceq I_{\text{eff}} \qquad \implies \qquad G_E \preceq I_{\text{eff}}.$$

Because both $G_E$ and $I_{\text{eff}}$ are positive definite, inversion reverses Loewner order:

$$I_{\text{eff}}^{-1} \preceq G_E^{-1}.$$

Combining this with (12) yields (11):

$$\left[I^{-1}\right]_{\boldsymbol{\omega}_E} = I_{\text{eff}}^{-1} \preceq G_E^{-1}.$$

Strictness follows from the strict Schur-complement condition. $\qquad\square$

## C. Details about the Proposed `SALaD` Algorithm

Here we present more details for the algorithm: Survival Analysis via Latent Decomposed Representation (`SALaD`). We first provide the pseudo-code for the overall algorithm in Algorithm 1.

The pseudo-code illustrates the algorithm using the probability density/mass function and the survival function as outputs (lines 8–11), with the (log-)likelihood loss serving as the objective function (line 12). This formulation applies to `N-MTLR` (Fotso, 2018), `AFTNN-Weibull` (Norman et al., 2024), and `AFTNN-LogLogistic` (Norman et al., 2024), as these models directly generate time-to-event and time-to-censoring distributions.

For `DeepSurv` (Katzman et al., 2018), the outputs correspond to the log of the (relative) risk scores, denoted as $g(\boldsymbol{x}_i)$, and the (relative) risk score itself, computed as $r(\boldsymbol{x}_i) = \exp(g(\boldsymbol{x}_i))$.

---

**Algorithm 1** Survival Analysis via Latent Decomposed Representation (SALaD)

---

**Require:** A survival dataset $\mathcal{D} = \{\boldsymbol{x}_i, t_i, \delta_i\}_{i=1}^N$; a survival model parametrized by $\mathcal{M} = \{\Phi^\epsilon, \Phi^\kappa, \Phi^\gamma, \Psi_E^{(1)}, \Psi_E^{(2)}, \Psi_C^{(1)}, \Psi_C^{(2)}\}$; balancing weights $\beta_1$ (for orthogonal term) and $\beta_2$ (for distance term); number of maximum training epoch $B$; learning rate $\alpha$; weight decay coefficient $\lambda$; distance function $\mathtt{disc}(\cdot, \cdot)$.

**Ensure:** The model with learned parameters.
1: Randomly partition $\mathcal{D}$ into a training set $\mathcal{D}^{\text{train}}$ and a validation set $\mathcal{D}^{\text{val}}$
2: $\mathcal{M} \leftarrow$ Instantiate $\left( \left\{ \Phi^\epsilon, \Phi^\kappa, \Phi^\gamma, \Psi_E^{(1)}, \Psi_E^{(2)}, \Psi_C^{(1)}, \Psi_C^{(2)} \right\} \right)$
3: $\widehat{S}_E(t) \leftarrow \{t_i, \delta_i\}_{i \in \mathcal{D}^{\text{train}}}$             ▷ Calculate the marginal event distribution using KM
4: $\widehat{S}_C(t) \leftarrow \{t_i, 1 - \delta_i\}_{i \in \mathcal{D}^{\text{train}}}$           ▷ Calculate the marginal censor distribution using KM
5: **repeat**
6:   **for** $\mathcal{D}^{\text{batch}} = \mathcal{D}^{\text{Batch1}}, \mathcal{D}^{\text{Batch2}}, \ldots$ **do**
7:    $\{\boldsymbol{h}_i^\epsilon, \boldsymbol{h}_i^\kappa, \boldsymbol{h}_i^\gamma\}_i \leftarrow \{\Phi^\epsilon(\boldsymbol{x}_i), \Phi^\kappa(\boldsymbol{x}_i), \Phi^\gamma(\boldsymbol{x}_i)\}_i$           ▷ $i \in \mathcal{D}^{\text{batch}}$
8:    $\left\{ \widehat{f}_E^{(1)}(t \mid \boldsymbol{x}_i), \widehat{S}_E^{(1)}(t \mid \boldsymbol{x}_i) \right\}_i \leftarrow \left\{ \Psi_E^{(1)}(\boldsymbol{h}_i^\epsilon) \right\}_i$         ▷ Event outcome 1
9:    $\left\{ \widehat{f}_E^{(2)}(t \mid \boldsymbol{x}_i), \widehat{S}_E^{(2)}(t \mid \boldsymbol{x}_i) \right\}_i \leftarrow \left\{ \Psi_E^{(2)}(\boldsymbol{h}_i^\epsilon, \boldsymbol{h}_i^\kappa) \right\}_i$       ▷ Event outcome 2
10:    $\left\{ \widehat{f}_C^{(1)}(t \mid \boldsymbol{x}_i), \widehat{S}_C^{(1)}(t \mid \boldsymbol{x}_i) \right\}_i \leftarrow \left\{ \Psi_C^{(1)}(\boldsymbol{h}_i^\gamma) \right\}_i$        ▷ Censor outcome 1
11:    $\left\{ \widehat{f}_C^{(2)}(t \mid \boldsymbol{x}_i), \widehat{S}_C^{(2)}(t \mid \boldsymbol{x}_i) \right\}_i \leftarrow \left\{ \Psi_C^{(2)}(\boldsymbol{h}_i^\gamma, \boldsymbol{h}_i^\kappa) \right\}_i$      ▷ Censor outcome 2
12:    $\mathcal{L}^{(2)}, \mathcal{L}^{(1)} \leftarrow$ Eq. (3) and (4)               ▷ Likelihood loss
13:    $\{W_E(t_i, \delta_i)\}_i \leftarrow \widehat{S}_E(t)$ and Eq. (5)
14:    $\{W_C(t_i, \delta_i)\}_i \leftarrow \widehat{S}_C(t)$ and Eq. (13)
15:    $\mathcal{L}^\gamma \leftarrow \{\boldsymbol{h}_i^\gamma\}_i, \{W_E(t_i, \delta_i)\}_i$, and Eq. (6)         ▷ Discrepancy for event
16:    $\mathcal{L}^\epsilon \leftarrow \{\boldsymbol{h}_i^\epsilon\}_i, \{W_C(t_i, \delta_i)\}_i$, and Eq. (7)         ▷ Discrepancy for censoring
17:    $\mathcal{L}_{\text{orth}} \leftarrow$ Eq. (14)                 ▷ Orthogonal loss
18:    $\mathcal{L}_{\text{total}} \leftarrow -\log(\mathcal{L}^{(2)} \cdot \mathcal{L}^{(1)}) + \beta_1 \mathcal{L}_{\text{orth}} + \beta_2(\mathcal{L}^\gamma + \mathcal{L}^\epsilon) + \lambda \mathcal{L}_{\text{reg}}$
19:    $\left\{ \Phi^\epsilon, \Phi^\kappa, \Phi^\gamma, \Psi_E^{(1)}, \Psi_E^{(2)}, \Psi_C^{(1)}, \Psi_C^{(2)} \right\} \leftarrow$ AdamW $\left( \left\{ \Phi^\epsilon, \Phi^\kappa, \Phi^\gamma, \Psi_E^{(1)}, \Psi_E^{(2)}, \Psi_C^{(1)}, \Psi_C^{(2)} \right\}, \mathcal{L}_{\text{total}}, \alpha \right)$
20:   **end for**
21:   Repeat lines 7-18 on $\mathcal{D}^{\text{val}}$ and get total loss $\mathcal{L}_{\text{val}}$
22: **until** convergence on $\mathcal{L}_{\text{val}}$ or reach the maximum epoch $B$

---

In SALaD (DeepSurv), the model estimates risk scores for time-to-event, $r_E^{(1)}(\boldsymbol{x}_i)$ and $r_E^{(2)}(\boldsymbol{x}_i)$, as well as for time-to-censoring, $r_C^{(1)}(\boldsymbol{x}_i)$ and $r_C^{(2)}(\boldsymbol{x}_i)$. Consequently, for SALaD (DeepSurv), the objective function follows the partial likelihood formulation (Cox, 1975), which consists of:

$$\mathcal{PL}^{(2)}\left( \mathcal{D}; \Phi^\epsilon, \Phi^\kappa, \Phi^\gamma, \Psi_E^{(2)}, \Psi_C^{(2)} \right) = \prod_{i=1}^N \frac{\delta_i \cdot r_E^{(2)}(\boldsymbol{x}_i)}{\sum_{j=1}^N \mathbb{1}[t_j > t_i] \cdot r_E^{(2)}(\boldsymbol{x}_i)} \times \prod_{i=1}^N \frac{(1 - \delta_i) \cdot r_C^{(2)}(\boldsymbol{x}_i)}{\sum_{j=1}^N \mathbb{1}[t_j > t_i] \cdot r_C^{(2)}(\boldsymbol{x}_i)},$$

and

$$\mathcal{PL}^{(1)}\left( \mathcal{D}; \Phi^\epsilon, \Phi^\gamma, \Psi_E^{(1)}, \Psi_C^{(1)} \right) = \prod_{i=1}^N \frac{\delta_i \cdot r_E^{(1)}(\boldsymbol{x}_i)}{\sum_{j=1}^N \mathbb{1}[t_j > t_i] \cdot r_E^{(1)}(\boldsymbol{x}_i)} \times \prod_{i=1}^N \frac{(1 - \delta_i) \cdot r_C^{(1)}(\boldsymbol{x}_i)}{\sum_{j=1}^N \mathbb{1}[t_j > t_i] \cdot r_C^{(1)}(\boldsymbol{x}_i)},$$

where

$$r_E^{(1)}(\boldsymbol{x}_i) = \exp\left( \Psi_E^{(1)}\left( \Phi^\epsilon(\boldsymbol{x}_i) \right) \right),$$
$$r_E^{(2)}(\boldsymbol{x}_i) = \exp\left( \Psi_E^{(2)}\left( \Phi^\epsilon(\boldsymbol{x}_i), \Phi^\kappa(\boldsymbol{x}_i) \right) \right),$$
$$r_C^{(1)}(\boldsymbol{x}_i) = \exp\left( \Psi_C^{(1)}\left( \Phi^\gamma(\boldsymbol{x}_i) \right) \right),$$
$$r_C^{(2)}(\boldsymbol{x}_i) = \exp\left( \Psi_C^{(2)}\left( \Phi^\gamma(\boldsymbol{x}_i), \Phi^\kappa(\boldsymbol{x}_i) \right) \right).$$

We can then replace $\mathcal{L}^{(2)}$ and $\mathcal{L}^{(1)}$ in line 18 in Algorithm 1 with $\mathcal{PL}^{(2)}$ and $\mathcal{PL}^{(1)}$, respectively.

## C.1. Integral Probability Metric

In probability theory, integral probability metrics (IPMs) define a class of distance functions between probability distributions, measuring how well a given class of functions can distinguish between two distributions. For two probability density functions $P$ and $Q$, IPMs are always symmetric and satisfy the triangle inequality (in contrast to the Kullback-Leibler divergence). Formally:

$$\text{IPM}_{\mathcal{G}}(P, Q) = \sup_{g \in \mathcal{G}} |\mathbb{E}_{X \sim P}[g(X)] - \mathbb{E}_{X \sim Q}[g(X)]|,$$

where $\mathcal{G}$ is a class of functions that determines the specific form of the metric.

In this paper, we use the weighted kernel Maximum Mean Discrepancy (MMD) (using linear kernel and RBF kernel) to compute the IPM. Maximum Mean Discrepancy (MMD) is a statistical measure used to compare two probability distributions $P$ and $Q$ based on samples drawn from them (Gretton et al., 2012). Given two distributions $P$ and $Q$ defined on a domain $\mathcal{X}$, MMD is defined as the norm of the difference between their embeddings in a reproducing kernel Hilbert space (RKHS) $\mathcal{H}$:

$$\text{MMD}^2(P, Q) = \| \mathbb{E}_{X \sim P}[\phi(X)] - \mathbb{E}_{Y \sim Q}[\phi(Y)] \|_{\mathcal{H}}^2,$$

where $\phi : \mathcal{X} \to \mathcal{H}$ is the feature mapping associated with the kernel function.

### C.1.1. WEIGHTED LINEAR MMD

In the weighted linear version of MMD, weights are assigned to samples to control their contribution. Given weight vectors $W^P = \{W_i^P\}_{i=1}^n$ and $W^Q = \{W_j^Q\}_{j=1}^m$ for samples from $P$ and $Q$, respectively, the weighted linear MMD is given by:

$$\widehat{\text{MMD}}_{\text{linear}}^2(P, Q) = \left\| \sum_{i=1}^n W_i^P \cdot x_i - \sum_{j=1}^m W_j^Q \cdot y_j \right\|_2^2.$$

### C.1.2. WEIGHTED MMD WITH RBF KERNEL

The radial basis function (RBF) kernel is commonly used in MMD calculations. It is defined as:

$$k(x, y) = \exp\left(-\frac{\|x - y\|^2}{2\sigma^2}\right).$$

The weighted version of MMD with an RBF kernel incorporates sample weights and is computed as:

$$\widehat{\text{MMD}}_{\text{RBF}}^2(P, Q) = \sum_{i=1}^n \sum_{j=1}^n W_i^P \cdot W_j^P \cdot k(x_i, x_j) - 2 \sum_{i=1}^n \sum_{j=1}^m W_i^P \cdot W_j^Q \cdot k(x_i, y_j) + \sum_{i=1}^m \sum_{j=1}^m W_i^Q \cdot W_j^Q \cdot k(y_i, y_j).$$

We use MMD to calculate the regularizers in (6) and (7), where $x_i$ and $y_i$ are the latent representations for two groups and weights are the probability of each representation falling into that group.

### C.1.3. OTHER IPM SCORES

The Wasserstein distance is another useful metric for computing Integral Probability Metrics (IPMs). Similar to MMD, it compares two distributions, $P$ and $Q$, by taking a supremum over a specific class of functions. However, unlike MMD, which uses the reproducing kernel Hilbert space (RKHS), Wasserstein distance relies on the 1-Lipschitz function class. This allows it to encode the cost of transporting mass between distributions, capturing the underlying geometry of the space.

Despite its advantages, computing the exact Wasserstein distance requires solving a linear program, making it more computationally expensive than MMD, especially in high-dimensional settings. Therefore, in this study, we focus on weighted linear MMD and RBF-MMD.

## C.2. Weights for Censoring

In Section 4.2.2, we presented our proposed solution for Propositions 4.2 and 4.3, and provided a complete derivation of (6) to enforce the condition $\boldsymbol{H}^\gamma \perp E$. The computation for (7) follows a similar approach – by conceptually treating censoring events as outcomes of interest and vice versa.

For completeness, we outline the key steps involved in this computation below.

Proposition 4.3 requires that $\boldsymbol{H}^\epsilon \perp C$, meaning that $\boldsymbol{H}^\epsilon$ should not convey any information about the censoring time. Analogous to the approach in Section 4.2.2, we split the dataset's censoring times into two groups (early vs. late), and define the probability that the $i$-th instance belongs to the early censoring group (i.e., censored before the median censoring time $\bar{c}$; see Figure 3, bottom) using the oracle survival function for censoring times $S_C(t)$:

$$W_C(t_i, \delta_i) \;=\; \mathbb{P}\left(C < \bar{c} \mid T = t_i, \Delta = \delta_i\right) \;=\; \mathbb{1}[S_C(t_i) > 0.5]\left(\delta_i + (1 - \delta_i)\frac{S_C(t_i) - 0.5}{S_C(t_i)}\right). \tag{13}$$

This expression reflects our use of $\bar{c}$ – the median censoring time under $S_C(t)$ – as a threshold to categorize each instance into one of four cases, analogous to those defined in Section 4.2.2:

1. Censored before $\bar{c}$;

2. Censored after $\bar{c}$;

3. Uncensored after $\bar{c}$;

4. Uncensored before $\bar{c}$.

For the first three cases, group membership is deterministic. For the fourth case, we estimate the group probability using Bayes' rule.

Visually (see Figure 3, bottom), for uncensored instances $e_2$ and $e_3$ whose event times are before $\bar{c}$, we interpret the probability of their belonging to the early group as the ratio of the blue arrows' length to the total height of the blue circles.

### C.3. Estimating Marginal Survival Distributions for Event and Censoring Times

To approximate the oracle marginal survival distributions for both event and censoring times, we can use a counting process approach, for example, the Kaplan-Meier (KM) estimator (Kaplan & Meier, 1958).

To estimate the marginal distribution for event times, let $N_E(t_s)$ denote the counting process that records the number of observed events up to time $t_s$,

$$N_E(t_s) \;=\; \left|\{\delta_i = 1, t_i \le t_s\}_{i=1}^N\right|,$$

where $|\cdot|$ means the size of the set. Then we can use $\mathrm{d}N_E(t)$ to represent the observed number of events in an infinitesimal time interval, i.e., the number of events that exactly happened at $t_s$.

$$\mathrm{d}N_E(t_s) \;=\; \left|\{\delta_i = 1, t_i = t_s\}_{i=1}^N\right|,$$

and let $Y(t)$ represent the at-risk process, indicating the number of individuals still under observation just before $t_s$:

$$Y(t_s) \;=\; \left|\{t_i \ge t_s\}_{i=1}^N\right|.$$

Then we can calculate the estimated survival function for event time using:

$$\widehat{S}_E^{\mathrm{KM}}(t) \;=\; \prod_{s:\, t_s \le t}\left(1 - \frac{\mathrm{d}N_E(t_s)}{Y(t_s)}\right).$$

To estimate the survival function for censoring times, we can treat the censoring as the "target event" and the original event as the "censoring". Therefore, we define $N_C(t_s)$ and $\mathrm{d}N_C(t_s)$ as the counting process that records the number of censoring incidents up to time $t_s$:

$$N_C(t_s) \;=\; \left|\{\delta_i = 0, t_i \le t_s\}_{i=1}^N\right|,$$
$$\mathrm{d}N_C(t_s) \;=\; \left|\{\delta_i = 0, t_i = t_s\}_{i=1}^N\right|,$$

and calculate the marginal survival distribution for censoring times as:

$$\widehat{S}_C^{\mathrm{KM}}(t) \;=\; \prod_{s:\, t_s \le t}\left(1 - \frac{\mathrm{d}N_C(t_s)}{Y(t_s)}\right).$$

## C.4. Bias for Kaplan Meier Estimator

We employ the KM estimator (Kaplan & Meier, 1958) to approximate the marginal survival distributions for both event times and censoring. However, KM can introduce bias when censoring time is not independent of event time (Campigotto & Weller, 2014).

We analyze this bias in detail. Consider the scenario where the KM estimator $\widehat{S}_E(t)$ underestimates the oracle marginal time-to-event survival function $S_E(t)$ – illustrated in Figure 7. As previously stated, our method relies on the estimated marginal distribution to partition instances into early and late groups.

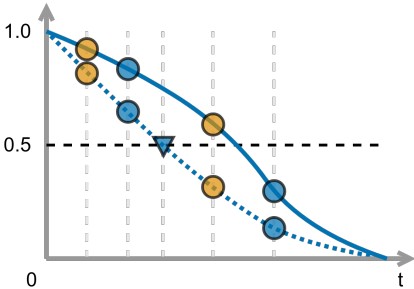

*Figure 7.* An illustration of the case when the Kaplan-Meier estimator is biased (underestimates the oracle marginal distribution), along with 4 example instances. The **blue and orange circles** share the same definitions as Figure 3. The **blue solid curve** indicates the oracle marginal time-to-event distribution. The **blue dotted curve** indicates the estimated time-to-event distribution using Kaplan-Meier. The **black dashed horizontal** indicates the 50% probability. The **gray dashed verticals** align points corresponding to the same subjects. The **blue triangle** (at the intersection of the estimated curve and 50% horizontal) indicates the median time ($\overline{t}$) calculated by the estimated distribution.

In cases where KM underestimates the oracle distribution, we determine the median survival time based on KM (denoted by the blue triangle) and use it as a threshold to split the data into two groups, each ostensibly containing 50% of the instances. However, when this median time is projected onto the oracle survival function (solid blue curve), it corresponds to a survival probability of approximately 75%. This discrepancy results in an imbalanced split: 75% of instances fall into the early group and 25% into the late group. However, as we anticipate, the size of the groups does not affect the validity of the proposed regularizers in (6) and (7).

Despite this imbalance, the validity of our algorithm (regularizer) mostly remains unaffected. This is because the relative ordering of instances is naturally preserved, except in cases where censored instances (resp., event instances) occur earlier than the median event time (resp., median censoring time). This corresponds to Case #4 depicted in Section 4.2.2 (resp., Case #4 in Appendix C.2).

## C.5. Details about the Orthogonal Regularizer

This appendix describes the *orthogonal regularizer for variable decomposition* we used in Section 4.2.3 to solve Proposition 4.4.

Before computing the loss, we need to quantify each variable's contribution for every representation network. Take one of the representation networks $\Phi^\epsilon(\cdot)$, with $K$ hidden layers and $M$-dimensional output, as an example. For each hidden layer $k \in \{1, \ldots, K\}$, let $\boldsymbol{\omega}_k^\epsilon$ be the trainable parameter matrix. We then calculate the contribution vector

$$\overline{\boldsymbol{\omega}^\epsilon} = \frac{1}{N} \mid \boldsymbol{\omega}_1^\epsilon \, \boldsymbol{\omega}_2^\epsilon \, \cdots \, \boldsymbol{\omega}_K^\epsilon \mid \cdot \mathbf{1} \quad \in \mathbb{R}^d,$$

where the absolute value is applied element-wise, and $\mathbf{1}$ denotes an all-ones $M \times 1$ vector. Each component in $\overline{\boldsymbol{\omega}^\epsilon}$ now indicates the average contribution of a particular feature in $\boldsymbol{X}$ to the representation $\boldsymbol{H}^\epsilon$. Similarly, we compute $\overline{\boldsymbol{\omega}^\gamma}$ and $\overline{\boldsymbol{\omega}^\kappa}$ for the other two representation network.

Finally, we try to minimize the orthogonal loss:

$$\mathcal{L}_{\text{orth}}(\Phi^\epsilon, \Phi^\gamma, \Phi^\kappa) = \overline{\boldsymbol{\omega}^\epsilon}^\top \, \overline{\boldsymbol{\omega}^\gamma} + \overline{\boldsymbol{\omega}^\gamma}^\top \, \overline{\boldsymbol{\omega}^\kappa} + \overline{\boldsymbol{\omega}^\kappa}^\top \, \overline{\boldsymbol{\omega}^\epsilon}, \tag{14}$$

to minimize the pairwise perpendicular scores between all 3 pairs of these average contribution vectors. By penalizing any overlap in the contributions, this constraint drives the model toward a "hard" decomposition where each variable's influence is isolated to a single representation. Notably, this approach does not require the three representation networks to share the same architecture or dimensionality – an advantage over traditional orthogonal losses that operate directly on the latent representations and thus require only matching representation lengths.

This regularizer should not be interpreted as a literal partition of the raw covariates. Rather, it is a soft constraint on how the encoders use input features, designed to promote non-overlap among the learned latent representations.

## D. Experimental Details

### D.1. Datasets

In this study, we evaluate the effectiveness of `2B` and `SALaD` across 2 semi-synthetic datasets and 8 real-world datasets. Specifically, the 8 real-world datasets we used are **HFCR**, **PBC**, **GBM**, **GBSG**, **METABRIC**, **NACD**, **SUPPORT**, and **MIMIC-IV**. Table 2 summarizes the data statistics. The detailed data description and preprocessing details can be found in Qi et al. (2024a) and Qi et al. (2024b).

*Table 2.* Key statistics of the datasets. Numbers in parentheses indicate the number of features after one-hot encoding.

| Dataset | #Sample | %Censor | Max $t$ | #Feature |
|---|---|---|---|---|
| **HFCR** | 299 | 67.89% | 285 | 11 |
| **PBC** | 418 | 61.48% | 4,795 | 17 |
| **GBM** | 595 | 17.23% | 3,881 | 8 (10) |
| **GBSG** | 686 | 56.41% | 2,659 | 8 |
| **METABRIC** | 1,981 | 55.17% | 9218 | 79 |
| **NACD** | 2,396 | 36.44% | 84.30 | 48 |
| **SUPPORT** | 9,105 | 31.89% | 2,029 | 26 (31) |
| **MIMIC-IV** | 38,520 | 66.65% | 4404 | 93 |

#### D.1.1. JUSTIFICATION FOR INDEPENDENT CENSORING

Because survival outcomes are only partially observed, there is generally no reliable, assumption-free test of the independent censoring assumption on real-world datasets (at least not without introducing additional, and often unverifiable, assumptions). Therefore, rather than attempting a formal test, we provide dataset-specific reasoning for why independent censoring may be plausible – that is, why any dependence between the event time and the censoring time is likely to be explainable by measured covariates.

**HFCR**   Unfortunately, we could not find a detailed description of the censoring mechanism. Chicco & Jurman (2020, Section "Dataset") also note that "The original dataset article unfortunately does not indicate if any patient had primary kidney disease, and provides no additional information about what type of follow-up was carried out." As a result, we cannot meaningfully justify independent censoring beyond acknowledging the lack of documentation.

**PBC**   The **PBC** dataset (Therneau, 2024) comprises patients diagnosed with Primary Biliary Cholangitis. The event of interest is death. Censoring may occur due to administrative end of follow-up or liver transplantation. Importantly, the likelihood and timing of transplantation are plausibly related to clinical status and laboratory measurements (*i.e.*, observed covariates). Hence, treating censoring as conditionally independent given covariates is reasonable in this setting.

**GBSG**   For the **GBSG** dataset (Schumacher et al., 1994), the endpoint is recurrence and/or death. The dominant censoring mechanism appears to be administrative right-censoring at the study analysis cutoff ("March 31, 1992"). However, Schumacher et al. (1994) report 63 patients whose last available information dates to before July 1990. The cause of censoring for these cases is not stated and is unlikely to be purely administrative. Therefore, we cannot confidently justify the independent censoring assumption for this dataset.

**GBM**    Unfortunately, we could not find a formal description of the dataset or its censoring mechanism, and thus cannot justify independent censoring.

**NACD**    Unfortunately, we could not find a formal description of the dataset or its censoring mechanism, and thus cannot justify independent censoring.

**METABRIC**    In **METABRIC**, the endpoint is disease-specific survival. Right-censoring primarily occurs when a patient is not observed to have died from the disease by the end of available follow-up. Concretely, follow-up time is measured from diagnosis to disease-specific death (if observed) or to the last recorded follow-up date otherwise.

**SUPPORT**    In the Study to Understand Prognoses Preferences Outcomes and Risks of Treatment (**SUPPORT**) dataset (Knaus et al., 1995), the event of interest is all-cause mortality, and follow-up time is measured from study entry until death or the end of follow-up. Right-censoring arises mainly from: (i) administrative censoring at the planned 6-month horizon (patients were prospectively followed for six months after enrollment), and (ii) incomplete post-discharge ascertainment, since many patients were discharged before the end of follow-up and were not continuously tracked thereafter (see https://archive.ics.uci.edu/dataset/880/support2). For (ii), the probability of being censored after discharge can plausibly depend on measured clinical severity, vital signs, and socio-demographic factors. Thus, conditional independent censoring given covariates is reasonable.

**MIMIC-IV**    We use the MIMIC-IV all-cause mortality datasets constructed by Qi et al. (2023a). The event is in-hospital mortality after admission, and the censoring time is defined as ICU discharge time. Covariates are laboratory measurements within the first 24 hours after admission. Since ICU discharge decisions and timing are strongly related to patient condition reflected in these measurements, censoring is plausibly dependent on the covariates; therefore, assuming independent censoring conditional on covariates is reasonable in this setting.

### D.1.2. DATA GENERATION PROCESS

We construct two *semi-synthetic dataset* in which the features are real, while event times and censoring times are generated with simulation approaches.

We generate synthetic right-censored time-to-event data by constructing three latent one-dimensional representations, and then defining event and censoring times through separate models that depend on different subsets of these representations. To make diversity about the datasets, we used a traditional methods where the mechanism are heuristic and the other mechanisms reflect those found in real-world data.

**semi-SUPPORT**    The data generation process (DGP) begins with a real dataset **SUPPORT**. We first obtain the feature $\boldsymbol{X}$ from this dataset. We sample three random projection vectors independently, via $\boldsymbol{\omega}^\epsilon, \boldsymbol{\omega}^\kappa, \boldsymbol{\omega}^\gamma \sim \mathcal{N}(\boldsymbol{0}, \boldsymbol{I}_d)$, and compute three candidate one-dimensional summaries of each individual (omit subscript $i$ for simplicity):

$$\begin{aligned} \boldsymbol{u}^\epsilon &= \boldsymbol{x}\,\boldsymbol{\omega}^\epsilon, \\ \boldsymbol{u}^\kappa &= \tanh(\boldsymbol{x}\,\boldsymbol{\omega}^\kappa), \\ \boldsymbol{u}^\gamma &= (\boldsymbol{x}\,\boldsymbol{\omega}^\gamma)^2. \end{aligned}$$

To remove large intercept components and obtain approximately orthogonal latent representations, we first center each $\boldsymbol{u}$:

$$\tilde{\boldsymbol{u}} = \boldsymbol{u} - \bar{\boldsymbol{u}}, \qquad \bar{\boldsymbol{u}} = \frac{1}{N}\sum_{i=1}^{N}\boldsymbol{u}_i,$$

and then apply Gram–Schmidt under the Euclidean inner product $\langle a, b\rangle = a^\top b$:

$$\boldsymbol{h}^\epsilon = \tilde{\boldsymbol{u}}^\epsilon, \tag{15}$$

$$\boldsymbol{h}^\kappa = \tilde{\boldsymbol{u}}^\kappa - \frac{\langle \tilde{\boldsymbol{u}}^\kappa, \boldsymbol{h}^\epsilon\rangle}{\langle \boldsymbol{h}^\epsilon, \boldsymbol{h}^\epsilon\rangle}\boldsymbol{h}^\epsilon, \tag{16}$$

$$\boldsymbol{h}^\gamma = \tilde{\boldsymbol{u}}^\gamma - \frac{\langle \tilde{\boldsymbol{u}}^\gamma, \boldsymbol{h}^\epsilon\rangle}{\langle \boldsymbol{h}^\epsilon, \boldsymbol{h}^\epsilon\rangle}\boldsymbol{h}^\epsilon - \frac{\langle \tilde{\boldsymbol{u}}^\gamma, \boldsymbol{h}^\kappa\rangle}{\langle \boldsymbol{h}^\kappa, \boldsymbol{h}^\kappa\rangle}\boldsymbol{h}^\kappa. \tag{17}$$

Each $h_k$ is then standardized to have zero mean and unit (population) standard deviation. We generate an event time $E$ and a censoring time $C$ using log-normal models with independent Gaussian noise:

$$\log T_E = b_{0e} + \boldsymbol{b}_{1e}\boldsymbol{h}^\epsilon + \boldsymbol{b}_{2e}\boldsymbol{h}^\kappa + \sigma_e\,\epsilon_e, \qquad \epsilon_e \sim \mathcal{N}(0,1), \tag{18}$$

$$\log T_C = b_{0c} + \boldsymbol{b}_{2c}\boldsymbol{h}^\kappa + \boldsymbol{b}_{3c}\boldsymbol{h}^\gamma + \sigma_c\,\epsilon_c, \qquad \epsilon_c \sim \mathcal{N}(0,1). \tag{19}$$

Thus, the event mechanism depends on $(\boldsymbol{h}^\epsilon, \boldsymbol{h}^\kappa)$, while the censoring mechanism depends on $(\boldsymbol{h}^\kappa, \boldsymbol{h}^\gamma)$, and $\boldsymbol{h}^\kappa$ is a shared factor that induces dependence between $T_E$ and $T_C$. The observed time and event indicator can then be derived.

**semi-METABRIC** The data generation process (DGP) begins with a real dataset **METABRIC**, denoted as $\mathcal{D} = \{(\boldsymbol{x}_i, t_i, \delta_i)\}_{i=1}^N$. We fit a `SALaD` (`DeepSurv`) model to estimate the event-time distribution $\widehat{S}_E(t \mid \boldsymbol{x}_i)$ and the censoring-time distribution $\widehat{S}_C(t \mid \boldsymbol{x}_i)$ using the modified dataset $\tilde{\mathcal{D}} = \{(\boldsymbol{x}_i, t_i, 1 - \delta_i)\}_{i=1}^N$, which swaps the event/censoring indicators.

For each instance $i$, we use the fitted model to obtain the predicted event and censoring distributions (with Breslow estimator for the baseline hazard) based on their features. Then, we apply *inverse transform sampling* to draw synthetic event times $e_i$ and censoring times $c_i$ from these respective distributions. If a distribution is not fully supported (*e.g.*, due to truncation), we apply linear extrapolation as described in Qi et al. (2023a). The observed time and event indicator are then derived accordingly.

This DGP ensures that the resulting semi-synthetic dataset contains:

- real-world features,

- realistic event-time distributions,

- and realistic censoring-time distributions.

Additionally, because both event and censoring distributions are conditioned on features, the DGP naturally introduces *conditional dependence between $E$ and $C$*.

Compared to the method proposed in Qi et al. (2023a), our approach has the added benefit of preserving the *original dataset's censoring rate*. In contrast, the method in Qi et al. (2023a) (which enforces conditional independent censoring) often results in unrealistically low censoring rates.

A potential limitation of our DGP is that the strength of dependency between $E$ and $C$ is not directly controllable. Since the event and censoring distributions are estimated using a empirical model, it is possible that the features exert strong influence on one but not the other. For example, the features may have large coefficients in the event model but small or negligible ones in the censoring model. In such cases, even if confounding factors exist, the actual dependency between $E$ and $C$ may be weak.

### D.2. Evaluation Metrics

**Concordance Index** Discriminative performance measures how accurately a model ranks subjects based on predicted risk scores. It is often measured by the concordance index (CI) (Harrell Jr et al., 1996), which is the proportion of all comparable subject pairs whose predicted and outcome orders are concordant, defined as

$$\mathrm{CI}\left(\widehat{r}(\cdot)\right) = \frac{\sum_{i=1}^N \sum_{j=1; j \neq i}^N \delta_i \cdot \mathbb{1}[\,t_i < t_j\,] \cdot \mathbb{1}[\,\widehat{r}(\boldsymbol{x}_i) > \widehat{r}(\boldsymbol{x}_j)\,]}{\sum_{i=1}^N \sum_{j=1; j \neq i}^N \delta_i \cdot \mathbb{1}[\,t_i < t_j\,]},$$

where $\widehat{r}(\boldsymbol{x}_i)$ denotes the predicted risk score of $i$-th subject. The risk scores are calculated as the negative of the predicted median survival time in our experiments.

**Integrated Brier Score** The Brier Score (BS) measures the accuracy of probabilistic predictions at a target time $t^*$ (Graf et al., 1999). It calculates the squared error between the predicted probability $\widehat{S}_E(t^* \mid \boldsymbol{x}_A)$ and the survival status of patient $A$ at $t^*$. To incorporate the censored subjects in the calculation, BS requires estimating the IPCW (Robins & Finkelstein,

2000) to uniformly transfer a censored subject's weight (originally 1) to subjects with known status at that time (Vock et al., 2016b). Formally:

$$\text{BS}\left(t^*; \widehat{S}_E(\cdot \mid \cdot), S_C(\cdot)\right) = \frac{1}{N} \sum_{i=1}^{N} \left( \frac{\delta_i \cdot \mathbb{1}[t_i \leq t^*] \cdot \widehat{S}_E(t^* \mid \boldsymbol{x}_i)^2}{S_C(t_i)} + \frac{\mathbb{1}[t_i > t^*] \cdot (1 - \widehat{S}_E(t^* \mid \boldsymbol{x}_i))^2}{S_C(t^*)} \right),$$

where $S_C(t)$ represents the marginal time-to-censoring probability at time $t$. Usually, it is estimated using the KM estimator on the censoring distribution; we refer to its reciprocal $\frac{1}{S_C(t)}$ as the IPCW. Then, we can calculate the integrated Brier score (IBS), which is the expectation of BS over all time, defined as:

$$\text{IBS}\left(t_{\max}; \widehat{S}_E(\cdot \mid \cdot), S_C(\cdot)\right) = \frac{1}{t_{\max}} \cdot \int_{u=0}^{t_{\max}} \text{BS}\left(u; \widehat{S}_E(\cdot \mid \cdot), S_C(\cdot)\right) \, du,$$

where $t_{\max}$ is usually the maximum event time of the combined training and validation datasets.

We note that the existing implementation of IPCW-BS and IPCW-IBS does not incorporate features, and thus implicitly assumes marginal independent censoring (*i.e.*, $E \perp C$). This violates our assumption of conditional independence (Assumption 2.2), which posits $E \perp C \mid \boldsymbol{X}$. However, to the best of our knowledge, no unbiased estimator of IPCW exists under conditional independence – this is a fundamental issue discussed earlier in Section 2.

In fact, choosing an arbitrary model (*e.g.*, CoxPH) to estimate covariate-dependent IPCW weights would introduce bias in favor of that model when computing the IBS, undermining its utility as a fair evaluation metric. Therefore, we deliberately use the marginal IPCW to ensure comparability across models, even though it does not satisfy the conditional independence assumption.

**Mean Absolute Error**   Mean absolute error measures the accuracy of the predicted time-to-event, *i.e.*, $\frac{1}{N}|e_i - \widehat{e}_i|$. To more accurately estimate error in the presence of censoring, Qi et al. (2023a) introduced a method called MAE using pseudo-observation (MAE-PO). The method tries to "de-censoring" a censored instance: MAE-PO uses pseudo-observations (Andersen & Pohar Perme, 2010), a jackknife estimator that calculates the contribution of a censored subject to the overall KM estimator. Furthermore, it weights the errors using a confidence score based on the estimated "de-censoring" times.

**Distribution Calibration**   To estimate the calibration of the predicted time-to-event distributions, we use distribution calibration (D-cal) (Haider et al., 2020), which checks whether the predicted survival probabilities at event times $e_i$ over the $\boldsymbol{x}_i$ in the dataset, $\{\widehat{S}_E(e_i \mid \boldsymbol{x}_i)\}_{i=1}^{N}$, matches the standard uniform distribution $\mathcal{U}_{[0,1]}$ (according to the probability integral transform (Angus, 1994)). A model is D-calibrated if the proportion of patients' prediction $\{\widehat{S}_E(e_i \mid \boldsymbol{x}_i)\}_{i=1}^{N}$ belongs to any percentile range $[\rho_1, \rho_2]$ is similar to the length of the interval $\rho_2 - \rho_1$. The calibration level can be quantified via a $\chi^2$ test. In this paper, we choose to report the number of times a model does not produce a significantly non-calibrated result over 10 random experiments (*i.e.*, the number of times the $p$-value $> 0.05$). To integrate censored patients, we can "blur" each censored patient uniformly to the subsequent probability intervals after its predicted survival probability at observed time $\widehat{S}_E(c_i \mid \boldsymbol{x}_i)$, please refer Haider et al. (2020) for more details.

All these metrics are implemented in the SurvivalEVAL package (Qi et al., 2023b). For semi-synthetic datasets, the event times for all instances in the test set are provided (with event indicator equals 1), while use the censored labels for training.

### D.3. Benchmarks

In this paper, we compared the proposed framework, SALaD, against the original baseline survival models and their two-branch variants 2B. The four baseline survival models selected for comparison are: DeepSurv, N-MTLR, AFTNN-Weibull, and AFTNN-LogLogistic. Here we briefly introduce them:

- The Cox proportional hazard (CoxPH) model is a semi-parametric model with a proportional hazard assumption (Cox, 1972). It consists of a population-level baseline hazard function (a non-parametric function) and a partial hazard function (relative hazard/relative risks). The original CoxPH model only predicts a relative hazard score (risk score) for each patient using the partial hazard function. To make ISD prediction, we use the Breslow method (Breslow, 1975) to estimate the population-level baseline hazard function. Here we use its neural network extension (DeepSurv) (Katzman et al., 2018).

*Table 3.* Comparison between the benchmarking survival analysis algorithms.

| Method | Continuous Time | Parametric | Proportional Hazard | Direct Output |
|---|---|---|---|---|
| `Nnet-survival` | ✗ | ✗ | ✗ | Hazard function |
| `RSF` | ✗ | ✗ | ✗ | Survival function |
| `GB` | ✗ | ✗ | ✓ | (Relative) hazard ratio |
| `DeepHit` | ✗ | ✗ | ✗ | PMF function |
| `CoxTime` | ✗ | ✗ | ✗ | (Relative) hazard ratios |
| `IWSG` | ✗ | ✗ | ✗ | PMF function |
| `SODEN` | ✓ | ✗ | ✗ | Hazard function |
| `CQRNN` | ✗ | ✗ | ✗ | Quantile function |
| `DCSurvival` | ✓ | ✗ | ✗ | Survival function |
| `SurvivalBoost` | ✗ | ✗ | ✗ | Hazard function |
| `DeepSurv` | ✗ | ✗ | ✓ | (Relative) hazard ratio |
| `N-MTLR` | ✗ | ✗ | ✗ | Survival function |
| `AFTNN-Weibull` | ✓ | ✓ | ✓ | (Relative) hazard ratio |
| `AFTNN-LogLogistic` | ✓ | ✓ | ✗ | (Relative) hazard ratio |

- The multi-task logistic regression model (`MTLR`) is a model that estimates the probability of survival of patients at each of a vector of discretized time points $[t_1, t_2, \ldots, t_{\max}]$. To achieve that, `MTLR` set up a series of logistic regression models, for each time point. Here we use its neural network extension (`N-MTLR`) proposed by Fotso (2018). The number of discrete times is determined by the square root of the number of uncensored patients, and we use quantiles to divide those uncensored instances evenly into each time interval, as suggested in (Jin, 2015; Haider et al., 2020).

- The Accelerated Failure Time (AFT) model is a widely used parametric survival model that directly models the effect of features on the survival time by assuming that the logarithm of the failure time is linearly related to the features (Prentice & Kalbfleisch, 1979). When the failure time follows a Weibull distribution, the `AFT-Weibull` is particularly useful due to its flexibility in accommodating increasing or decreasing hazard functions. The Weibull distribution is parameterized by a shape parameter $k$ and a scale parameter $\lambda$. Here we use its neural network extension, `AFTNN-Weibull` (Norman et al., 2024).

- Another popular choice for the AFT model is the LogLogistic distribution, which is particularly useful when modeling survival data with non-monotonic hazard functions, such as those exhibiting an initial increase followed by a decrease (Farewell & Prentice, 1977). The LogLogistic distribution is parameterized by a scale parameter $\lambda$ and a shape parameter $\gamma$. The `AFT-LogLogistic` model is particularly advantageous in cases where hazard functions peak and then decline, making it a valuable choice for certain biomedical and reliability studies. Here we use its neural network extension, `AFTNN-LogLogistic` (Norman et al., 2024).

These four baseline models were selected for their diversity. Specifically,

- `DeepSurv` and `N-MTLR` are non-parametric models, whereas `AFTNN-Weibull` and `AFTNN-LogLogistic` are fully parametric models (which follow the Weibull and LogLogistic distributions, respectively);

- `DeepSurv` and `N-MTLR` produce discrete-time distributions, whereas `AFTNN-Weibull` and `AFTNN-LogLogistic` can generate continuous outputs;

- `DeepSurv` and `AFTNN-Weibull` share the proportional hazards assumption, while `N-MTLR` and `AFTNN-LogLogistic` do not.

We reimplement all these four methods in our code base, as well as their `2B` and `SALaD` extensions.

Additionally, we evaluated 10 SOTA models, each selected for its distinct methodological properties:

- `Nnet-survival` (Gensheimer & Narasimhan, 2019) is a discrete-time survival model that directly estimates the conditional hazard function rather than the conditional probability mass function (PMF) or survival function. It parameterizes hazards at multiple discrete time points and optimizes the survival likelihood. The model is also known as Partial Logistic Regression (Biganzoli et al., 1998), it follows the same discretization strategy as `N-MTLR` for a fair comparison. We implemented this model using the `pycox` package (Pölsterl, 2020).

- Random survival forest (`RSF`) (Ishwaran et al., 2008) is an ensemble learning method that extends the Random Forest for handling survival data. It constructs multiple survival trees, each trained on a bootstrap sample of the data, using randomly selected features at each split. The trees are built using a splitting criterion, such as log-rank statistics or maximization of survival difference, to optimally partition the data while accounting for censoring. The model is implemented in `scikit-survival` packages (Pölsterl, 2020).

- Gradient Boost (`GB`) (Hothorn et al., 2006) is an ensemble method with component-wise least squares as the base learner. In this method, each base learner is a simple least squares regression model applied to individual features, allowing for efficient and interpretable updates at each boosting stage. We use the 100 boosting stages with partial likelihood loss for optimization and 100% subsampling for fitting each base learner. The model is implemented in `scikit-survival` packages (Pölsterl, 2020).

- `DeepHit` (Lee et al., 2018) is also a discrete model, like `N-MTLR` and `Nnet-survival`. It models the conditional PMF of the event for each individual (and the PMF can be used to calculate the survival distribution accordingly). Furthermore, apart from the standard likelihood loss, it also contains a ranking loss term approximating the partial likelihood loss (Cox, 1975). The discretization strategy matches that of `N-MTLR` and `Nnet-survival` for consistency. The model is implemented in `pycox` packages (Kvamme et al., 2019).

- `CoxTime` (Kvamme et al., 2019) is a neural network-based extension of the `DeepSurv` model that does not assume proportional hazards. Unlike DeepSurv, which only adds a multilayer perceptron (MLP) layer, `CoxTime` models interactions between time $t$ and features $x_i$, offering greater flexibility. The model is implemented in `pycox` packages (Kvamme et al., 2019).

- Inverse-weighted Survival Games (`IWSG`) (Han et al., 2021) is a framework that incorporates IPCW to mitigate censoring-induced bias. It formulates survival analysis as a two-player game, where the event model and the censoring model serve as the players. Each model's loss function is derived using IPCW estimates based on the predictions of the other model (see Figure 2b). Both models are discrete and aim to estimate the PMF of the respective distributions. We reimplemented `IWSG` using the original code from Han et al. (2021). In the original code, the number of training epochs is treated as a hyperparameter, and models from each epoch are saved. The best-performing failure and censoring models are then selected independently and separately based on their validation performance (binomial log-likelihood, BLL). For fair comparison with other models, we adopt an early stopping strategy in our implementation: training is stopped at the epoch with the lowest BLL on the validation set, and the models from that epoch are used as the final models.

- Survival model through Ordinary Differential Equation Networks (`SODEN`) (Tang et al., 2022) is a continuous-time survival model that avoids (semi-)parametric assumptions. It estimates the survival function and its derivative (*i.e.*, the probability density function) using an ordinary differential equation solver. `SODEN` is highly general, encompassing discrete, continuous, parametric, and non-parametric models. However, training is computationally expensive and prone to numerical instability. We implemented `SODEN` based on the example code from Chen et al. (2024).

- Censored Quantile Regression Neural Networks (`CQRNN`) (Pearce et al., 2022) is a quantile regression-based method (Portnoy, 2003) that predicts survival time given a specified quantile rather than survival probability at a given time. It employs Portnoy's pinball loss and an expectation-maximization step for estimation. To ensure monotonicity in survival predictions, we apply bootstrap-rearranging post-processing (Chernozhukov et al., 2010). We reimplemented this model based on the original code from Pearce et al. (2022).

- Deep Copula Survival (`DCSurvival`) (Zhang et al., 2024) is a copula-based deep survival model designed for dependent censoring. It jointly models the event-time marginal distribution, the censoring-time marginal distribution, and their dependency structure through a learnable Archimedean copula. The model is trained end-to-end by maximizing the dependent-censoring likelihood, with neural density estimators (Rindt et al., 2022) used for flexible survival and censoring marginals. We reimplemented `DCSurvival` based on the original code from Zhang et al. (2024).

- `SurvivalBoost` (Alberge et al., 2025) employs gradient boosting trees to estimate the cumulative incidence function in competing risks settings. The model is trained by minimizing a negative log-likelihood objective reweighted using IPCW (as illustrated by Figure 2b). As in `IWSG`, the IPCW component in `SurvivalBoost` is a conditional estimator, aligning closely with the objective in this study, where we propose to explicitly model censoring distributions.

### D.4. Model Implementation

Table 4 presents the parameters and hyperparameter settings for the neural-network-based methods. The selected hyperparameter ranges allow for systematic exploration and optimization of the model's performance. The selected hyperparameter ranges allow for systematic exploration and optimization of the model's performance. We use the random hyperparameter optimization strategy with a maximum of 60 search trials to identify the best-performing hyperparameter configuration. Tables 5, 6, 7, 8 list all optimal hyper-parameters of the `SALaD` method on four baselines, used for each dataset in the paper's experiments.

*Table 4.* Parameters and hyper-parameters setting for the neural-network-based methods. The first half of the table (above the dashed line) is the fixed parameters and their corresponding values. The second half of the table (below the dashed line) includes the hyper-parameters and the respective ranges for fine-tuning.

| Name | Value(s) |
|---|---|
| Optimizer | `AdamW` |
| Activation function | `ReLU` |
| Maximum epoch $B$ | `10000` |
| Batch normalization | `True` |
| Representation normalization | `True` |
| Early stopping | `True` |
| Batch size | `256` |
| Learning rate $\alpha$ | `0.001` |
| IPM discrepancy function $\text{disc}(\cdot,\cdot)$ | `{linear-MMD, RBF-MMD}` |
| Weight decay $\lambda$ | `{0.01, 0.1}` |
| Balancing weight $\beta_1$ | `{0.1, 1, 10}` |
| Balancing weight $\beta_2$ | `{0.001, 0.01, 0.1}` |
| Representation networks $\Phi^\epsilon(\cdot), \Phi^\kappa(\cdot), \Phi^\gamma(\cdot)$ | `{[64], [64, 64], [64, 64, 16],` `[32], [32, 32], [32, 32, 16]}` |
| Distribution networks $\Psi_E^{(1)}(\cdot), \Psi_E^{(2)}(\cdot,\cdot), \Psi_C^{(1)}(\cdot), \Psi_C^{(2)}(\cdot,\cdot)$ | `{[], [16], [16, 16]}` |
| Dropout probability $p$ | `{0.2, 0.4, 0.6}` |

*Table 5.* Optimal hyper-parameters setting for `SALaD` (`DeepSurv`) methods. The symbols are consistent with the hyper-parameters in the second half of Table 4.

| | $\text{disc}(\cdot,\cdot)$ | $\lambda$ | $\beta_1$ | $\beta_2$ | Rep. nets | Dist. nets | $p$ |
|---|---|---|---|---|---|---|---|
| **HFCR** | RBF-MMD | 0.01 | 10 | 0.01 | [32, 32, 16] | [16, 16] | 0.6 |
| **PBC** | linear-MMD | 0.1 | 1 | 0.001 | [32] | [] | 0.4 |
| **GBM** | linear-MMD | 0.1 | 0.1 | 0.001 | [32, 32] | [16, 16] | 0.4 |
| **GBSG** | linear-MMD | 0.01 | 0.1 | 0.001 | [64] | [16, 16] | 0.4 |
| **METABRIC** | linear-MMD | 0.01 | 0.1 | 0.001 | [32, 32] | [16, 16] | 0.4 |
| **NACD** | linear-MMD | 0.01 | 1 | 0.01 | [32] | [] | 0.4 |
| **SUPPORT** | RBF-MMD | 0.1 | 0.1 | 0.1 | [64] | [] | 0.4 |
| **MIMIC-IV** | linear-MMD | 0.01 | 0.1 | 0.01 | [64, 64] | [16] | 0.6 |

*Table 6.* Optimal hyper-parameters setting for `SALaD` (`N-MTLR`) methods. The symbols are consistent with the hyper-parameters in the second half of Table 4.

| | disc$(\cdot,\cdot)$ | $\lambda$ | $\beta_1$ | $\beta_2$ | Rep. nets | Dist. nets | $p$ |
|---|---|---|---|---|---|---|---|
| **HFCR** | RBF-MMD | 0.01 | 10 | 0.1 | [64, 64] | [] | 0.4 |
| **PBC** | RBF-MMD | 0.1 | 0.1 | 0.01 | [64] | [] | 0.4 |
| **GBM** | RBF-MMD | 0.1 | 10 | 0.1 | [32, 32, 16] | [] | 0.4 |
| **GBSG** | RBF-MMD | 0.01 | 10 | 0.1 | [32] | [16] | 0.4 |
| **METABRIC** | RBF-MMD | 0.1 | 0.1 | 0.01 | [32, 32] | [16, 16] | 0.4 |
| **NACD** | RBF-MMD | 0.1 | 0.1 | 0.1 | [32] | [] | 0.4 |
| **SUPPORT** | linear-MMD | 0.1 | 0.1 | 0.1 | [64] | [16] | 0.6 |
| **MIMIC-IV** | linear-MMD | 0.1 | 1 | 0.001 | [64, 64] | [16] | 0.6 |

*Table 7.* Optimal hyper-parameters setting for `SALaD` (`AFTNN-Weibull`) methods. The symbols are consistent with the hyper-parameters in the second half of Table 4.

| | disc$(\cdot,\cdot)$ | $\lambda$ | $\beta_1$ | $\beta_2$ | Rep. nets | Dist. nets | $p$ |
|---|---|---|---|---|---|---|---|
| **HFCR** | linear-MMD | 0.01 | 1 | 0.001 | [32, 32, 16] | [16, 16] | 0.4 |
| **PBC** | linear-MMD | 0.01 | 10 | 0.001 | [32] | [] | 0.4 |
| **GBM** | linear-MMD | 0.01 | 10 | 0.1 | [32, 32] | [16, 16] | 0.4 |
| **GBSG** | RBF-MMD | 0.1 | 0.1 | 0.1 | [32] | [] | 0.4 |
| **METABRIC** | RBF-MMD | 0.1 | 1 | 0.001 | [64, 64] | [16] | 0.6 |
| **NACD** | linear-MMD | 0.1 | 0.1 | 0.1 | [32] | [16, 16] | 0.6 |
| **SUPPORT** | linear-MMD | 0.1 | 10 | 0.001 | [64] | [16, 16] | 0.6 |
| **MIMIC-IV** | RBF-MMD | 0.1 | 1 | 0.001 | [64, 64] | [16] | 0.6 |

# E. Complete Results

## E.1. Comparison with SOTA

Tables 11, 12, 13, 14, 15, 16, 17 and 18 present benchmarking results for various SOTA survival analysis methods across eight real-world datasets: **HFCR**, **PBC**, **GBM**, **GBSG**, **METABRIC**, **NACD**, **SUPPORT**, and **MIMIC-IV**, respectively.

Key Observations:

- Consistent CI Improvement: across all datasets, `SALaD` consistently enhances CI, demonstrating its ability to improve predictive ranking accuracy compared to the original baseline methods.

- Lower IBS for More Reliable Predictions: `SALaD` often reduces IBS, indicating better overall prediction accuracy.

- Maintained or Improved Calibration (D-cal): In most cases, `SALaD` maintains comparable or higher D-calibrated times, ensuring well-calibrated risk predictions.

Across all datasets, `SALaD` consistently enhances the performance of baseline survival analysis methods, demonstrating its effectiveness in improving risk prediction, calibration, and robustness. Its ability to increase CI, reduce IBS, and maintain well-calibrated predictions highlights its potential as a superior survival modeling approach.

## E.2. Comparison between baselines, `2B`, and `SALaD`

Table 19-26 demonstrate the ablation studies.

For the **HFCR**, **PBC**, **GBM**, **GBSG**, **METABRIC**, **NACD**, and **MIMIC-IV** datasets, `SALaD` achieves the highest CI across all the other variant models, indicating its strong predictive capability. The improvements in CI are particularly notable when applied to `N-MTLR` and AFT models, where the `SALaD`-enhanced versions consistently yield the best results.

Regarding IBS, MAE, and D-cal, `SALaD` generally provides competitive or superior results. While `2B` occasionally demonstrates marginally better IBS and D-cal values in some cases, the differences are extremely small, often within

*Table 8.* Optimal hyper-parameters setting for `SALaD` (`AFTNN-LogLogistic`) methods. The symbols are consistent with the hyper-parameters in the second half of Table 4.

|  | $\text{disc}(\cdot,\cdot)$ | $\lambda$ | $\beta_1$ | $\beta_2$ | Rep. nets | Dist. nets | $p$ |
|---|---|---|---|---|---|---|---|
| **HFCR** | RBF-MMD | 0.01 | 10 | 0.01 | [32, 32, 16] | [16, 16] | 0.6 |
| **PBC** | linear-MMD | 0.1 | 1 | 0.001 | [32] | [] | 0.4 |
| **GBM** | RBF-MMD | 0.1 | 10 | 0.001 | [64, 64, 16] | [16, 16] | 0.4 |
| **GBSG** | RBF-MMD | 0.1 | 0.1 | 0.01 | [64, 64, 16] | [16, 16] | 0.4 |
| **METABRIC** | linear-MMD | 0.1 | 10 | 0.001 | [64, 64] | [16, 16] | 0.6 |
| **NACD** | linear-MMD | 0.1 | 0.1 | 0.001 | [32, 32, 16] | [16, 16] | 0.6 |
| **SUPPORT** | RBF-MMD | 0.1 | 0.1 | 0.01 | [64] | [16, 16] | 0.6 |
| **MIMIC-IV** | RBF-MMD | 0.01 | 1 | 0.01 | [64, 64] | [16, 16] | 0.6 |

*Table 9.* Comparison of baselines, `2B`, and `SALaD` on the **semi-SUPPORT** dataset. Results are reported as the mean and standard deviation over 10 runs. Within each comparison group, the better-performing method is indicated with underline.

| Method | CI(%) ↑ | IBS ↓ | MAE ↓ | D-cal ↑ |
|---|---|---|---|---|
| `DeepSurv` | $78.53_{\pm 1.59}$ | $0.0136_{\pm 0.0091}$ | $12.131_{\pm 1.295}$ | 3/10 |
| `2B(DeepSurv)` | $78.61_{\pm 1.82}$ | $0.0122_{\pm 0.0063}$ | $12.126_{\pm 1.312}$ | 3/10 |
| `SALaD(DeepSurv)` | $\underline{78.81}_{\pm 3.16}$ | $\underline{0.0112}_{\pm 0.0059}$ | $\underline{12.124}_{\pm 1.297}$ | $\underline{6/10}$ |
| `N-MTLR` | $78.43_{\pm 1.59}$ | $0.0111_{\pm 0.0050}$ | $\underline{12.124}_{\pm 1.305}$ | 10/10 |
| `2B(N-MTLR)` | $78.52_{\pm 1.54}$ | $\underline{0.0111}_{\pm 0.0048}$ | $12.128_{\pm 1.304}$ | 10/10 |
| `SALaD(N-MTLR)` | $\underline{78.59}_{\pm 1.72}$ | $0.0140_{\pm 0.0062}$ | $12.128_{\pm 1.232}$ | 10/10 |
| `AFTNN-Weibull` | $78.61_{\pm 1.33}$ | $0.0103_{\pm 0.0041}$ | $12.231_{\pm 1.268}$ | 0/10 |
| `2B(AFTNN-Weibull)` | $78.71_{\pm 1.55}$ | $0.0101_{\pm 0.0039}$ | $12.259_{\pm 1.333}$ | 0/10 |
| `SALaD(AFTNN-Weibull)` | $\underline{79.81}_{\pm 1.97}$ | $\underline{0.0090}_{\pm 0.0039}$ | $\underline{12.077}_{\pm 1.271}$ | 0/10 |
| `AFTNN-LogLogistic` | $78.60_{\pm 1.78}$ | $0.0099_{\pm 0.0064}$ | $12.154_{\pm 1.334}$ | 1/10 |
| `2B(AFTNN-LogLogistic)` | $78.65_{\pm 1.53}$ | $\underline{0.0087}_{\pm 0.0034}$ | $\underline{12.030}_{\pm 1.274}$ | 3/10 |
| `SALaD(AFTNN-LogLogistic)` | $\underline{80.69}_{\pm 2.75}$ | $0.0098_{\pm 0.0057}$ | $12.077_{\pm 1.282}$ | $\underline{3/10}$ |

statistical fluctuations, and do not outweigh the clear advantage that `SALaD` offers in CI. Since CI is the primary performance metric in survival analysis, `SALaD`'s consistent improvements in CI highlight its robustness and effectiveness.

### E.3. Adversarial training.

Originally popularised by the generative adversarial network (GAN) framework of Goodfellow et al. (2014), adversarial training casts learning as a two–player minimax game in which a **primary model** (the *generator* or feature encoder) competes against an **adversary** that tries to expose its weaknesses. By alternately optimising the players' objectives,

$$\min_{\theta} \max_{\phi} \ \mathcal{L}_{\text{task}}(f_\theta) - \lambda \, \mathcal{L}_{\text{adv}}\big(D_\phi \circ f_\theta\big),$$

The method encourages the primary model to produce representations that *fool* the adversary, thereby injecting desired invariances without hand-crafted regularisers. The idea has quickly migrated beyond image synthesis: Ganin et al. (2016) use a gradient-reversal layer to align feature distributions across domains.

In our context, adversarial training can be used to enforce the desired disentanglement properties in Propositions 4.2 and 4.3. Specifically, building on the proposed architecture in Figure 2d, we introduce two additional (adversarial) representation heads: $\Psi_C^{\text{adv}}(\cdot)$ and $\Psi_E^{\text{adv}}(\cdot)$. These networks attempt to predict the censoring and event distributions from the *irrelevant* representations, thereby functioning as adversaries.

*Table 10.* Comparison of baselines, 2B, and SALaD on the **semi-METABRIC** dataset. Results are reported as the mean and standard deviation over 10 runs. Within each comparison group, the better-performing method is indicated with underline. Note that Cox-based methods (DeepSurv and its extensions) are excluded from this comparison because the data generation process (DGP) is based on a CoxPH model, which would introduce bias in their favor.

| Method | CI(%) ↑ | IBS ↓ | MAE ↓ | D-cal ↑ |
|---|---|---|---|---|
| N-MTLR | $63.90_{\pm 3.57}$ | $0.1614_{\pm 0.0344}$ | $2618.075_{\pm 252.805}$ | $10/10$ |
| 2B (N-MTLR) | $63.96_{\pm 3.82}$ | $\underline{0.1583}_{\pm 0.0340}$ | $2630.615_{\pm 243.070}$ | $\underline{10/10}$ |
| SALaD (N-MTLR) | $\underline{65.08}_{\pm 3.57}$ | $0.1604_{\pm 0.0281}$ | $2664.831_{\pm 281.578}$ | $9/10$ |
| AFTNN-Weibull | $\underline{63.69}_{\pm 0.38}$ | $0.1846_{\pm 0.0290}$ | $4025.766_{\pm 658.561}$ | $\underline{9/10}$ |
| 2B (AFTNN-Weibull) | $63.39_{\pm 3.73}$ | $\underline{0.1808}_{\pm 0.0265}$ | $3860.822_{\pm 552.026}$ | $9/10$ |
| SALaD (AFTNN-Weibull) | $\underline{63.80}_{\pm 3.90}$ | $0.1838_{\pm 0.0294}$ | $\underline{3642.241}_{\pm 795.458}$ | $6/10$ |
| AFTNN-LogLogistic | $62.97_{\pm 3.33}$ | $0.1988_{\pm 0.0278}$ | $4221.478_{\pm 546.332}$ | $5/10$ |
| 2B (AFTNN-LogLogistic) | $63.49_{\pm 3.60}$ | $0.1963_{\pm 0.0275}$ | $\underline{3513.241}_{\pm 398.595}$ | $2/10$ |
| SALaD (AFTNN-LogLogistic) | $\underline{63.59}_{\pm 3.97}$ | $\underline{0.1930}_{\pm 0.0238}$ | $3877.654_{\pm 598.778}$ | $7/10$ |

The adversarial objective is defined as the following likelihood (to be maximized):

$$\mathcal{L}^{\mathrm{adv}} = \prod_{i=1}^{N} \left[ \widehat{f}_E^{\mathrm{adv}}(t_i \mid \boldsymbol{h}_i^\gamma) \cdot \widehat{S}_C^{\mathrm{adv}}(t_i \mid \boldsymbol{h}_i^\epsilon) \right]^{\delta_i} \cdot \left[ \widehat{f}_C^{\mathrm{adv}}(t_i \mid \boldsymbol{h}_i^\epsilon) \cdot \widehat{S}_E^{\mathrm{adv}}(t_i \mid \boldsymbol{h}_i^\gamma) \right]^{1-\delta_i},$$

where $\widehat{f}_E^{\mathrm{adv}}$ and $\widehat{S}_E^{\mathrm{adv}}$ are the density and survival predictions produced by $\Psi_E^{\mathrm{adv}}(\cdot)$ given $\boldsymbol{h}_i^\gamma$, and $\widehat{f}_C^{\mathrm{adv}}$ and $\widehat{S}_C^{\mathrm{adv}}$ are those produced by $\Psi_C^{\mathrm{adv}}(\cdot)$ given $\boldsymbol{h}_i^\epsilon$.

This setup encourages $\boldsymbol{H}^\epsilon$ to retain information useful for predicting event times (via the primary likelihood objectives in Equations (3) and (4)), while discouraging its ability to predict censoring times – captured by maximizing the adversarial likelihood. Symmetrically, $\boldsymbol{H}^\gamma$ is optimized to predict censoring but not event times.

The overall loss function becomes:

$$\mathcal{L}_{\mathrm{total}} = -\log(\mathcal{L}^{(2)} \cdot \mathcal{L}^{(1)}) + \beta_3 \cdot \log(\mathcal{L}^{\mathrm{adv}}) + \beta_1 \cdot \mathcal{L}_{\mathrm{orth}} + \lambda \cdot \mathcal{L}_{\mathrm{reg}},$$

where the adversarial term $\log(\mathcal{L}^{\mathrm{adv}})$ replaces the IPM-based alignment losses, and $\mathcal{L}$orth and $\mathcal{L}_{\mathrm{reg}}$ are the orthogonality and regularization terms, respectively.

Table 27 compares this adversarial training method with the proposed SALaD method.

### E.4. Effect of Decomposed Representation

To further illustrate how SALaD decomposes latent representations into event-specific, confounder, and censoring-specific factors, we present two-dimensional $t$-distributed stochastic neighbor embedding (t-SNE) visualizations of latent representations trained with SALaD (N-MTLR) in Figure 6. The first row of this figure shows only the uncensored instances with their event times. The second row of this figure shows only the censored instances with their censoring times.

In this figure, we can easily see that

- The event-specific latent representation $\boldsymbol{H}^\epsilon$ can distinguish the event times (upper left), where the instances with late event times are on the right side of the plot. However, it cannot distinguish the censoring times (lower left), as the early and late censoring instances are entangled.

- The confounding latent representation $\boldsymbol{H}^\kappa$ can distinguish both the event times (upper middle) and censoring times (lower middle). For both cases, the earlier times are located on the right side of the figure.

- The censoring-specific latent representation $\boldsymbol{H}^\gamma$ can distinguish the censoring times (lower right), where the instances with late event times are on the lower left side of the plot. However, it does not have as good distinguishability wrt the event times (upper right).

*Table 11.* Benchmarking on the **HFCR** dataset, with mean and standard deviation over 10 experiments. The best performance is highlighted in **bold**, and for each pair of "$\rho$" and "SALaD ($\rho$)", the better one is indicated using underline.

| Method | C-index(%) ↑ | IBS ↓ | MAE-PO ↓ | D-cal ↑ |
|---|---|---|---|---|
| Nnet-survival | $68.31_{\pm 9.48}$ | $0.2305_{\pm 0.1804}$ | $248.780_{\pm 33.548}$ | 8/10 |
| RSF | $68.52_{\pm 8.95}$ | $0.1511_{\pm 0.0191}$ | $370.398_{\pm 71.335}$ | **10/10** |
| GB | $64.98_{\pm 10.47}$ | $0.1551_{\pm 0.0222}$ | $248.378_{\pm 41.274}$ | **10/10** |
| DeepHit | $64.24_{\pm 11.36}$ | $0.2958_{\pm 0.0554}$ | $267.043_{\pm 19.558}$ | 0/10 |
| CoxTime | $66.00_{\pm 9.46}$ | $0.1510_{\pm 0.0182}$ | $248.419_{\pm 33.593}$ | **10/10** |
| IWSG | $68.62_{\pm 8.53}$ | $0.1475_{\pm 0.0276}$ | $330.049_{\pm 173.190}$ | **10/10** |
| SODEN | $60.96_{\pm 7.61}$ | $0.2607_{\pm 0.1551}$ | $858.726_{\pm 600.231}$ | 7/10 |
| CQRNN | $70.23_{\pm 8.67}$ | $0.3526_{\pm 0.1077}$ | $252.868_{\pm 24.496}$ | 0/10 |
| DCSurvival | $61.56_{\pm 8.89}$ | $0.1770_{\pm 0.0363}$ | $254.350_{\pm 12.895}$ | **10/10** |
| SurvivalBoost | $64.58_{\pm 5.88}$ | $0.1649_{\pm 0.0298}$ | $769.580_{\pm 457.549}$ | **10/10** |
| DeepSurv | $67.36_{\pm 10.58}$ | $0.1467_{\pm 0.0253}$ | $245.740_{\pm 31.907}$ | **10/10** |
| SALaD(DeepSurv) | $\mathbf{72.44_{\pm 9.91}}$ | $\underline{0.1459_{\pm 0.0238}}$ | $\underline{234.612_{\pm 48.801}}$ | **10/10** |
| N-MTLR | $69.72_{\pm 9.25}$ | $0.1459_{\pm 0.0293}$ | $233.654_{\pm 42.157}$ | **10/10** |
| SALaD(N-MTLR) | $\underline{70.70_{\pm 8.61}}$ | $\mathbf{0.1453_{\pm 0.0220}}$ | $\mathbf{231.878_{\pm 23.480}}$ | **10/10** |
| AFTNN-Weibull | $69.05_{\pm 7.39}$ | $0.1593_{\pm 0.0262}$ | $275.962_{\pm 83.844}$ | **10/10** |
| SALaD(AFTNN-Weibull) | $\underline{69.48_{\pm 10.46}}$ | $\underline{0.1516_{\pm 0.0389}}$ | $\underline{261.607_{\pm 56.790}}$ | **10/10** |
| AFTNN-LogLogistic | $67.80_{\pm 8.33}$ | $0.1734_{\pm 0.0346}$ | $377.943_{\pm 127.823}$ | **10/10** |
| SALaD(AFTNN-LogLogistic) | $\underline{70.05_{\pm 15.49}}$ | $\underline{0.1635_{\pm 0.0236}}$ | $\underline{370.736_{\pm 144.442}}$ | **10/10** |

### E.5. Complexity Analysis

We acknowledge that the proposed SALaD method introduces more time and space complexity to the baseline models. However, in this section, we perform complexity analysis to show that the SALaD model exhibits a moderate increase in training time and parameter count compared to its 2B and baseline counterparts. Overall, it achieves a favorable trade-off between computational cost and predictive performance, remaining comparable to other state-of-the-art survival models.

The computational efficiency of the SALaD method is evaluated based on training time, inference time, parameter count, and floating-point operations (FLOPs). The analysis is conducted on the **SUPPORT** dataset using 10 random seeds to ensure robustness. Detailed settings of the computational complexity analysis are presented in the following section.

Compared to its corresponding baseline and 2B variants, SALaD exhibits an increase in computational demand, particularly in training time and model size. This is expected given its design, which incorporates additional complexity to enhance predictive performance.

While SALaD requires more computation than 2B and its respective baseline (*e.g.*, DeepSurv, N-MTLR, AFTNN-Weibull, and AFTNN-LogLogistic), the increase in training time remains moderate. For instance, SALaD (DeepSurv) trains in approximately 50.3 seconds, which is higher than 2B (DeepSurv) (18.9s) but remains manageable. Similarly, SALaD (N-MTLR) and SALaD (AFTNN-LogLogistic) train in 37.8s and 98.0s, respectively, showing an increase but not a drastic overhead. The inference time remains largely unaffected, demonstrating that the additional complexity does not compromise the model's efficiency in real-time applications.

When compared to other state-of-the-art (SOTA) survival models, SALaD achieves a favorable trade-off between computational cost and predictive capability. For instance, models like SODEN exhibit significantly higher training time (1472.3s), making them less practical for many applications. Additionally, SALaD maintains a parameter count and FLOPs comparable to its baselines, ensuring that its resource requirements remain within reasonable bounds.

Nnet-survival, DeepHit, and CoxTime achieve shorter training and inference times compared to other methods. However, this does not necessarily indicate superior computational efficiency on the algorithm. These models are imple-

*Table 12.* Benchmarking on the **PBC** dataset. Notations are consistent with Table 11.

| Method | CI(%) ↑ | IBS ↓ | MAE-PO ↓ | D-cal ↑ |
|---|---|---|---|---|
| Nnet-survival | $76.88_{\pm 8.75}$ | $0.2042_{\pm 0.1107}$ | $2788.976_{\pm 838.865}$ | 9/10 |
| RSF | $79.56_{\pm 6.23}$ | $0.1465_{\pm 0.0298}$ | $3544.205_{\pm 785.535}$ | **10/10** |
| GB | $79.63_{\pm 6.36}$ | $0.1642_{\pm 0.0511}$ | $2681.729_{\pm 775.421}$ | **10/10** |
| DeepHit | $78.74_{\pm 6.21}$ | $0.2867_{\pm 0.0863}$ | $3100.378_{\pm 849.149}$ | 0/10 |
| CoxTime | $76.89_{\pm 8.40}$ | $0.1741_{\pm 0.0497}$ | $2783.454_{\pm 829.431}$ | **10/10** |
| IWSG | $81.86_{\pm 5.64}$ | $0.1468_{\pm 0.0341}$ | $2662.089_{\pm 995.426}$ | **10/10** |
| SODEN | $67.74_{\pm 12.57}$ | $0.2786_{\pm 0.0798}$ | $3786.240_{\pm 1070.175}$ | 0/10 |
| CQRNN | $78.33_{\pm 5.97}$ | $0.4115_{\pm 0.1361}$ | $3132.329_{\pm 929.698}$ | 0/10 |
| SurvivalBoost | $80.27_{\pm 6.14}$ | $0.1860_{\pm 0.0552}$ | $3027.887_{\pm 742.533}$ | **10/10** |
| DeepSurv | $80.91_{\pm 4.69}$ | $0.1641_{\pm 0.0539}$ | $2973.744_{\pm 897.652}$ | **10/10** |
| SALaD (DeepSurv) | $\underline{81.91}_{\pm 4.93}$ | $\underline{0.1491}_{\pm 0.0441}$ | $\underline{2832.108}_{\pm 693.064}$ | **10/10** |
| N-MTLR | $81.31_{\pm 5.55}$ | $0.1481_{\pm 0.0295}$ | $2561.176_{\pm 703.162}$ | **10/10** |
| SALaD (N-MTLR) | $\underline{81.52}_{\pm 5.78}$ | $\mathbf{0.1449}_{\pm \mathbf{0.0378}}$ | $\mathbf{2529.522}_{\pm \mathbf{828.327}}$ | **10/10** |
| AFTNN-Weibull | $77.43_{\pm 4.40}$ | $0.1929_{\pm 0.0634}$ | $7267.801_{\pm 3341.530}$ | **10/10** |
| SALaD (AFTNN-Weibull) | $\mathbf{81.94}_{\pm \mathbf{4.76}}$ | $\underline{0.1586}_{\pm 0.0314}$ | $14681.209_{\pm 4004.739}$ | **10/10** |
| AFTNN-LogLogistic | $80.75_{\pm 4.63}$ | $0.1758_{\pm 0.0482}$ | $37633.629_{\pm 89330.883}$ | **10/10** |
| SALaD (AFTNN-LogLogistic) | $\underline{81.22}_{\pm 4.14}$ | $\underline{0.1707}_{\pm 0.0217}$ | $\underline{14678.786}_{\pm 4353.339}$ | **10/10** |

mented in the `pycox` package[3], which leverages Numba (Lam et al., 2015) for compilation, whereas other models rely on pure Python or PyTorch.

In summary, `SALaD` achieves competitive computational efficiency, offering improved modeling capacity with a slight increase in training time while maintaining inference efficiency. Its computation remains comparable to other SOTA methods, reinforcing its practicality in survival analysis tasks.

### E.5.1. COMPUTATIONAL ANALYSIS SETTINGS

The computational efficiency of the `SALaD` model is evaluated based on training time, inference time, parameter count, and floating-point operations (FLOPs). The analysis is conducted on the **SUPPORT** dataset using 10 random seeds to ensure robustness. To maintain a fair comparison, all models are trained with the same architecture and hyperparameters, with the exception of `SODEN`, which is trained using only a single hidden layer with 8 neurons[4]. The hyperparameters used across all models are as follows:

- IPM distance function $\text{IPM}(\cdot, \cdot)$: RBF-MMD

- Weight decay $\lambda$: 0.001

- Balancing weight $\beta_1$: 10

- Balancing weight $\beta_2$: 0.001

- Hidden neurons: [64, 64]

- Dropout probability: 0.4

- Learning rate $\alpha$: 0.001

The number of trainable parameters and FLOPs for each model are computed using the Python library `thop`.

---

[3] https://github.com/havakv/pycox
[4] This is because `SODEN` will lead to numerical overflow if we use too many neurons.

*Table 13.* Benchmarking on the **GBM** dataset. Notations are consistent with Table 11.

| Method | C-index(%) ↑ | IBS ↓ | MAE-PO ↓ | D-cal ↑ |
|---|---|---|---|---|
| Nnet-survival | $67.56_{\pm 3.27}$ | $0.0689_{\pm 0.0195}$ | $\mathbf{373.547_{\pm 53.999}}$ | 9/10 |
| RSF | $68.24_{\pm 1.99}$ | $0.0702_{\pm 0.0187}$ | $373.719_{\pm 58.699}$ | **10/10** |
| GB | $66.92_{\pm 2.67}$ | $0.0694_{\pm 0.0187}$ | $388.572_{\pm 57.038}$ | **10/10** |
| DeepHit | $64.22_{\pm 4.56}$ | $0.0751_{\pm 0.0208}$ | $409.681_{\pm 69.751}$ | **10/10** |
| CoxTime | $66.09_{\pm 3.69}$ | $0.0698_{\pm 0.0199}$ | $390.787_{\pm 56.130}$ | **10/10** |
| IWSG | $67.58_{\pm 2.84}$ | $0.0696_{\pm 0.0182}$ | $376.569_{\pm 56.994}$ | **10/10** |
| SODEN | $59.44_{\pm 6.40}$ | $0.1474_{\pm 0.0588}$ | $1026.501_{\pm 553.850}$ | 0/10 |
| CQRNN | $62.40_{\pm 6.91}$ | $0.1078_{\pm 0.0313}$ | $495.495_{\pm 141.014}$ | 0/10 |
| SurvivalBoost | $67.89_{\pm 3.21}$ | $0.0712_{\pm 0.0196}$ | $383.691_{\pm 56.833}$ | 8/10 |
| DeepSurv | $67.08_{\pm 2.59}$ | $0.0691_{\pm 0.0198}$ | $380.012_{\pm 56.639}$ | **10/10** |
| SALaD (DeepSurv) | $\underline{67.84_{\pm 2.83}}$ | $\mathbf{0.0684_{\pm 0.0195}}$ | $378.797_{\pm 58.390}$ | **10/10** |
| N-MTLR | $68.04_{\pm 2.66}$ | $0.0696_{\pm 0.0188}$ | $377.373_{\pm 57.426}$ | **10/10** |
| SALaD (N-MTLR) | $\underline{68.56_{\pm 2.13}}$ | $\underline{0.0689_{\pm 0.0176}}$ | $\underline{377.271_{\pm 50.899}}$ | **10/10** |
| AFTNN-Weibull | $66.20_{\pm 1.70}$ | $\underline{0.0732_{\pm 0.0178}}$ | $\underline{408.149_{\pm 73.191}}$ | 9/10 |
| SALaD (AFTNN-Weibull) | $\underline{67.39_{\pm 1.58}}$ | $0.0773_{\pm 0.0266}$ | $448.301_{\pm 186.327}$ | 8/10 |
| AFTNN-LogLogistic | $68.03_{\pm 2.53}$ | $\underline{0.0744_{\pm 0.0151}}$ | $390.248_{\pm 45.071}$ | 9/10 |
| SALaD (AFTNN-LogLogistic) | $\underline{\mathbf{68.64_{\pm 2.72}}}$ | $0.0769_{\pm 0.0163}$ | $\underline{385.883_{\pm 50.120}}$ | 9/10 |

*Table 14.* Benchmarking on the **GBSG** dataset. Notations are consistent with Table 11.

| Method | C-index(%) ↑ | IBS ↓ | MAE-PO ↓ | D-cal ↑ |
|---|---|---|---|---|
| Nnet-survival | $68.65_{\pm 3.35}$ | $0.1943_{\pm 0.1070}$ | $1435.856_{\pm 402.877}$ | 9/10 |
| RSF | $67.37_{\pm 4.64}$ | $0.1671_{\pm 0.0112}$ | $1645.449_{\pm 285.327}$ | **10/10** |
| GB | $65.63_{\pm 3.49}$ | $0.1644_{\pm 0.0131}$ | $1492.729_{\pm 348.883}$ | **10/10** |
| DeepHit | $66.18_{\pm 4.74}$ | $0.2614_{\pm 0.0490}$ | $1502.598_{\pm 333.418}$ | 0/10 |
| CoxTime | $64.37_{\pm 2.96}$ | $0.1696_{\pm 0.0216}$ | $1489.939_{\pm 415.866}$ | **10/10** |
| IWSG | $69.00_{\pm 3.03}$ | $0.1578_{\pm 0.0153}$ | $1414.552_{\pm 316.015}$ | **10/10** |
| SODEN | $56.91_{\pm 6.63}$ | $0.2504_{\pm 0.0393}$ | $5772.550_{\pm 4793.4157}$ | 1/10 |
| CQRNN | $67.71_{\pm 3.14}$ | $0.2595_{\pm 0.0579}$ | $1382.191_{\pm 337.135}$ | 0/10 |
| SurvivalBoost | $63.43_{\pm 5.28}$ | $0.1852_{\pm 0.0238}$ | $1465.684_{\pm 308.699}$ | 7/10 |
| DeepSurv | $\underline{68.53_{\pm 2.90}}$ | $0.1592_{\pm 0.0106}$ | $1474.481_{\pm 326.988}$ | **10/10** |
| SALaD (DeepSurv) | $68.49_{\pm 3.48}$ | $\mathbf{0.1563_{\pm 0.0113}}$ | $\underline{1441.355_{\pm 297.823}}$ | **10/10** |
| N-MTLR | $68.65_{\pm 3.61}$ | $0.1593_{\pm 0.0210}$ | $\underline{1378.425_{\pm 327.393}}$ | **10/10** |
| SALaD (N-MTLR) | $\mathbf{69.08_{\pm 4.95}}$ | $\underline{0.1576_{\pm 0.0156}}$ | $\mathbf{1356.486_{\pm 330.189}}$ | **10/10** |
| AFTNN-Weibull | $64.61_{\pm 4.78}$ | $0.1770_{\pm 0.0181}$ | $10949.725_{\pm 11621.157}$ | **10/10** |
| SALaD (AFTNN-Weibull) | $\underline{68.51_{\pm 3.60}}$ | $\underline{0.1756_{\pm 0.0157}}$ | $2298.406_{\pm 394.830}$ | **10/10** |
| AFTNN-LogLogistic | $67.21_{\pm 5.39}$ | $\underline{0.1743_{\pm 0.0085}}$ | $\underline{1549.247_{\pm 352.367}}$ | 5/10 |
| SALaD (AFTNN-LogLogistic) | $\underline{68.48_{\pm 6.03}}$ | $0.1748_{\pm 0.0160}$ | $1746.032_{\pm 392.453}$ | $\underline{9/10}$ |

*Table 15.* Benchmarking on the **METABRIC** dataset. Notations are consistent with Table 11.

| Method | C-index(%) ↑ | IBS ↓ | MAE-PO ↓ | D-cal ↑ |
|---|---|---|---|---|
| Nnet-survival | $68.16_{\pm 4.20}$ | $0.1870_{\pm 0.0300}$ | $3270.884_{\pm 290.981}$ | **10/10** |
| RSF | $68.64_{\pm 4.20}$ | $0.1832_{\pm 0.0234}$ | $3888.363_{\pm 522.871}$ | **10/10** |
| GB | $65.34_{\pm 5.76}$ | $0.1820_{\pm 0.0294}$ | $3299.974_{\pm 334.261}$ | **10/10** |
| DeepHit | $69.09_{\pm 3.36}$ | $0.2615_{\pm 0.0477}$ | $3538.250_{\pm 380.383}$ | 0/10 |
| CoxTime | $66.33_{\pm 3.81}$ | $0.1783_{\pm 0.0293}$ | $3297.123_{\pm 324.781}$ | **10/10** |
| IWSG | $68.37_{\pm 5.10}$ | $0.1784_{\pm 0.0271}$ | $3181.499_{\pm 367.816}$ | **10/10** |
| SODEN | $60.59_{\pm 4.25}$ | $0.3444_{\pm 0.0638}$ | $36692.133_{\pm 46292.962}$ | 0/10 |
| CQRNN | $66.44_{\pm 3.75}$ | $0.3342_{\pm 0.0538}$ | $3369.661_{\pm 343.344}$ | 0/10 |
| SurvivalBoost | $66.59_{\pm 5.42}$ | $0.1825_{\pm 0.0402}$ | $3176.944_{\pm 382.011}$ | 7/10 |
| DeepSurv | $68.48_{\pm 3.98}$ | $0.1793_{\pm 0.0284}$ | $\underline{3221.162_{\pm 347.012}}$ | **10/10** |
| SALaD(DeepSurv) | $\underline{69.33_{\pm 3.68}}$ | $\mathbf{0.1782_{\pm 0.0290}}$ | $3226.559_{\pm 382.018}$ | **10/10** |
| N-MTLR | $69.07_{\pm 3.99}$ | $0.1803_{\pm 0.0336}$ | $3094.728_{\pm 382.784}$ | **10/10** |
| SALaD(N-MTLR) | $\underline{69.48_{\pm 2.82}}$ | $\underline{0.1796_{\pm 0.0266}}$ | $\mathbf{3087.063_{\pm 367.562}}$ | **10/10** |
| AFTNN-Weibull | $66.99_{\pm 3.61}$ | $\underline{0.2038_{\pm 0.0163}}$ | $\underline{5686.467_{\pm 1441.061}}$ | 5/10 |
| SALaD(AFTNN-Weibull) | $\underline{69.06_{\pm 3.73}}$ | $0.2127_{\pm 0.0264}$ | $9494.431_{\pm 2143.021}$ | $\underline{\mathbf{10/10}}$ |
| AFTNN-LogLogistic | $68.04_{\pm 3.22}$ | $\underline{0.1994_{\pm 0.0190}}$ | $9690.535_{\pm 5052.471}$ | 7/10 |
| SALaD(AFTNN-LogLogistic) | $\mathbf{69.56_{\pm 3.77}}$ | $0.2012_{\pm 0.0199}$ | $\underline{7831.367_{\pm 2562.350}}$ | $\underline{8/10}$ |

*Table 16.* Benchmarking on the **NACD** dataset. Notations are consistent with Table 11.

| Method | C-index(%) ↑ | IBS ↓ | MAE-PO ↓ | D-cal ↑ |
|---|---|---|---|---|
| Nnet-survival | $75.70_{\pm 1.96}$ | $0.1437_{\pm 0.0208}$ | $21.151_{\pm 1.698}$ | **10/10** |
| RSF | $75.05_{\pm 2.05}$ | $0.1408_{\pm 0.0149}$ | $20.839_{\pm 1.586}$ | **10/10** |
| GB | $73.28_{\pm 2.09}$ | $0.1493_{\pm 0.0157}$ | $21.727_{\pm 1.212}$ | **10/10** |
| DeepHit | $75.70_{\pm 2.06}$ | $0.1932_{\pm 0.0273}$ | $25.035_{\pm 1.171}$ | 0/10 |
| CoxTime | $73.68_{\pm 2.24}$ | $0.1409_{\pm 0.0144}$ | $21.202_{\pm 1.286}$ | **10/10** |
| IWSG | $75.59_{\pm 2.21}$ | $\mathbf{0.1347_{\pm 0.0119}}$ | $20.985_{\pm 1.136}$ | **10/10** |
| SODEN | $61.19_{\pm 6.95}$ | $0.2724_{\pm 0.0263}$ | $31.644_{\pm 1.266}$ | 0/10 |
| CQRNN | $75.24_{\pm 2.12}$ | $0.1715_{\pm 0.0290}$ | $20.935_{\pm 1.215}$ | **10/10** |
| DCSurvival | $74.84_{\pm 1.96}$ | $0.1479_{\pm 0.0220}$ | $21.117_{\pm 1.293}$ | 5/10 |
| SurvivalBoost | $75.25_{\pm 1.83}$ | $0.1464_{\pm 0.0255}$ | $\mathbf{20.533_{\pm 1.382}}$ | 3/10 |
| DeepSurv | $75.29_{\pm 1.92}$ | $\underline{0.1374_{\pm 0.0111}}$ | $21.698_{\pm 1.181}$ | 9/10 |
| SALaD(DeepSurv) | $\underline{75.52_{\pm 2.21}}$ | $0.1376_{\pm 0.0096}$ | $\underline{21.394_{\pm 1.414}}$ | $\underline{\mathbf{10/10}}$ |
| N-MTLR | $75.52_{\pm 2.02}$ | $0.1372_{\pm 0.0155}$ | $20.767_{\pm 1.265}$ | **10/10** |
| SALaD(N-MTLR) | $\underline{75.57_{\pm 2.13}}$ | $\underline{0.1356_{\pm 0.0108}}$ | $\underline{20.693_{\pm 1.151}}$ | 8/10 |
| AFTNN-Weibull | $74.86_{\pm 1.73}$ | $\underline{0.1413_{\pm 0.0102}}$ | $24.575_{\pm 1.984}$ | 4/10 |
| SALaD(AFTNN-Weibull) | $\underline{76.52_{\pm 1.84}}$ | $0.1511_{\pm 0.0118}$ | $24.648_{\pm 2.180}$ | $\underline{7/10}$ |
| AFTNN-LogLogistic | $75.00_{\pm 2.12}$ | $\underline{0.1402_{\pm 0.0150}}$ | $23.637_{\pm 2.315}$ | **10/10** |
| SALaD(AFTNN-LogLogistic) | $\mathbf{76.86_{\pm 2.06}}$ | $0.1432_{\pm 0.0105}$ | $\underline{22.208_{\pm 1.234}}$ | **10/10** |

*Table 17.* Benchmarking on the **SUPPORT** dataset. Notations are consistent with Table 11.

| Method | C-index(%) ↑ | IBS ↓ | MAE-PO ↓ | D-cal ↑ |
|---|---|---|---|---|
| Nnet-survival | $69.26_{\pm0.86}$ | $0.1616_{\pm0.0078}$ | $669.242_{\pm13.712}$ | 7/10 |
| RSF | $69.87_{\pm0.81}$ | $0.1609_{\pm0.0063}$ | $680.679_{\pm20.001}$ | 3/10 |
| GB | $66.79_{\pm0.71}$ | $0.1704_{\pm0.0072}$ | $693.526_{\pm11.262}$ | 6/10 |
| DeepHit | $68.80_{\pm0.78}$ | $0.2092_{\pm0.0084}$ | $737.259_{\pm10.120}$ | 0/10 |
| CoxTime | $68.68_{\pm0.87}$ | $0.1651_{\pm0.0071}$ | $668.605_{\pm17.902}$ | 5/10 |
| IWSG | $66.80_{\pm0.82}$ | $0.1609_{\pm0.0074}$ | $714.248_{\pm19.268}$ | 2/10 |
| SODEN | $64.25_{\pm1.99}$ | $0.2855_{\pm0.1070}$ | $3937.410_{\pm6659.257}$ | 0/10 |
| CQRNN | $68.75_{\pm0.70}$ | $0.1860_{\pm0.0164}$ | $703.652_{\pm10.417}$ | 0/10 |
| DCSurvival | $68.83_{\pm1.04}$ | $0.1550_{\pm0.0090}$ | $680.806_{\pm15.146}$ | 0/10 |
| SurvivalBoost | $69.54_{\pm1.09}$ | $\mathbf{0.1449_{\pm0.0070}}$ | $672.830_{\pm17.370}$ | 2/10 |
| DeepSurv | $68.82_{\pm0.72}$ | $0.1637_{\pm0.0081}$ | $685.827_{\pm23.752}$ | **9/10** |
| SALaD (DeepSurv) | $\underline{68.90_{\pm0.50}}$ | $\underline{0.1609_{\pm0.0070}}$ | $\underline{685.269_{\pm15.232}}$ | **9/10** |
| N-MTLR | $69.00_{\pm0.86}$ | $0.1611_{\pm0.0076}$ | $670.871_{\pm15.735}$ | **9/10** |
| SALaD (N-MTLR) | $\underline{69.23_{\pm0.80}}$ | $\underline{0.1592_{\pm0.0083}}$ | $\mathbf{667.836_{\pm15.878}}$ | **9/10** |
| AFTNN-Weibull | $68.04_{\pm0.61}$ | $0.1639_{\pm0.0077}$ | $703.423_{\pm31.857}$ | 0/10 |
| SALaD (AFTNN-Weibull) | $\underline{69.32_{\pm1.02}}$ | $\underline{0.1638_{\pm0.0085}}$ | $707.741_{\pm30.946}$ | 0/10 |
| AFTNN-LogLogistic | $69.36_{\pm0.71}$ | $0.1652_{\pm0.0077}$ | $673.718_{\pm16.445}$ | 0/10 |
| SALaD (AFTNN-LogLogistic) | $\mathbf{69.88_{\pm0.83}}$ | $\underline{0.1651_{\pm0.0075}}$ | $\underline{670.408_{\pm17.918}}$ | $\underline{1/10}$ |

*Table 18.* Benchmarking on the **MIMIC-IV** dataset. Notations are consistent with Table 11.

| Method | C-index(%) ↑ | IBS ↓ | MAE-PO ↓ | D-cal ↑ |
|---|---|---|---|---|
| Nnet-survival | $73.57_{\pm1.00}$ | $0.0403_{\pm0.0037}$ | $277.152_{\pm16.710}$ | 5/10 |
| RSF | $73.58_{\pm1.02}$ | $0.0397_{\pm0.0032}$ | $270.876_{\pm15.975}$ | 2/10 |
| GB | $69.90_{\pm0.92}$ | $0.0406_{\pm0.0035}$ | $309.339_{\pm84.209}$ | **10/10** |
| DeepHit | $73.86_{\pm0.77}$ | $0.0416_{\pm0.0035}$ | $287.632_{\pm16.982}$ | 8/10 |
| CoxTime | $74.22_{\pm0.96}$ | $\mathbf{0.0385_{\pm0.0031}}$ | $273.106_{\pm24.951}$ | 9/10 |
| IWSG | $66.32_{\pm1.82}$ | $0.0397_{\pm0.0032}$ | $289.898_{\pm16.862}$ | 0/10 |
| CQRNN | $64.20_{\pm1.02}$ | $0.0477_{\pm0.0046}$ | $313.410_{\pm17.710}$ | 0/10 |
| SurvivalBoost | $73.24_{\pm1.09}$ | $0.0386_{\pm0.0030}$ | $270.415_{\pm13.232}$ | 6/10 |
| DeepSurv | $75.04_{\pm0.73}$ | $0.0386_{\pm0.0033}$ | $264.175_{\pm16.225}$ | 8/10 |
| SALaD (DeepSurv) | $\underline{75.60_{\pm1.10}}$ | $\mathbf{0.0385_{\pm0.0032}}$ | $\underline{263.202_{\pm15.941}}$ | 8/10 |
| N-MTLR | $75.76_{\pm1.05}$ | $0.0388_{\pm0.0032}$ | $263.364_{\pm15.540}$ | **10/10** |
| SALaD (N-MTLR) | $\underline{76.18_{\pm1.09}}$ | $\underline{0.0387_{\pm0.0032}}$ | $\mathbf{261.945_{\pm15.249}}$ | **10/10** |
| AFTNN-Weibull | $74.54_{\pm0.86}$ | $0.0386_{\pm0.0032}$ | $274.142_{\pm24.529}$ | 0/10 |
| SALaD (AFTNN-Weibull) | $\underline{76.24_{\pm1.09}}$ | $\underline{0.0389_{\pm0.0032}}$ | $\underline{269.539_{\pm14.643}}$ | 0/10 |
| AFTNN-LogLogistic | $75.36_{\pm9.47}$ | $0.0395_{\pm0.0033}$ | $268.271_{\pm14.487}$ | 6/10 |
| SALaD (AFTNN-LogLogistic) | $\mathbf{76.91_{\pm1.06}}$ | $\underline{0.0393_{\pm0.0030}}$ | $\underline{264.299_{\pm14.981}}$ | **10/10** |

*Table 19.* Ablation on the **HFCR** datasets, with mean and standard deviation over 10 experiments. For each group of comparison, the better one is indicated using underline.

| Method | CI(%) $\uparrow$ | IBS $\downarrow$ | MAE $\downarrow$ | D-cal $\uparrow$ |
|---|---|---|---|---|
| 2B (DeepSurv) | $68.76_{\pm 11.40}$ | $0.1485_{\pm 0.0257}$ | $242.411_{\pm 39.409}$ | 10/10 |
| SALaD (DeepSurv, $\beta_1 = 0$) | $66.45_{\pm 9.72}$ | $0.1514_{\pm 0.0227}$ | $365.417_{\pm 172.007}$ | 10/10 |
| SALaD (DeepSurv, $\beta_2 = 0$) | $71.02_{\pm 9.61}$ | $0.1468_{\pm 0.0234}$ | $273.122_{\pm 91.254}$ | 10/10 |
| SALaD (DeepSurv) | $\underline{72.44}_{\pm 9.91}$ | $\underline{0.1459}_{\pm 0.0238}$ | $\underline{234.612}_{\pm 48.801}$ | 10/10 |
| 2B (N-MTLR) | $69.93_{\pm 11.88}$ | $0.1480_{\pm 0.0292}$ | $233.470_{\pm 34.217}$ | 10/10 |
| SALaD (N-MTLR, $\beta_1 = 0$) | $70.60_{\pm 9.36}$ | $0.1473_{\pm 0.0316}$ | $233.275_{\pm 25.915}$ | 10/10 |
| SALaD (N-MTLR, $\beta_2 = 0$) | $70.10_{\pm 9.47}$ | $0.1529_{\pm 0.0253}$ | $245.691_{\pm 23.531}$ | 10/10 |
| SALaD (N-MTLR) | $\underline{70.70}_{\pm 8.61}$ | $\underline{0.1453}_{\pm 0.0220}$ | $\underline{231.878}_{\pm 23.480}$ | 10/10 |
| 2B (AFTNN-Weibull) | $69.29_{\pm 9.11}$ | $0.1558_{\pm 0.0215}$ | $263.912_{\pm 81.868}$ | 10/10 |
| SALaD (AFTNN-Weibull, $\beta_1 = 0$) | $68.71_{\pm 10.36}$ | $0.1786_{\pm 0.0246}$ | $447.415_{\pm 106.455}$ | 10/10 |
| SALaD (AFTNN-Weibull, $\beta_2 = 0$) | $67.09_{\pm 13.11}$ | $0.1668_{\pm 0.0272}$ | $988.828_{\pm 881.852}$ | 10/10 |
| SALaD (AFTNN-Weibull) | $\underline{69.48}_{\pm 10.46}$ | $\underline{0.1516}_{\pm 0.0389}$ | $\underline{261.607}_{\pm 56.790}$ | 10/10 |
| 2B (AFTNN-LogLogistic) | $68.25_{\pm 9.06}$ | $0.1669_{\pm 0.0245}$ | $784.196_{\pm 611.142}$ | 10/10 |
| SALaD (AFTNN-LogLogistic, $\beta_1 = 0$) | $68.00_{\pm 7.62}$ | $0.1808_{\pm 0.0300}$ | $1013.644_{\pm 428.057}$ | 10/10 |
| SALaD (AFTNN-LogLogistic, $\beta_2 = 0$) | $69.12_{\pm 12.18}$ | $0.1701_{\pm 0.0311}$ | $734.095_{\pm 311.142}$ | 10/10 |
| SALaD (AFTNN-LogLogistic) | $\underline{70.05}_{\pm 15.49}$ | $\underline{0.1635}_{\pm 0.0236}$ | $\underline{370.736}_{\pm 144.442}$ | 10/10 |

**Reproducibility settings** All experiments were implemented and executed in `Python` 3.11.7 with `PyTorch` 2.0.1, `PyCox` 0.2.3, `scikit-survival` 0.22.2, and `SurvivalEVAL` 0.3.0. All computational experiments was conducted on a server equipped with an Intel Xeon Silver 4216 CPU, and an NVIDIA Tesla V100 SXM2 32GB GPU.

*Table 20.* Ablation on the **PBC** datasets, with the same notation as Table 19.

| Method | CI(%) ↑ | IBS ↓ | MAE ↓ | D-cal ↑ |
|---|---|---|---|---|
| 2B (DeepSurv) | $81.21_{\pm 4.97}$ | $0.1644_{\pm 0.0433}$ | $4338.602_{\pm 2969.036}$ | 10/10 |
| SALaD (DeepSurv, $\beta_1 = 0$) | $81.55_{\pm 4.55}$ | $0.1620_{\pm 0.0498}$ | $5354.108_{\pm 2261.152}$ | 10/10 |
| SALaD (DeepSurv, $\beta_2 = 0$) | $81.47_{\pm 4.40}$ | $0.1659_{\pm 0.0431}$ | $4780.838_{\pm 1459.729}$ | 10/10 |
| SALaD (DeepSurv) | $\underline{81.91}_{\pm 4.93}$ | $\underline{0.1491}_{\pm 0.0441}$ | $\underline{2832.108}_{\pm 693.064}$ | 10/10 |
| 2B (N-MTLR) | $81.43_{\pm 5.13}$ | $\underline{0.1416}_{\pm 0.0348}$ | $\underline{2561.433}_{\pm 758.755}$ | 10/10 |
| SALaD (N-MTLR, $\beta_1 = 0$) | $80.31_{\pm 6.50}$ | $0.1587_{\pm 0.0580}$ | $4382.268_{\pm 2524.998}$ | 10/10 |
| SALaD (N-MTLR, $\beta_2 = 0$) | $81.37_{\pm 6.08}$ | $0.1456_{\pm 0.0365}$ | $2614.975_{\pm 808.089}$ | 10/10 |
| SALaD (N-MTLR) | $\underline{81.52}_{\pm 5.78}$ | $0.1449_{\pm 0.0378}$ | $2529.522_{\pm 828.327}$ | 10/10 |
| 2B (AFTNN-Weibull) | $80.71_{\pm 5.67}$ | $0.1711_{\pm 0.0405}$ | $9645.091_{\pm 4191.260}$ | 10/10 |
| SALaD (AFTNN-Weibull, $\beta_1 = 0$) | $77.10_{\pm 5.60}$ | $0.1992_{\pm 0.0669}$ | $11703.342_{\pm 4562.648}$ | $\underline{10/10}$ |
| SALaD (AFTNN-Weibull, $\beta_2 = 0$) | $80.96_{\pm 6.44}$ | $0.1815_{\pm 0.0438}$ | $285220.470_{\pm 450220.890}$ | $\underline{9/10}$ |
| SALaD (AFTNN-Weibull) | $\underline{81.94}_{\pm 4.76}$ | $\underline{0.1586}_{\pm 0.0314}$ | $14681.209_{\pm 4004.739}$ | $\underline{10/10}$ |
| 2B (AFTNN-LogLogistic) | $79.17_{\pm 4.69}$ | $0.1735_{\pm 0.0469}$ | $15391.885_{\pm 32965.041}$ | 10/10 |
| SALaD (AFTNN-LogLogistic, $\beta_1 = 0$) | $79.40_{\pm 5.14}$ | $0.1764_{\pm 0.0397}$ | $18829.080_{\pm 15403.893}$ | 10/10 |
| SALaD (AFTNN-LogLogistic, $\beta_2 = 0$) | $80.45_{\pm 6.39}$ | $0.1900_{\pm 0.0433}$ | $215959.710_{\pm 431214.350}$ | 10/10 |
| SALaD (AFTNN-LogLogistic) | $\underline{81.22}_{\pm 4.14}$ | $\underline{0.1707}_{\pm 0.0217}$ | $\underline{14678.786}_{\pm 4353.3394}$ | 10/10 |

*Table 21.* Ablation on the **GBM**, with the same notation as in Table 19.

| Method | CI(%) ↑ | IBS ↓ | MAE ↓ | D-cal ↑ |
|---|---|---|---|---|
| 2B (DeepSurv) | $67.05_{\pm 2.94}$ | $0.0702_{\pm 0.0206}$ | $386.574_{\pm 64.275}$ | 10/10 |
| SALaD (DeepSurv, $\beta_1 = 0$) | $67.21_{\pm 3.33}$ | $0.0726_{\pm 0.0174}$ | $391.971_{\pm 51.934}$ | 10/10 |
| SALaD (DeepSurv, $\beta_2 = 0$) | $67.04_{\pm 2.48}$ | $0.0695_{\pm 0.0193}$ | $381.174_{\pm 60.718}$ | 10/10 |
| SALaD (DeepSurv) | $\underline{67.84}_{\pm 2.83}$ | $\underline{0.0684}_{\pm 0.0195}$ | $378.797_{\pm 58.390}$ | 10/10 |
| 2B (N-MTLR) | $67.93_{\pm 3.18}$ | $0.0698_{\pm 0.0187}$ | $377.636_{\pm 55.888}$ | $\underline{10/10}$ |
| SALaD (N-MTLR, $\beta_1 = 0$) | $68.00_{\pm 2.70}$ | $0.0700_{\pm 0.0181}$ | $381.486_{\pm 46.832}$ | $\underline{8/10}$ |
| SALaD (N-MTLR, $\beta_2 = 0$) | $67.96_{\pm 2.52}$ | $0.0692_{\pm 0.0180}$ | $382.460_{\pm 51.843}$ | $\underline{9/10}$ |
| SALaD (N-MTLR) | $\underline{68.56}_{\pm 2.13}$ | $0.0689_{\pm 0.0176}$ | $\underline{377.271}_{\pm 50.899}$ | $\underline{10/10}$ |
| 2B (AFTNN-Weibull) | $\underline{67.63}_{\pm 2.68}$ | $\underline{0.0694}_{\pm 0.0178}$ | $\underline{379.612}_{\pm 53.426}$ | $\underline{9/10}$ |
| SALaD (AFTNN-Weibull, $\beta_1 = 0$) | $64.82_{\pm 4.29}$ | $0.0741_{\pm 0.0209}$ | $404.737_{\pm 81.139}$ | $\underline{4/10}$ |
| SALaD (AFTNN-Weibull, $\beta_2 = 0$) | $66.97_{\pm 2.24}$ | $0.1325_{\pm 0.0168}$ | $673.635_{\pm 53.067}$ | $\underline{0/10}$ |
| SALaD (AFTNN-Weibull) | $67.39_{\pm 1.58}$ | $0.0773_{\pm 0.0266}$ | $448.301_{\pm 186.327}$ | $\underline{8/10}$ |
| 2B (AFTNN-LogLogistic) | $68.17_{\pm 3.04}$ | $\underline{0.0750}_{\pm 0.0155}$ | $388.808_{\pm 62.199}$ | 7/10 |
| SALaD (AFTNN-LogLogistic, $\beta_1 = 0$) | $66.54_{\pm 2.03}$ | $\underline{0.0761}_{\pm 0.0184}$ | $395.130_{\pm 61.415}$ | 5/10 |
| SALaD (AFTNN-LogLogistic, $\beta_2 = 0$) | $66.93_{\pm 2.08}$ | $0.0757_{\pm 0.0175}$ | $389.585_{\pm 61.348}$ | 7/10 |
| SALaD (AFTNN-LogLogistic) | $\underline{68.64}_{\pm 2.72}$ | $0.0769_{\pm 0.0163}$ | $385.883_{\pm 50.120}$ | $\underline{9/10}$ |

*Table 22.* Ablation on the **GBSG** dataset, with the same notation as in Table 19.

| Method | CI(%) ↑ | IBS ↓ | MAE ↓ | D-cal ↑ |
|---|---|---|---|---|
| 2B (DeepSurv) | $67.19_{\pm 2.80}$ | $0.1601_{\pm 0.0139}$ | $1632.650_{\pm 475.930}$ | 10/10 |
| SALaD (DeepSurv, $\beta_1 = 0$) | $64.38_{\pm 7.50}$ | $0.1637_{\pm 0.0128}$ | $1450.547_{\pm 319.770}$ | 10/10 |
| SALaD (DeepSurv, $\beta_2 = 0$) | $66.87_{\pm 3.69}$ | $0.1578_{\pm 0.0102}$ | $1446.856_{\pm 292.488}$ | 10/10 |
| SALaD (DeepSurv) | $\underline{68.49}_{\pm 3.48}$ | $\underline{0.1563}_{\pm 0.0113}$ | $\underline{1441.355}_{\pm 297.823}$ | 10/10 |
| 2B (N-MTLR) | $68.95_{\pm 3.11}$ | $0.1610_{\pm 0.0213}$ | $1359.914_{\pm 334.071}$ | 10/10 |
| SALaD (N-MTLR, $\beta_1 = 0$) | $67.63_{\pm 3.10}$ | $0.1619_{\pm 0.0132}$ | $1410.131_{\pm 301.199}$ | 10/10 |
| SALaD (N-MTLR, $\beta_2 = 0$) | $68.28_{\pm 3.25}$ | $0.1606_{\pm 0.0174}$ | $1387.106_{\pm 340.861}$ | 10/10 |
| SALaD (N-MTLR) | $\underline{69.08}_{\pm 4.95}$ | $\underline{0.1576}_{\pm 0.0156}$ | $\underline{1356.486}_{\pm 330.189}$ | 10/10 |
| 2B (AFTNN-Weibull) | $68.02_{\pm 3.67}$ | $\underline{0.1615}_{\pm 0.0141}$ | $2678.262_{\pm 2376.948}$ | 10/10 |
| SALaD (AFTNN-Weibull, $\beta_1 = 0$) | $66.21_{\pm 5.32}$ | $0.1911_{\pm 0.0239}$ | $2927.924_{\pm 555.786}$ | 9/10 |
| SALaD (AFTNN-Weibull, $\beta_2 = 0$) | $66.47_{\pm 4.56}$ | $0.1776_{\pm 0.0208}$ | $3486.753_{\pm 1411.000}$ | 10/10 |
| SALaD (AFTNN-Weibull) | $\underline{68.51}_{\pm 3.60}$ | $0.1756_{\pm 0.0157}$ | $\underline{2298.406}_{\pm 394.830}$ | $\underline{10/10}$ |
| 2B (AFTNN-LogLogistic) | $68.23_{\pm 3.64}$ | $\underline{0.1677}_{\pm 0.0112}$ | $\underline{1632.557}_{\pm 332.103}$ | 10/10 |
| SALaD (AFTNN-LogLogistic, $\beta_1 = 0$) | $68.06_{\pm 3.14}$ | $0.1850_{\pm 0.0111}$ | $2400.538_{\pm 334.237}$ | $\underline{10/10}$ |
| SALaD (AFTNN-LogLogistic, $\beta_2 = 0$) | $67.43_{\pm 9.55}$ | $0.1862_{\pm 0.0192}$ | $3279.363_{\pm 1764.159}$ | $\underline{10/10}$ |
| SALaD (AFTNN-LogLogistic) | $\underline{68.48}_{\pm 6.03}$ | $0.1748_{\pm 0.0160}$ | $1746.032_{\pm 392.453}$ | 9/10 |

*Table 23.* Ablation on the **METABRIC**, with the same notation as in Table 19.

| Method | CI(%) ↑ | IBS ↓ | MAE ↓ | D-cal ↑ |
|---|---|---|---|---|
| 2B (DeepSurv) | $68.60_{\pm 3.73}$ | $0.1786_{\pm 0.0291}$ | $\underline{3184.870}_{\pm 371.282}$ | 10/10 |
| SALaD (DeepSurv, $\beta_1 = 0$) | $68.00_{\pm 4.21}$ | $0.1802_{\pm 0.0244}$ | $3247.395_{\pm 378.317}$ | 10/10 |
| SALaD (DeepSurv, $\beta_2 = 0$) | $68.87_{\pm 3.63}$ | $0.1821_{\pm 0.0266}$ | $3516.913_{\pm 411.765}$ | 10/10 |
| SALaD (DeepSurv) | $\underline{69.33}_{\pm 3.68}$ | $\underline{0.1782}_{\pm 0.0290}$ | $3226.559_{\pm 382.018}$ | 10/10 |
| 2B (N-MTLR) | $68.39_{\pm 3.93}$ | $0.1802_{\pm 0.0335}$ | $3142.638_{\pm 355.598}$ | 10/10 |
| SALaD (N-MTLR, $\beta_1 = 0$) | $68.84_{\pm 4.08}$ | $0.1806_{\pm 0.0284}$ | $3119.312_{\pm 361.546}$ | 10/10 |
| SALaD (N-MTLR, $\beta_2 = 0$) | $69.10_{\pm 3.60}$ | $0.1826_{\pm 0.0286}$ | $3162.855_{\pm 378.542}$ | 10/10 |
| SALaD (N-MTLR) | $\underline{69.48}_{\pm 2.82}$ | $\underline{0.1796}_{\pm 0.0266}$ | $\underline{3087.063}_{\pm 367.562}$ | 10/10 |
| 2B (AFTNN-Weibull) | $66.93_{\pm 3.57}$ | $\underline{0.2000}_{\pm 0.0259}$ | $\underline{5671.532}_{\pm 1613.259}$ | 10/10 |
| SALaD (AFTNN-Weibull, $\beta_1 = 0$) | $67.96_{\pm 4.80}$ | $0.2118_{\pm 0.0280}$ | $29587.786_{\pm 15624.165}$ | 10/10 |
| SALaD (AFTNN-Weibull, $\beta_2 = 0$) | $68.08_{\pm 4.22}$ | $0.2012_{\pm 0.2280}$ | $10233.876_{\pm 4171.566}$ | 10/10 |
| SALaD (AFTNN-Weibull) | $\underline{69.06}_{\pm 3.73}$ | $0.2127_{\pm 0.2640}$ | $9494.431_{\pm 2143.021}$ | $\underline{10/10}$ |
| 2B (AFTNN-LogLogistic) | $68.72_{\pm 3.81}$ | $\underline{0.1945}_{\pm 0.0193}$ | $6806.676_{\pm 2528.757}$ | 9/10 |
| SALaD (AFTNN-LogLogistic, $\beta_1 = 0$) | $67.74_{\pm 3.67}$ | $0.2150_{\pm 0.0130}$ | $7860.445_{\pm 1092.275}$ | 7/10 |
| SALaD (AFTNN-LogLogistic, $\beta_2 = 0$) | $67.65_{\pm 2.64}$ | $0.2055_{\pm 0.0167}$ | $\underline{5432.907}_{\pm 849.347}$ | 9/10 |
| SALaD (AFTNN-LogLogistic) | $\underline{69.56}_{\pm 3.77}$ | $0.2012_{\pm 0.0199}$ | $7831.367_{\pm 2562.350}$ | 8/10 |

*Table 24.* Ablation on the **NACD** dataset, with the same notation as in Table 19.

| Method | CI(%) ↑ | IBS ↓ | MAE ↓ | D-cal ↑ |
|---|---|---|---|---|
| 2B (DeepSurv) | $75.45_{\pm 1.91}$ | $0.1371_{\pm 0.0112}$ | $21.032_{\pm 1.278}$ | 9/10 |
| SALaD (DeepSurv, $\beta_1 = 0$) | $75.31_{\pm 1.71}$ | $0.1386_{\pm 0.0137}$ | $22.935_{\pm 3.104}$ | 8/10 |
| SALaD (DeepSurv, $\beta_2 = 0$) | $75.01_{\pm 1.84}$ | $0.1391_{\pm 0.0126}$ | $21.512_{\pm 1.349}$ | 10/10 |
| SALaD (DeepSurv) | $75.52_{\pm 2.21}$ | $0.1376_{\pm 0.0096}$ | $21.394_{\pm 1.414}$ | 10/10 |
| 2B (N-MTLR) | $75.56_{\pm 2.06}$ | $0.1379_{\pm 0.0157}$ | $20.998_{\pm 1.321}$ | 10/10 |
| SALaD (N-MTLR, $\beta_1 = 0$) | $75.42_{\pm 2.00}$ | $0.1391_{\pm 0.0118}$ | $21.148_{\pm 1.204}$ | 8/10 |
| SALaD (N-MTLR, $\beta_2 = 0$) | $75.44_{\pm 2.08}$ | $0.1439_{\pm 0.0081}$ | $21.633_{\pm 1.023}$ | 3/10 |
| SALaD (N-MTLR) | $75.57_{\pm 2.13}$ | $0.1356_{\pm 0.0108}$ | $20.693_{\pm 1.151}$ | 8/10 |
| 2B (AFTNN-Weibull) | $75.49_{\pm 2.24}$ | $0.1362_{\pm 0.0123}$ | $22.781_{\pm 1.869}$ | 7/10 |
| SALaD (AFTNN-Weibull, $\beta_1 = 0$) | $75.01_{\pm 3.20}$ | $0.1821_{\pm 0.0185}$ | $48.444_{\pm 9.111}$ | 6/10 |
| SALaD (AFTNN-Weibull, $\beta_2 = 0$) | $75.10_{\pm 1.87}$ | $0.1657_{\pm 0.0109}$ | $41.719_{\pm 4.772}$ | 6/10 |
| SALaD (AFTNN-Weibull) | $76.52_{\pm 1.84}$ | $0.1511_{\pm 0.0118}$ | $24.648_{\pm 2.180}$ | 7/10 |
| 2B (AFTNN-LogLogistic) | $75.25_{\pm 1.86}$ | $0.1373_{\pm 0.0105}$ | $23.580_{\pm 2.910}$ | 10/10 |
| SALaD (AFTNN-LogLogistic, $\beta_1 = 0$) | $76.27_{\pm 1.64}$ | $0.1521_{\pm 0.0093}$ | $25.032_{\pm 2.932}$ | 9/10 |
| SALaD (AFTNN-LogLogistic, $\beta_2 = 0$) | $75.77_{\pm 2.25}$ | $0.1542_{\pm 0.0133}$ | $41.576_{\pm 10.509}$ | 8/10 |
| SALaD (AFTNN-LogLogistic) | $76.86_{\pm 2.06}$ | $0.1432_{\pm 0.0105}$ | $22.208_{\pm 1.234}$ | 10/10 |

*Table 25.* Ablation studies on the **SUPPORT** dataset, with the same notation as in Table 19.

| Method | CI(%) ↑ | IBS ↓ | MAE ↓ | D-cal ↑ |
|---|---|---|---|---|
| 2B (DeepSurv) | $68.97_{\pm 0.76}$ | $0.1618_{\pm 0.0077}$ | $668.349_{\pm 17.621}$ | 7/10 |
| SALaD (DeepSurv, $\beta_1 = 0$) | $68.85_{\pm 0.90}$ | $0.1622_{\pm 0.0085}$ | $706.122_{\pm 22.998}$ | 8/10 |
| SALaD (DeepSurv, $\beta_2 = 0$) | $68.48_{\pm 0.87}$ | $0.1637_{\pm 0.0089}$ | $712.127_{\pm 30.945}$ | 10/10 |
| SALaD (DeepSurv) | $68.90_{\pm 0.50}$ | $0.1609_{\pm 0.0070}$ | $685.269_{\pm 15.232}$ | 9/10 |
| 2B (N-MTLR) | $69.17_{\pm 0.74}$ | $0.1608_{\pm 0.0067}$ | $675.020_{\pm 10.118}$ | 10/10 |
| SALaD (N-MTLR, $\beta_1 = 0$) | $68.87_{\pm 0.78}$ | $0.1619_{\pm 0.0067}$ | $672.267_{\pm 11.024}$ | 8/10 |
| SALaD (N-MTLR, $\beta_2 = 0$) | $68.78_{\pm 0.89}$ | $0.1611_{\pm 0.0071}$ | $673.470_{\pm 15.994}$ | 10/10 |
| SALaD (N-MTLR) | $69.23_{\pm 0.80}$ | $0.1592_{\pm 0.0083}$ | $667.836_{\pm 15.878}$ | 9/10 |
| 2B (AFTNN-Weibull) | $68.85_{\pm 0.75}$ | $0.1598_{\pm 0.0086}$ | $672.090_{\pm 17.784}$ | 0/10 |
| SALaD (AFTNN-Weibull, $\beta_1 = 0$) | $68.82_{\pm 0.79}$ | $0.1659_{\pm 0.0097}$ | $786.899_{\pm 46.403}$ | 0/10 |
| SALaD (AFTNN-Weibull, $\beta_2 = 0$) | $68.95_{\pm 0.62}$ | $0.1770_{\pm 0.0090}$ | $909.054_{\pm 51.187}$ | 0/10 |
| SALaD (AFTNN-Weibull) | $69.32_{\pm 1.02}$ | $0.1638_{\pm 0.0085}$ | $707.741_{\pm 30.946}$ | 0/10 |
| 2B (AFTNN-LogLogistic) | $69.42_{\pm 0.75}$ | $0.1627_{\pm 0.0087}$ | $678.962_{\pm 21.778}$ | 3/10 |
| SALaD (AFTNN-LogLogistic, $\beta_1 = 0$) | $69.40_{\pm 1.00}$ | $0.1661_{\pm 0.0083}$ | $674.994_{\pm 16.259}$ | 1/10 |
| SALaD (AFTNN-LogLogistic, $\beta_2 = 0$) | $69.17_{\pm 0.80}$ | $0.1674_{\pm 0.0098}$ | $757.821_{\pm 91.367}$ | 2/10 |
| SALaD (AFTNN-LogLogistic) | $69.88_{\pm 0.83}$ | $0.1651_{\pm 0.0075}$ | $670.408_{\pm 17.918}$ | 1/10 |

*Table 26.* Ablation on the **MIMIC-IV** dataset, with the same notation as in Table 19.

| Method | CI(%) ↑ | IBS ↓ | MAE ↓ | D-cal ↑ |
|---|---|---|---|---|
| 2B (DeepSurv) | $75.33_{\pm 0.85}$ | $0.0387_{\pm 0.0031}$ | $268.354_{\pm 15.772}$ | $\underline{8/10}$ |
| SALaD (DeepSurv, $\beta_1 = 0$) | $74.96_{\pm 0.89}$ | $0.0394_{\pm 0.0033}$ | $270.794_{\pm 16.482}$ | $\overline{7/10}$ |
| SALaD (DeepSurv, $\beta_2 = 0$) | $75.09_{\pm 0.98}$ | $0.0386_{\pm 0.0030}$ | $\underline{262.976_{\pm 16.701}}$ | $8/10$ |
| SALaD (DeepSurv) | $\underline{75.60_{\pm 1.10}}$ | $\underline{0.0385_{\pm 0.0032}}$ | $263.202_{\pm 15.941}$ | $8/10$ |
| 2B (N-MTLR) | $75.77_{\pm 1.01}$ | $\underline{0.0377_{\pm 0.0066}}$ | $261.624_{\pm 15.838}$ | $10/10$ |
| SALaD (N-MTLR, $\beta_1 = 0$) | $75.89_{\pm 1.02}$ | $0.0388_{\pm 0.0034}$ | $263.109_{\pm 16.614}$ | $10/10$ |
| SALaD (N-MTLR, $\beta_2 = 0$) | $75.96_{\pm 1.29}$ | $0.0391_{\pm 0.0035}$ | $264.116_{\pm 16.929}$ | $10/10$ |
| SALaD (N-MTLR) | $\underline{76.18_{\pm 1.09}}$ | $0.0387_{\pm 0.0032}$ | $\underline{261.945_{\pm 15.249}}$ | $10/10$ |
| 2B (AFTNN-Weibull) | $74.74_{\pm 0.86}$ | $\underline{0.0385_{\pm 0.0069}}$ | $270.636_{\pm 16.305}$ | $0/10$ |
| SALaD (AFTNN-Weibull, $\beta_1 = 0$) | $76.07_{\pm 1.11}$ | $0.0391_{\pm 0.0033}$ | $272.307_{\pm 16.276}$ | $0/10$ |
| SALaD (AFTNN-Weibull, $\beta_2 = 0$) | $76.21_{\pm 0.90}$ | $0.0433_{\pm 0.0026}$ | $300.467_{\pm 14.128}$ | $0/10$ |
| SALaD (AFTNN-Weibull) | $\underline{76.24_{\pm 1.09}}$ | $0.0389_{\pm 0.0032}$ | $\underline{269.539_{\pm 14.643}}$ | $0/10$ |
| 2B (AFTNN-LogLogistic) | $74.98_{\pm 0.91}$ | $0.0396_{\pm 0.0032}$ | $266.799_{\pm 15.816}$ | $9/10$ |
| SALaD (AFTNN-LogLogistic, $\beta_1 = 0$) | $76.71_{\pm 1.26}$ | $\underline{0.0393_{\pm 0.0031}}$ | $265.330_{\pm 15.633}$ | $9/10$ |
| SALaD (AFTNN-LogLogistic, $\beta_2 = 0$) | $76.70_{\pm 1.03}$ | $0.0396_{\pm 0.0032}$ | $265.453_{\pm 15.672}$ | $8/10$ |
| SALaD (AFTNN-LogLogistic) | $\underline{76.91_{\pm 1.06}}$ | $\underline{0.0393_{\pm 0.0030}}$ | $\underline{264.299_{\pm 14.981}}$ | $\underline{10/10}$ |

*Table 27.* Comparing the Adversarial training strategy with the proposed IPM regularizer.

| Dataset | Method | CI(%) ↑ | IBS ↓ | MAE ↓ | D-cal ↑ |
|---|---|---|---|---|---|
| **HFCR** | SALaD (N-MTLR, adv) | $69.09_{\pm 9.11}$ | $0.1528_{\pm 0.0370}$ | $315.768_{\pm 87.591}$ | $10/10$ |
|  | SALaD (N-MTLR) | $\underline{70.70_{\pm 8.61}}$ | $\underline{0.1453_{\pm 0.0220}}$ | $\underline{231.878_{\pm 23.480}}$ | $10/10$ |
| **PBC** | SALaD (N-MTLR, adv) | $79.10_{\pm 5.00}$ | $0.1699_{\pm 0.0470}$ | $2627.063_{\pm 770.624}$ | $10/10$ |
|  | SALaD (N-MTLR) | $\underline{81.52_{\pm 5.78}}$ | $\underline{0.1449_{\pm 0.0378}}$ | $\underline{2529.522_{\pm 828.327}}$ | $10/10$ |
| **GBM** | SALaD (N-MTLR, adv) | $67.07_{\pm 2.89}$ | $0.0713_{\pm 0.0191}$ | $382.465_{\pm 59.675}$ | $9/10$ |
|  | SALaD (N-MTLR) | $\underline{68.56_{\pm 2.13}}$ | $\underline{0.0689_{\pm 0.0176}}$ | $\underline{377.271_{\pm 50.899}}$ | $\underline{10/10}$ |
| **GBSG** | SALaD (N-MTLR, adv) | $68.11_{\pm 3.34}$ | $\underline{0.1565_{\pm 0.0113}}$ | $1381.757_{\pm 326.433}$ | $10/10$ |
|  | SALaD (N-MTLR) | $\underline{69.08_{\pm 4.95}}$ | $0.1576_{\pm 0.0156}$ | $\underline{1356.486_{\pm 330.189}}$ | $10/10$ |
| **METABRIC** | SALaD (N-MTLR, adv) | $68.89_{\pm 3.26}$ | $\underline{0.1778_{\pm 0.0258}}$ | $3116.642_{\pm 360.829}$ | $10/10$ |
|  | SALaD (N-MTLR) | $\underline{69.48_{\pm 2.82}}$ | $0.1796_{\pm 0.0266}$ | $\underline{3087.063_{\pm 367.562}}$ | $10/10$ |
| **NACD** | SALaD (N-MTLR, adv) | $74.99_{\pm 1.58}$ | $0.1444_{\pm 0.0165}$ | $21.740_{\pm 1.268}$ | $\underline{10/10}$ |
|  | SALaD (N-MTLR) | $\underline{75.57_{\pm 2.13}}$ | $\underline{0.1356_{\pm 0.0108}}$ | $\underline{20.693_{\pm 1.151}}$ | $8/10$ |
| **SUPPORT** | SALaD (N-MTLR, adv) | $69.08_{\pm 0.97}$ | $0.1611_{\pm 0.0090}$ | $670.528_{\pm 17.120}$ | $9/10$ |
|  | SALaD (N-MTLR) | $\underline{69.23_{\pm 0.80}}$ | $\underline{0.1592_{\pm 0.0083}}$ | $\underline{667.836_{\pm 15.878}}$ | $9/10$ |
| **MIMIC-IV** | SALaD (N-MTLR, adv) | $75.43_{\pm 0.91}$ | $\underline{0.0377_{\pm 0.0033}}$ | $263.965_{\pm 15.156}$ | $7/10$ |
|  | SALaD (N-MTLR) | $\underline{76.18_{\pm 1.09}}$ | $0.0387_{\pm 0.0032}$ | $\underline{261.945_{\pm 15.249}}$ | $\underline{10/10}$ |

*Table 28.* Computation analysis for the benchmarks. The training time and inference time are averaged over 10 runs on the **SUPPORT** dataset. The number of active parameters is the number of parameters that are involved in the inference stage.

| Method | Training Time (seconds) | Inference Time (seconds) | # Active Parameters | # FLOPs |
|---|---|---|---|---|
| Nnet-survival | $4.098_{\pm 0.498}$ | $0.0012_{\pm 0.0003}$ | 10618 | 10682 |
| RSF | $36.671_{\pm 0.268}$ | $0.2395_{\pm 0.0092}$ | 1330360 | - |
| GB | $53.872_{\pm 0.346}$ | $0.0239_{\pm 0.0016}$ | - | - |
| DeepHit | $5.345_{\pm 0.658}$ | $0.0010_{\pm 0.0001}$ | 10618 | 10682 |
| CoxTime | $3.5502_{\pm 0.730}$ | $1.2954_{\pm 0.1739}$ | 6592 | - |
| IWSG | $623.551_{\pm 238.967}$ | $0.0004_{\pm 0.0000}$ | 10618 | 10682 |
| SODEN $^\dagger$ | $1472.272_{\pm 312.793}$ | $0.2337_{\pm 0.0097}$ | 297 | 18848 |
| CQRNN | $17.451_{\pm 6.344}$ | $0.0002_{\pm 0.0000}$ | 7049 | 7168 |
| DCSurvival | $292.743_{\pm 11.933}$ | $0.4230_{\pm 0.0057}$ | 10368 | 16579756 |
| SurvivalBoost | $20.138_{\pm 2.122}$ | $0.2374_{\pm 0.0217}$ | - | - |
| DeepSurv | $8.630_{\pm 1.711}$ | $0.0104_{\pm 0.0007}$ | 6529 | 6656 |
| 2B(DeepSurv) | $18.921_{\pm 4.0634}$ | $0.0110_{\pm 0.0004}$ | 6594 | 6720 |
| SALaD(DeepSurv) | $50.328_{\pm 25.923}$ | $0.0110_{\pm 0.0004}$ | 13057 | 13312 |
| N-MTLR | $11.842_{\pm 2.062}$ | $0.0004_{\pm 0.0001}$ | 6464 | 6592 |
| 2B(N-MTLR) | $20.558_{\pm 3.532}$ | $0.0004_{\pm 0.0000}$ | 6464 | 6592 |
| SALaD(N-MTLR) | $37.802_{\pm 4.927}$ | $0.0006_{\pm 0.0001}$ | 12928 | 13184 |
| AFTNN-Weibull | $21.464_{\pm 3.649}$ | $0.0005_{\pm 0.0005}$ | 6529 | 6656 |
| 2B(AFTNN-Weibull) | $42.671_{\pm 10.656}$ | $0.0006_{\pm 0.0006}$ | 6594 | 6720 |
| SALaD(AFTNN-Weibull) | $105.501_{\pm 5.327}$ | $0.0006_{\pm 0.0001}$ | 13057 | 13312 |
| AFTNN-LogLogistic | $21.340_{\pm 3.660}$ | $0.0005_{\pm 0.0006}$ | 6529 | 6656 |
| 2B(AFTNN-LogLogistic) | $61.545_{\pm 22.574}$ | $0.0005_{\pm 0.0006}$ | 6594 | 6720 |
| SALaD(AFTNN-LogLogistic) | $98.021_{\pm 11.269}$ | $0.0006_{\pm 0.0001}$ | 13057 | 13312 |

