# OpenReview forum: "Explicitly Modeling Censoring Produces Superior Survival Predictors"
_ICML.cc/2026/Conference — ICML 2026 regular_

### Official Review · Reviewer_FciL · 2026-03-02

**Soundness:** 1
**Presentation:** 2
**Significance:** 1
**Originality:** 1
**Overall Recommendation:** 3
**Confidence:** 5

**Summary:**

This paper argues that most survival models, which ignore censoring-related terms in the likelihood under the independent censoring assumption, may lose useful information when the event and censoring processes share parameters. The authors show that discarding censoring contributions is justified only under a stronger non-informative censoring condition, and that jointly modeling both processes improves statistical efficiency. They first introduce a simple two-branch (2B) framework that maximizes the full likelihood and theoretically demonstrate reduced asymptotic variance when censoring is informative. Building on this, they propose SALaD (Survival Analysis via Latent Decomposed Representation), which learns disentangled latent representations separating event-specific, censoring-specific, and shared confounding factors, enforced through likelihood terms and regularization strategies. Experiments on two semi-synthetic and eight real-world datasets show that SALaD consistently improves discrimination, calibration, and error metrics over standard deep survival models, their 2B extensions, and multiple state-of-the-art methods, while incurring moderate computational overhead.

**Compliance With Llm Reviewing Policy:**

Affirmed.

**Key Questions For Authors:**

The framework assumes $E \perp C \mid X$. How does the method behave if this assumption is violated, i.e., if $E$ and $C$ remain dependent even after conditioning on $X$?

Can the authors clarify whether their notion of “informative censoring” (shared parameters between $E \mid X$ and $C \mid X$) is intended to cover structural dependence between $E$ and $C$, or only parameter coupling under conditional independence?

In practice, how can one diagnose whether the relevant issue is parameter sharing versus genuine dependence in the joint distribution of $(E, C)$?

Why is the joint survival function $P(T > t, U > u)$ not explicitly modeled? Would the framework extend to parameterizing a joint model such as
$P(T > t, U > u) = C(S_T(t), S_U(u))$
for some copula $C(\cdot,\cdot)$?

How does SALaD compare theoretically to models that directly parameterize the joint distribution of $(E, C)$ rather than assuming factorization?

If censoring is truly dependent (not just parameter-coupled), would maximizing the full factorized likelihood still be consistent?

The invariance penalties rely on marginal survival curves estimated via Kaplan–Meier. Since KM is consistent only under independent censoring, what happens if censoring depends on covariates?

Lastly, have the authors studied the dependent censoring line of work? Namely, the following and references therein:

Gharari et al. (2023). Copula-Based Estimation with Dependent Censoring. UAI
Zhant et al. (2024). Deep Copula-Based Survival Analysis for Dependent Censoring with Identifiability Guarantees. AAAI.

**Limitations:**

The paper did not explicitly discuss the limitations.

**Strengths And Weaknesses:**

Strengths:

1. The $2B$ setup (shared encoder, two heads for $E\mid X$ and $C\mid X$) is easy to implement on top of common deep survival models and gives a concrete, testable alternative to event-only training.


Weaknesses:

1. The paper assumes conditional independence:
$E \perp C \mid X$ and argues that modeling censoring improves efficiency when parameters are shared between the event and censoring models. However, the dependent-censoring literature addresses a different issue: genuine statistical dependence between event and censoring times, even after conditioning on covariates. In such settings, the standard factorized likelihood is not merely inefficient but biased. SALaD does not model the joint dependence structure between $E$ and $C$. Therefore, it does not address the primary bias problem studied in that line of work.

2. The paper defines informative censoring as a violation of a “non-informative” assumption due to shared parameters between the event and censoring models. In contrast, the copula framework defines informative censoring through dependence in the joint survival distribution. These are distinct concepts: SALaD: informative $=$ parameter coupling under conditional independence. Copula line: informative $=$ structural dependence in the joint distribution.
By not formally distinguishing these notions, the paper narrows the meaning of “modeling censoring” and omits the broader dependent-censoring framework.

3. SALaD proposes decomposing covariates into latent factors:

&nbsp;&nbsp;&nbsp;&nbsp;&nbsp;&nbsp; $H^\epsilon$: event-specific

&nbsp;&nbsp;&nbsp;&nbsp;&nbsp;&nbsp; $H^\gamma$: censoring-specific

&nbsp;&nbsp;&nbsp;&nbsp;&nbsp;&nbsp; $H^\kappa$: shared confounders

&nbsp;&nbsp;&nbsp;&nbsp;&nbsp;&nbsp; $H^\varnothing$: irrelevant components

However, it does not provide identifiability conditions for recovering this decomposition from right-censored data. The separation is enforced via regularization penalties rather than derived from statistical identification arguments.

In contrast, dependent-censoring models typically specify an explicit joint structure and discuss identifiability under stated assumptions. The absence of comparable analysis weakens the theoretical grounding of the latent representation claims.

4. The invariance regularizers rely on early/late group splits derived from marginal survival curves estimated using Kaplan–Meier.

Yet Kaplan–Meier is only consistent under independent censoring. When censoring depends on features or unobserved factors, the marginal survival estimator can be biased. If the grouping mechanism itself is biased, then the supervision signal used to enforce representation invariance is also distorted.

This creates a circular dependency: the method relies on independent-censoring behavior in order to correct censoring-related representation issues.

5. The paper presents explicit modeling of censoring in deep survival models as relatively under-explored. However, dependent-censoring literature already:

&nbsp;&nbsp;&nbsp;&nbsp;&nbsp;&nbsp; Optimizes likelihoods that include censoring contributions

&nbsp;&nbsp;&nbsp;&nbsp;&nbsp;&nbsp; Models joint dependence explicitly

&nbsp;&nbsp;&nbsp;&nbsp;&nbsp;&nbsp; Provides identification and asymptotic arguments

Failure to engage with this line of work weakens the novelty positioning and risks overstating the scope of the contribution.

---

> ### Author Rebuttal · Authors · 2026-03-31
>
> We thank the reviewer for highlighting dependent censoring. We first clarify the key distinction:
>
> **This paper studies informative censoring under conditional independence, not dependent censoring.**
>
> We assume $E\perp C|X$, and study the case where the event and censoring models may still share parameters (informative), so dropping censoring terms in likelihood can be inefficient even though the factorization remains valid. This differs from dependent censoring, where E and C remain dependent given X.
>
> ---
> We now address weaknesses [W] and questions [Q]:
> > [W1, Q4, Q5, Q6] not address dep.censoring; P(E, C) not modeled; SALaD vs. Copula models in dep.censoring
>
> As stated above, this work does **not** attempt to address dep.censoring. Under $E\perp C|X$, factorizing $P(E,C|X)=P(E|X)P(C|X)$ yields a more tractable problem.
>
> Under dep.censoring, SALaD is not theoretically valid as it is based on factorized likelihood, which does not apply here. Note we do not claim that SALaD applies to that setting.
> > [W2, Q2] Unclear notions for dependent and informative
>
> We respectfully disagree. Section 2 already makes this distinction explicit: the likelihood derivation (lines 68-133) and illustrative example (lines 138-151) show that independent and non-informative censoring are orthogonal. Appendix A2 further reinforces this point with additional notation and examples.
>
> We recognize that some papers conflate these concepts. However, we did not introduce this distinction; it appears in many survival textbooks — see Appendix A2.2. We also disagree with the assertion that “copula framework defines informative censoring differently” — a copula survival textbook ([Emura & Chen](https://link.springer.com/book/10.1007/978-981-10-7164-5) pp.20-22) uses the same notion.
> > [W3] No identifiability for recovering this decomposition
>
> We agree. Our method enforces this structure through regularization, not through a statistical identification argument. We already note this as a limitation (lines 191-192).
> > [W4, Q7] KM bias
>
> We agree that KM can be biased when C depends on X. We discuss this explicitly in Remark 4.5 and Appendix D.4, where we argue that this limitation has minimal practical impact because KM is used only to form early/late groups. In practice, the bias mainly changes the group proportions (eg, 40/60 instead of 50/50), which should have little effect with a sufficiently large sample.
> > [W5, Q8] Copula works already considers 2B structure
>
> Copula methods coupled the 2B structure with an explicit copula-based optimization. We wanted to disentangle whether the performance gain comes from the copula itself or the 2B structure. We therefore conducted a control analysis by removing only the copula while keeping 2B. Empirically, this did not reduce performance, motivating us to explore why the 2B structure is effective in its own right. This in turn led to SALaD.
>
> Following the reviewer’s suggestion, we benchmarked DCSurvival. Across the 9 real datasets (8 in the paper + BRCA-miRNA suggested by Reviewer d89c [Q2, Q3]), DCSurvival returned results on only 3 datasets due to convergence issue:
>
>  DCSurvival | CI(%) | IBS | MAE-PO | D-cal
> ---|---|---|---|---
>  HFCR | 61.56$\pm$8.89 | 0.1770$\pm$0.0363 | 254.350$\pm$12.895 | 10/10
>  NACD | 74.84$\pm$1.96 | 0.1479$\pm$0.0220 | 21.117$\pm$1.293 | 5/10
>  SUPPORT | 68.83$\pm$1.04 | 0.1550$\pm$0.0090 | 680.806$\pm$15.146 | 0/10
>
> Compared with Tables 11, 16, and 17, SALaD outperforms DCSurvival in all cases except IBS on SUPPORT.
> > [Q1] Dependent censoring experiment
>
> We added a frailty-based dep.censoring setting by randomly masking half of the features in semi-SUPPORT; the masked variables serve as unobserved confounders. Due to space limits, we report only the mean and omit D-cal (all are 0/10 calibrated). SALaD has comparable results with SOTAs.
>
>  Method | CI(%) | IBS | MAE
> ---|---|---|---
>  Nnet-survival | 55.76 | 0.0017 | 4.614
>  RSF | 55.78 | 0.0545 | 4.281
>  GB | 55.28 | 0.0022 | 4.011
>  DeepHit | 55.48 | 0.0021 | 4.311
>  CoxTime | 55.01 | 0.0022 | 4.017
>  IWSG | 55.37 | 0.0034 | 4.020
>  SODEN | 52.78 | 0.1542 | 108.259
>  CQRNN | 53.29 | 0.0017 | 6.455
>  DCSurvival | 55.44 | 0.0017 | 4.201
>  SurvivalBoost | 52.62 | 0.0026 | 4.201
>  **DeepSurv** |  |  |  |
>  base | 55.57 | 0.0024 | 3.998
>  2B | 55.72 | 0.0025 | 3.974
>  SALaD | 55.76 | 0.0026 | 3.973
>  **N-MTLR** |  |  |  |
>  base | 55.69 | 0.0025 | 3.978
>  2B | 55.93 | 0.0023 | 3.961
>  SALaD | 55.76 | 0.0024 | 3.959
>  **AFTNN-Weibull** |  |  |  |
>  base | 55.02 | 0.0017 | 4.364
>  2B | 55.29 | 0.0017 | 4.332
>  SALaD | 55.25 | 0.0017 | 4.307
>  **AFTNN-LogLogistic** |  |  |  |
>  base | 55.71 | 0.0018 | 4.146
>  2B | 55.79 | 0.0018 | 4.124
>  SALaD | 55.77 | 0.0018 | 4.212
> > [Q3] Diagnose parameter sharing vs. dependence of E,C?
>
> In general, neither issue can be identified from observed data (Tsiatis, 1975). Our goal is not to diagnose whether parameter sharing is present, but to show that SALaD is robust across both informative and non-informative censoring.

---

> > ### Author Rebuttal · Reviewer_FciL · 2026-04-01
> >
> > Rebuttal acknowledged and score raised.

---

> > > ### Author Response · Authors · 2026-04-07
> > >
> > > Thank you for confirming that our rebuttal **fully addressed** your concerns. We appreciate your time and your decision to raise the score.
> > >
> > > The revised version will include (1) the additional experiments mentioned in our rebuttal here related to DCSurvival, and (2) a discussion/comparison about the line of the copula works (Gharari et al. 2023 and Zhang et al. 2024) in the related work.

---

### Official Review · Reviewer_Rjx1 · 2026-03-11

**Soundness:** 2
**Presentation:** 3
**Significance:** 3
**Originality:** 2
**Overall Recommendation:** 4
**Confidence:** 3

**Summary:**

This paper argues that the common simplification of discarding censoring-related terms in survival models is justified only when the event and censoring distributions are non-informative, i.e., they do not share parameters. When this condition is violated, ignoring censoring-related terms may discard information useful for identifying and efficiently estimating the event-time distribution. The authors propose a framework in which the covariates X are re-expressed into four latent groups: those affecting only the event process, only the censoring process, both processes, or neither. Based on this structure, they derive the corresponding density or survival functions and the likelihood for right-censored outcomes. This paper introduces a simple extension (2B) of standard survival models and Survival Analysis via Latent Decomposed representation (SALaD), and shows empirically that SALaD outperforms strong baselines, including 2B and multiple SOTA models, across 2 semi-synthetic and 8 real-world datasets.

**Compliance With Llm Reviewing Policy:**

Affirmed.

**Final Justification:**

I maintain my original score.

**Key Questions For Authors:**

1.The paper suggests that incorporating censoring-related information can improve likelihood-based estimation. However, it is unclear whether adding irrelevant variables could also increase the likelihood. This situation is somewhat analogous to linear regression, where adding additional covariates may mechanically increase R^2 even if the variables are not truly informative. It would therefore be helpful to analyze this issue both theoretically and through additional experiments on the semi-synthetic datasets, and to examine whether the method can reliably distinguish different types of variables.

2.In the semi-synthetic datasets, it would be useful to additionally consider settings where the event and censoring processes are independent, and evaluate whether the proposed method still maintains advantages over existing approaches.

3.The paper would benefit from reporting the computational cost of the method, including runtime, especially when compared with the baseline models.

4.When explaining the motivation and novelty of the method, the paper presents Figure 2 (Overview of survival-model architectures). Since IPCW plays an important transitional role in this framework, it would be helpful if the authors could provide a more detailed description of IPCW in the paper (at least in the appendix).

**Limitations:**

The evaluation mainly focuses on prediction performance for event time, while the interpretability aspects related to the covariate categorization are not explored in detail.

**Strengths And Weaknesses:**

Soundness:This paper argues that the common simplification of discarding censoring-related terms in survival models is justified only when the event and censoring distributions are non-informative, i.e., they do not share parameters. When this condition is violated, ignoring censoring-related terms may discard information useful for identifying and efficiently estimating the event-time distribution. The authors propose a framework in which the covariates X are re-expressed into four latent groups: those affecting only the event process, only the censoring process, both processes, or neither. The paper provides a thorough analysis of the underlying assumptions and presents a clear overview of existing methods, highlighting the differences between the proposed approach and prior work. However, the experimental design could benefit from considering a broader range of scenarios. In addition, while the authors evaluate the strengths of their approach, the paper does not sufficiently discuss or empirically examine the potential weaknesses of the proposed method.

Presentation: Overall, the paper is reasonably easy to follow, and the authors clearly discuss how their work differs from prior and concurrent literature. However, some details would benefit from further clarification in the paper, including additional explanation of the proposed variable grouping, discussion of the role of irrelevant variables, and a more detailed description of the IPCW method.

Significance: The paper addresses an important problem in survival analysis, namely how censoring information is treated in likelihood-based survival models. Since the assumption of non-informative censoring is commonly adopted but may be violated in practice, revisiting this assumption and studying the interaction between event and censoring processes is relevant for both methodological development and practical applications. The potential impact of the work is somewhat specialized to survival analysis and time-to-event modeling, but this scope is appropriate given the methodological focus of the paper. Even if the improvements are modest or problem-specific, the framework may encourage future research.

Originality: This paper introduces a simple extension (2B) of standard survival models and Survival Analysis via Latent Decomposed representation (SALaD), and shows empirically that SALaD outperforms strong baselines, including 2B and multiple SOTA models, across 2 semi-synthetic and 8 real-world datasets. The motivation is to to advancing understanding by providing a shared-parameter perspective that explicitly characterizes how covariates may affect the event and censoring processes. However, this perspective is not fully reflected in the empirical analysis. In particular, the paper does not further investigate the proposed variable grouping or provide deeper insights into how different types of variables influence the two processes. In the data analysis, the evaluation mainly focuses on prediction performance for event time, while the interpretability aspects related to the covariate categorization are not explored in detail.

---

> ### Author Rebuttal · Authors · 2026-03-31
>
> We thank the reviewer for the comments. We address weaknesses [W] and questions [Q]:
> > [W1] Broader range of scenarios + examine the weaknesses
>
> Thanks for this suggestion. The revision will expand the empirical study to include:
>
> (1) noisy covariates,
>
> (2) marginally independent censoring,
>
> (3) a high-dimensional, extremely censored dataset, and
>
> (4) dependent censoring.
>
> For (1) and (2), see our replies below. For (3), see our response to Reviewer d89C [Q2, Q3], and for (4), see our response to Reviewer FciL [Q1].
>
> > [W2, W4, Q4] Details clarification (variable grouping, irrelevant variables, add IPCW description)
>
> We apologize for the confusion. SALaD decomposes the **latent representation**, not the raw covariates, so we do not expect each observed covariate to map uniquely to a single component.
>
> We agree with you and Reviewer d89C that our wording may have suggested otherwise — e.g., the cancer example (lines 74–79) was only illustrative. We will revise the manuscript to make this explicit and add a brief description of IPCW.
>
> > [Q1] Add noise covariates in semi-synthetic data and exam likelihood
>
> We thank the reviewer for this suggestion. To test whether irrelevant covariates can mechanically improve likelihood, we ran a noise-covariate experiment on semi-SUPPORT with the DeepSurv backbone. We augmented the original covariates with 50% Gaussian and 50% Bernoulli noise, at 0%, 25%, 50%, and 100% of the original feature dimension.
>
> Table below reports the change in mean test negative log-likelihood (NLL) over 10 runs (omit std due to space limit). A small amount of noise (+25%) slightly improves test NLL for all, but performance worsens at +50% and +100%. Thus, irrelevant covariates do not systematically improve held-out likelihood.
>
>  Method | No Noise | +25% Noise | +50% Noise | +100% Noise
> ---|---|---|---|---
>  base | 54.719 | -0.02% | +0.37% | +0.37%
>  2B | 54.697 | -0.05% | +0.21% | +0.20%
>  SALaD | 54.675 | -0.25% | +0.28% | +0.35%
>
> We also evaluated predictive performance under the same four settings. Overall, SALaD remains more stable than base and 2B in CI, IBS, and D-cal.
>
> **No noise**
>  Method | CI(%) | IBS | MAE | D-cal
> ---|---|---|---|---
>  base | 71.31 | 0.0065 | 2.941 | 0/10
>  2B | 71.30 | 0.0066 | 2.951 | 0/10
>  SALaD | 73.59 | 0.0063 | 2.997 | 1/10
>
> **+25% noise**
>  Method | CI(%) | IBS | MAE | D-cal
> ---|---|---|---|---
>  base | 71.10 | 0.0063 | 2.986 | 0/10
>  2B | 71.10 | 0.0066 | 2.940 | 0/10
>  SALaD | 74.07 | 0.0066 | 2.996 | 1/10
>
> **+50% noise**
>  Method | CI(%) | IBS | MAE | D-cal
> ---|---|---|---|---
>  base | 71.17 | 0.0067 | 2.973 | 0/10
>  2B | 71.27 | 0.0067 | 2.971 | 0/10
>  SALaD | 72.56 | 0.0066 | 3.088 | 3/10
>
> **+100%**
>  Method | CI(%) | IBS | MAE | D-cal
> ---|---|---|---|---
>  base | 70.86 | 0.0066 | 2.992 | 0/10
>  2B | 70.89 | 0.0066 | 2.973 | 0/10
>  SALaD | 71.88 | 0.0065 | 3.081 | 1/10
>
> > [Q2] Add independent experiments
>
> Thank you for the suggestion. We added a new synthetic dataset with 1000 instances and 10 features, drawn iid from truncated normal distributions on [-1.5, 1.5] with mean 0 and std 0.5. The event time is sampled from a two-component LogNormal mixture,
> $$E \mid X \sim 0.5 LogNormal(f(X), \sigma(X)) + 0.5 LogNormal(g(X), \sigma(X)),$$
> where
>    $$f(X) = (X_1 − 1)^2 + X_2 \quad g(X) = 0.5 * I(X_3 \le −0.5) \sqrt{X_4 + 0.5} \quad \sigma(X) = \max(0, \sqrt{0.25|X_5|}).$$
> Only the first five features determine the event distribution; the remaining five are noise. The censoring time is sampled from $0.5 U(0, \max(E)) + 0.1$, and observed time and event indicator are defined in the standard way. Thus, the dataset satisfies both:
>
>  (1) marginal independence between E and C, and
>
>  (2) non-informative censoring, since the event and censoring mechanisms do not share parameters.
>
> The table below reports means over 10 runs. In this deliberately simple setting, base, 2B, and SALaD perform similarly overall. SALaD is slightly better for DeepSurv and N-MTLR, but not consistently stronger for the two parametric models. This is expected in a simple independent, non-informative setting.
>
>  Method | CI(%) | IBS | MAE | D-cal
> ---|---|---|---|---
>  **DeepSurv** |  |  |  |  |
>  base | 63.58 | 0.0471 | 1.970 | 5/10
>  2B | 62.89 | 0.0479 | 1.958 | 7/10
>  SALaD | 63.41 | 0.0464 | 1.944 | 7/10
>  **N-MTLR** |  |  |  |  |
>  base | 62.94 | 0.0453 | 1.960 | 8/10
>  2B | 63.30 | 0.0458 | 1.948 | 7/10
>  SALaD | 63.92 | 0.0451 | 1.934 | 7/10
>  **AFTNN-Weibull** |  |  |  |  |
>  base | 63.97 | 0.0465 | 2.037 | 10/10
>  2B | 63.97 | 0.0467 | 2.022 | 10/10
>  SALaD | 63.78 | 0.0488 | 2.125 | 10/10
>  **AFTNN-LogLogistic** |  |  |  |  |
>  base | 63.44 | 0.0498 | 1.961 | 7/10
>  2B | 63.58 | 0.0496 | 1.958 | 9/10
>  SALaD | 63.41 | 0.0491 | 1.985 | 7/10
>
> > [Q3] Computational cost
>
> The paper already includes a complexity analysis. The main text reports the key findings in lines 417-417. The full setup and detailed results (training time, inference time, active parameters, and FLOPs) are provided in Appendix F.5 and Table 28.

---

> > ### Author Rebuttal · Reviewer_Rjx1 · 2026-04-03
> >
> > I acknowledge the authors’ rebuttal. However, my main concern remains only partially addressed. There lacks theoretical justification. The additional results are all based on 10 runs (consistent with the main paper), which makes it difficult to rule out randomness. Given that my initial score was already positive, I will keep my score unchanged.

---

> > > ### Author Response · Authors · 2026-04-07
> > >
> > > Thanks for your continued support of our work. We sincerely appreciate your time and thoughtful evaluation. Regarding the remaining concerns:
> > >
> > >
> > > ---
> > >
> > > ### Theoretical justification for whether adding irrelevant variables could also increase the likelihood
> > >
> > > We apologize for not making this point sufficiently clear in the initial rebuttal. As noted in that response, SALaD decomposes the latent representation, rather than the raw observed variables, so we do not expect each observed variable to map uniquely to a single component.
> > >
> > > More generally, adding irrelevant variables can indeed improve resubstitution performance, that is, apparent fit measured on the same data used for training. For example, it may increase training likelihood or improve resubstitution $R^2$, simply because the model has more flexibility to fit noise. However, this does not imply a genuine improvement in predictive performance. What matters is the true performance, as estimated on held-out data.
> > >
> > > In this respect, SALaD should not behave differently from other models in its handling of noisy features. This is supported by the additional experiments in the rebuttal: similar to both the base model and the 2B extension, adding a small amount of noise (+25%) yields a slight improvement in test NLL for SALaD, whereas performance deteriorates at +50% and +100%. Therefore, irrelevant covariates do not systematically improve held-out likelihood.
> > >
> > >
> > > ### Randomness for 10 runs
> > > Thank you for pointing this out. We agree that there may still be some randomness across the 10 runs. However, we would argue that reporting the mean and standard deviation over 5 or 10 runs is standard practice. More importantly, our method shows consistently stronger performance across 9 real-world datasets, which makes it unlikely that the observed advantage is simply due to randomness.

---

### Official Review · Reviewer_6SYF · 2026-03-12

**Soundness:** 4
**Presentation:** 4
**Significance:** 3
**Originality:** 3
**Overall Recommendation:** 5
**Confidence:** 3

**Summary:**

The authors argue that explicitly modeling censoring and not just events in survival model training can improve predictions. They convincingly illustrate that is helpful when the event and censoring processes share parameters. They then propose a novel latent decomposition of covariates into event-specific, censoring-specific, shared/confounding, and irrelevant ones. They introduce SALaD as framework to do this, which can be applied on top of existing deep survival models. The authors evaluate SALaD and a simpler two branch approach extensively on a suite of 10 datasets and compare with against established and state of the art baselines with thorough cross validation and hyperparameter tuning.

**Compliance With Llm Reviewing Policy:**

Affirmed.

**Final Justification:**

The paper is technically sound, clearly written, and well motivated, and its main contribution, explicitly modeling censoring and learning decomposed latent factors for event, censoring, and shared information, is original and practically relevant for deep survival modeling. I found the empirical evaluation particularly strong, with broad benchmarking across 10 datasets and multiple base models, although the absolute gains are often modest, which tempers the significance somewhat. My main concerns were about the larger hyperparameter space of SALaD and the lack of nested cross-validation on smaller datasets; the rebuttal addressed these adequately by clarifying that all methods were given the same tuning budget and by candidly discussing the computational and methodological tradeoffs around more robust tuning. While these issues remain reasonable limitations, they do not undermine the overall contribution.

**Key Questions For Authors:**

* SALaD has more hyperparameters than the baseline methods. Can you comment on whether this might impact the evaluation?
* Have you considered a nested CV setup to improve robustness of the reported results, especially for the smaller datasets?

**Limitations:**

yes

**Strengths And Weaknesses:**

* The paper is very well motivated and written. I particularly liked the motivation and high-level illustration of the key ideas in sections 1 and 2.
* Figure 2 provides a very clear overview of the key differences between standard models, IPCW weighted models, 2B, and SALaD.
* Likewise, I found the t-SNE visualizations quite helpful and convincing that the proposed factorization does what it is supposed to do.
* The distinction between Assumption 2.2 and 2.3 was not obvious to me before but is very sensible.
* I also found the connections to causal and representation learning helpful and interesting.
* The empirical parts of the paper are excellent - The collection of datasets is extensive and, with the inclusion of the MIMIC-IV benchmark, goes beyond standard and commonly used small datasets in survival analysis papers.
* The paper convincingly shows that SALaD is a component that can be added on top of established deep survival models and tends to improve performance while not hurting calibration, although the improvements in performance are marginal in many cases.
* The related work section seems short given the breadth of deep learning for survival analysis.

---

> ### Author Rebuttal · Authors · 2026-03-31
>
> We thank the reviewer for the thoughtful and constructive comments. Below we address weaknesses [W] and questions [Q]:
>
> > [W1] Short related work section
>
> Due to the page limit, Section 2 focuses only on the works most directly related to our method, while also covering substantial survival-analysis background. To provide a broader picture, we included a much more comprehensive related-work discussion in Appendix B (page 15), which covers additional and more recent works that are less directly tied to our contribution.
>
> In addition, following the suggestion of Reviewer FciL [Q1], we have now added experiments with a copula-based model (DCSurvival). The revised manuscript will also expand the related-work discussion to better cover the copula-based line of research.
>
> > [Q1] SALaD has more hyperparameters than the baseline methods. Can you comment on whether this might impact the evaluation?
>
> We agree that SALaD has more hyperparameters than the naive baselines and the 2B extensions. However, its complexity is still comparable to many deep-learning and contrastive-learning survival models.
>
> To ensure a fair evaluation, we used the same random hyperparameter search budget of 60 trials for every model. Thus, although SALaD has a larger search space, it was not given a larger tuning budget. We used the same number of trials across all methods to keep the comparison fair. More details are provided in Appendix E.4 (page 30 and Table 4).
>
> > [Q2] Have you considered a nested CV setup to improve robustness of the reported results, especially for the smaller datasets?
>
> Thank you for the suggestion. We did not use nested cross-validation in the current paper.
>
> That said, during the development of SALaD, we were aware of the concern that additional hyperparameters may increase tuning burden. To reduce this burden, we also experimented with a multi-objective optimization method [1], where the weights of the regularization terms are treated as trainable parameters. In principle, this could remove the need to tune $\beta_1$, $\beta_2$, and $\lambda$, and thereby substantially reduce computation.
>
> However, in our experiments, this approach did not perform well (see the attached tables below for results on METABRIC, where "manual" indicates our current way of selecting hyperparameters, "auto" indicates using the multi-objective optimization method). Empirically, we observed this method tended to balance all four objective terms to similar magnitudes, whereas in our setting the likelihood term should usually receive substantially more weight, since the other three terms are regularizers. This appears to be a limitation of applying a generic multi-objective method here, as it tends to treat all objectives more equally than is desirable in our formulation.
>
>
>  Method | CI(%) $\uparrow$ | IBS $\downarrow$ | MAE-PO $\downarrow$ | D-cal $\uparrow$
> ---|---|---|---|---
>  **DeepSurv** |  |  |  |  |
>  SALaD (manual) | 69.33$\pm$3.68 | 0.1782$\pm$0.0290 | 3226.559$\pm$382.018 | 10/10
>  SALaD (auto) | 67.03$\pm$2.82 | 0.1829$\pm$0.0270 | 3260.120$\pm$349.537 | 10/10
>  **N-MTLR** |  |  |  |  |
>  SALaD (manual) | 69.48$\pm$2.82 | 0.1796$\pm$0.0266 | 3087.063$\pm$367.562 | 10/10
>  SALaD (auto) | 68.12$\pm$2.78 | 0.1936$\pm$0.0297 | 3235.406$\pm$343.138 | 8/10
>  **AFTNN-Weibull** |  |  |  |  |
>  SALaD (manual) | 69.06$\pm$3.73 | 0.2127$\pm$0.0264 | 9494.431$\pm$2143.021 | 10/10
>  SALaD (auto) | 66.32$\pm$3.88 | 0.1983$\pm$0.0169 | 4609.629$\pm$1304.007 | 0/10
>  **AFTNN-LogLogistic** |  |  |  |  |
>  SALaD (manual) | 69.56$\pm$3.77 | 0.2012$\pm$0.0199 | 7831.367$\pm$2562.350 | 8/10
>  SALaD (auto) | 66.32$\pm$4.03 | 0.2105$\pm$0.0171 | 7346.831$\pm$2547.849 | 1/10
>
> [1] Sener & Koltun, Multi-Task Learning as Multi-Objective Optimization, NeurIPS 2018

---

> > ### Author Rebuttal · Reviewer_6SYF · 2026-04-01
> >
> > Thank you for the thorough rebuttal. The authors have adequately addressed my concerns. I maintain my recommendation for acceptance.

---

> > > ### Author Response · Authors · 2026-04-07
> > >
> > > Thank you for confirming that our rebuttal addressed your concerns. We appreciate your time and thoughtful evaluation.

---

### Official Review · Reviewer_d89C · 2026-03-12

**Soundness:** 3
**Presentation:** 3
**Significance:** 3
**Originality:** 3
**Overall Recommendation:** 4
**Confidence:** 4

**Summary:**

The paper proposes a survival analysis approach that allows for the explicit modeling of the censoring distribution to reach better generalization performance. The idea is to use a shared representation first, and then try to discriminate between features concerning only time-to-event, features only related to censoring, and confounding features. The method is presented as four main propositions that are tackled and satisfied using different loss and regularization terms. The paper presents an empirical evaluation on 2 semi-synthetic and eight real-world datasets. The results show superior performance of SALaD compared to the SOTA methods.

**Compliance With Llm Reviewing Policy:**

Affirmed.

**Final Justification:**

I acknowledge the authors' rebuttal and as my score is already on the positive side, there will be no further changes.

**Key Questions For Authors:**

Questions:
- Can you show that methods like Cox or DeepSurv effectively depend on Assumption 2.3?
- I am wondering why you used only datasets with a small number of features. The largest was 93. How would the performance change if you used mRNA and miRNA from the TCGA data, such as the ones used in [1]?
- I also think that an analysis is needed to show how performance changes with different censoring levels. Also, the performance on higher censoring levels, such as 86% censoring on BRCA-miRNA [1].
- Can you explain why in the t-SNE diagrams, the censoring-specific representation does not show any separation with respect to the censoring time? Would that not be a more confirming result than looking at the separation with respect to event times?

[1] Shaker, Ammar, and Carolin Lawrence. "Multi-source survival domain adaptation." Proceedings of the AAAI conference on artificial intelligence. Vol. 37. No. 8. 2023.

**Limitations:**

Yes

**Strengths And Weaknesses:**

Strength:
- The problem of modeling the censoring distribution is of high importance to the machine learning survival analysis community. It has been investigated before, but was not put into practice as done in this work.
- The method is presented as a two-layered approach: first 2B, then SALaD. This presentation is well motivated and justified.
- The empirical evaluation shows that SALaD is superior on the used datasets compared to the used baselines.

Weakness:
- I agree that Assumption 2.3 does not hold, but disagree on the “censoring-free” term. I think the paper uses misleading phrases in a few places, such as framing previous work as censoring-free methods. The objective in (2) has no censoring model: no (f_C), no (S_C), no parameters for (C\mid X), but the data are still censored ((\delta_i = 0) for censored subjects). The likelihood (2) depends on censoring through those (\delta_i). It is “free” of the censoring mechanism as a modeled part of the likelihood—it does not estimate or use the distribution of (C). So I agree that (2) underlies that family of models and that it does not model censoring; I would avoid “censoring-free” without clarification and prefer something like “censoring-mechanism-free likelihood” to avoid implying that censoring plays no role at all.
- Similarly, the paper confuses covariates with the latent representation. In the abstract and a few other places, the paper says that the factorization is on the covariates; that is not accurate, since the decoupling occurs in the latent representation.
- I do not see what is new from a causal perspective in: “…censoring-induced bias can be viewed as arising from a spurious association between E and C through shared causes.” The result you aim for is already assumed in the independent censoring assumption (Assumption 2.2), also from a causal perspective. Your phrasing makes it sound as if you need (\Phi) to close the backdoor. What about Figure 2(b)—does not conditioning on X do the same thing by blocking the backdoors?

---

> ### Author Rebuttal · Authors · 2026-03-31
>
> We thank the reviewer for the thoughtful and constructive comments. Below we address weaknesses [W] and questions [Q]:
>
> > [W1] Censoring-free terminology
>
> Thanks to your helpful suggestion (that “censoring-free” can be misleading), the revised paper will use “censoring-mechanism-free likelihood” throughout.
>
> > [W2] Confusing covariates with latent representation
>
> Yes, the decomposition is not on the raw covariates themselves, but on the learned latent representation — we apologize for the confusion. We will revise the abstract and related text to make this precise.
>
> > [W3] Causal interpretation and the role of \Phi
>
> Thank you for pointing this out; we are glad you found the connection between survival and causal. In the manuscript, we try to connect these two domains to help readers understand the problem better.
>
> Under Assumption 2.2, conditioning on $X$ is already sufficient to block the backdoor path. Thus, our purpose in introducing $\Phi$ goes beyond backdoor adjustment. Rather, we aim to improve estimation when Assumption 2.3 is violated — that is, when the event and censoring processes share parameters — by showing that modeling censoring can help learn a better representation. In fact, Figure 2b, which already conditions on $X$, is sufficient to close the backdoor path, but it still relies on Assumption 2.3. We will revise the discussion accordingly to make this point clearer.
>
> > [Q1] Why do Cox & DeepSurv depend on Assumption 2.3
>
> Assumption 2.2 only justifies the factorization for the joint probability of $P(E, C \mid X)$ into two conditional probabilities $P(E \mid X) P(C \mid X)$. To further drop the censoring-mechanism terms and optimize only Eq. (2), those terms must be constant with respect to the parameters governing $P(E \mid X)$. That is exactly Assumption 2.3. Hence Cox-type methods, including DeepSurv, rely on Assumption 2.3 when trained with the standard event-mechanism-only likelihood.
>
> > [Q2, Q3] High-dimensional / high-censoring data.
>
> We agree that this is important. Following the reviewer's suggestion, the revised version will also consider the BRCA-miRNA dataset.
>
> This did not change the overall conclusion: in general the 2B and SALaD extensions are both better than the baselines (with SALaD wins in more cases). See table below:
>
>
>  Method | CI(%) $\uparrow$ | IBS $\downarrow$ | MAE-PO $\downarrow$ | D-cal $\uparrow$
> ---|---|---|---|---
>  Nnet-survival | 63.94$\pm$8.93 | 0.2382$\pm$0.1183 | 3024.977$\pm$1123.583 | 0/10
>  RSF | 53.49$\pm$10.29 | 0.1585$\pm$0.0303 | 3111.542$\pm$730.070 | 10/10
>  GB | 51.95$\pm$14.67 | 0.1388$\pm$0.0378 | 2723.210$\pm$748.627 | 10/10
>  DeepHit | 56.99$\pm$10.54 | 0.1976$\pm$0.0849 | 2916.085$\pm$928.315 | 2/10
>  CoxTime | 56.20$\pm$18.35 | 0.1502$\pm$0.0409 | 3683.281$\pm$3038.919 | 10/10
>  IWSG | 61.77$\pm$11.24 | 0.1644$\pm$0.0627 | 2786.526$\pm$1031.790 | 10/10
>  SODEN | 55.06$\pm$10.74 | 0.4039$\pm$0.1067 | 3994.819$\pm$762.994 | 0/10
>  CQRNN | 61.84$\pm$10.25 | 0.4619$\pm$0.1144 | 4005.691$\pm$894.918 | 0/10
>  SurvivalBoost | 57.72$\pm$10.34 | 0.1501$\pm$0.0962 | 2599.046$\pm$782.373 | 10/10
>  **DeepSurv** |  |  |  |  |
>  base | 54.46$\pm$14.54 | 0.1329$\pm$0.0420 | 2703.502$\pm$843.247 | 10/10
>  2B | 55.76$\pm$12.69 | 0.1263$\pm$0.0282 | 2751.079$\pm$762.778 | 10/10
>  SALaD | 60.74$\pm$5.40 | 0.1364$\pm$0.0416 | 2663.112$\pm$804.453 | 10/10
>  **N-MTLR** |  |  |  |  |
>  base | 63.25$\pm$9.75 | 0.1584$\pm$0.0540 | 3195.793$\pm$703.162 | 10/10
>  2B | 64.03$\pm$11.46 | 0.1508$\pm$0.0496 | 2614.482$\pm$770.318 | 10/10
>  SALaD | 64.65$\pm$10.76 | 0.1579$\pm$0.0318 | 2737.099$\pm$690.025 | 10/10
>  **AFTNN-Weibull** |  |  |  |  |
>  base | 59.46$\pm$15.25 | 0.2937$\pm$0.0664 | 22319.632$\pm$30362.188 | 10/10
>  2B | 57.07$\pm$14.91 | 0.2786$\pm$0.0618 | 17529.598$\pm$17605.689 | 10/10
>  SALaD | 59.93$\pm$10.89 | 0.2415$\pm$0.0447 | 6572.786$\pm$1701.783 | 10/10
>  **AFTNN-LogLogistic** |  |  |  |  |
>  base | 55.90$\pm$11.23 | 0.2498$\pm$0.0570 | 8220.222$\pm$4036.577 | 10/10
>  2B | 59.21$\pm$9.38 | 0.2574$\pm$0.0490 | 8600.992$\pm$1907.950 | 10/10
>  SALaD | 61.65$\pm$9.27 | 0.2665$\pm$0.0705 | 8646.507$\pm$5559.646 | 10/10
>
> > [Q4] t-SNE, the censoring-specific representation does not show any separation wrt the censoring time
>
> We believe the reviewer may be referring to the wrong subplot. In Figures 5&6, the censoring-specific representation $H^\gamma$ with respect to censoring time is shown in the bottom-right panel. By contrast, the top-right panel shows H^\gamma with respect to event time.
>
> If the reviewer was referring to the top-right panel, that would explain the concern. In the correct panels, however, the censoring-specific representations are clearly distinguishable — e.g. in Figure 6 (bottom right), the darker blob (earlier censoring instances) is located on the upper side.

---

> > ### Author Rebuttal · Reviewer_d89C · 2026-04-01
> >
> > I acknowledge the authors' rebuttal and as my score is already on the positive side, there will be no further changes.

---

> > > ### Author Response · Authors · 2026-04-07
> > >
> > > Thank you for confirming that our rebuttal addressed your concerns. We appreciate your time and thoughtful evaluation.

---

### Decision · Program_Chairs · 2026-04-30

**Decision:**

Accept (regular)

**Comment:**

The authors propose a new survival model (SALaD, for Survival Analysis via Latent Decomposed representation) that explicitly models the censoring mechanism (in a "missing not at random" fashion).

A strength of the method is that it is broadly applicable (they apply it to various survival models: DeepSurv, N-MTLR, and AFTNN).
They explore the theoretical gains of their approach (in particular in terms of reducing the asymptotic variance of maximum likelihood estimates). The empirical results are fairly compelling (and I particularly appreciated that they looked at the calibration of the survival models).

The most critical reviewer raised their score and indicated that their concerns were addressed, and there was a consensus among remaining reviewers that the paper should be accepted. I found the paper well-written and the results compelling, and am happy to recommend acceptance.